# What "Not" to Detect: Negation-Aware VLMs via Structured Reasoning and Token Merging

**Inha Kang**[1] **Youngsun Lim** [2] **Seonho Lee** [1] **Jiho Choi** [1] **Junsuk Choe** [3] **Hyunjung Shim**[1*]
[1]KAIST AI    [2]Boston University    [3]Sogang University

{rkswlsj13, glanceyes, jihochoi, kateshim}@kaist.ac.kr
youngsun@bu.edu, jschoe@sogang.ac.kr

## Abstract

State-of-the-art vision-language models (VLMs) suffer from a critical failure in understanding negation, often referred to as affirmative bias. This limitation is particularly severe in described object detection (DOD) tasks. To address this, we propose two primary contributions: (1) a new dataset pipeline and (2) a novel, lightweight adaptation recipe. First, we introduce CoVAND, a dataset constructed with a systematic chain-of-thought (CoT) and VQA-based pipeline to generate high-quality, instance-grounded negation data. Second, we propose NEGTOME, a novel text token merging module that directly tackles the architectural cause of affirmative bias. NEGTOME fundamentally addresses the structural loss of negation cues in tokenization, grouping them with attributes into coherent semantic phrases. It maintains correct polarity at the input level, enabling robust negation understanding even with limited data. For instance, to prevent a model from treating the fragmented tokens `not` and `girl` as simply `girl`, NEGTOME binds them into a single token whose meaning is correctly distinguished from that of `girl` alone. This module is integrated with a parameter-efficient and strategic LoRA fine-tuning approach. Our method significantly improves performance on challenging negation benchmarks with a lowered false positive rate, boosting NMS-AP by up to +10.8 points on OVDEval and demonstrating generalization to SoTA VLMs. This work marks a crucial step forward in addressing negation understanding for real-world detection applications.

## 1 Introduction

Even state-of-the-art Vision-Language Models (VLMs) exhibit a critical failure in understanding negation due to an affirmative bias (Alhamoud et al., 2025). This bias reflects a model's tendency to prioritize nouns while ignoring crucial negation cues. The issue is particularly pronounced in *described object detection* (DOD) (Xie et al., 2023; Schulter et al., 2023; Yao et al., 2024; Dang et al., 2023), a task requiring fine-grained compositional reasoning. As in Figure 1a, this bias causes models to treat phrases like *"person with skateboard"* and *"person **without** skateboard"* as semantically equivalent, leading to identical and incorrect detections. This failure extends to more complex logical structures, such as double negatives (*e.g., "not" + "un-"*). Since humans naturally use negation in natural communication (Sarabi & Blanco, 2016; Morante & Blanco, 2021; Beukeboom et al., 2020; Morante & Sporleder, 2012), failing to handle negation poses a serious barrier to real-world scenarios. This shortcoming can be particularly dangerous in safety-critical domains. For example, in medical imaging, misinterpreting the distinction between *"a tumor that is **not** malignant"* and *"a tumor that is malignant"* can lead to critical misdiagnoses. Therefore, bridging and improving negation understanding is an important step toward building robust VLM-based detection systems.

One key reason for the limited negation capability of VLMs is the lack of negated expressions in existing pre-training datasets. For example, large-scale datasets such as LAION-400M (Schuhmann et al., 2021) contain about $0.08\%$ negation words (Park et al., 2025). Likewise, Flickr30k (Plummer et al., 2015), a widely used captioning dataset, exhibits only $0.04\%$ negation words (Figure 1b). In contrast, negation is much more prevalent in real-world language. For instance, $13.76\%$ of words

---

*Corresponding author

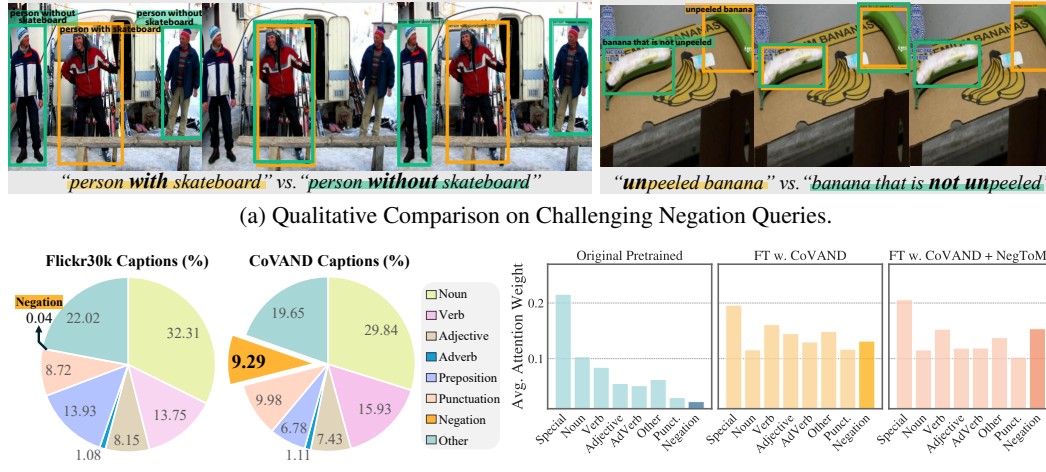

(a) Qualitative Comparison on Challenging Negation Queries.

(b) Word Class Distribution

(c) Average Attention Weight by Word Classes

Figure 1: **Challenges with Negation Expressions.** (a) Standard VLMs exhibit an affirmative bias, failing to distinguish contradictory negation queries. This issue stems from two causes: (b) the scarcity of negation words in standard datasets and (c) the model's tendency to assign low attention to negation cues. Our solutions, COVAND and the NEGTOME, directly address both problems.

in scientific papers (Szarvas et al., 2008) and 22.23% of words in Conan Doyle's stories involve negation (Morante & Daelemans, 2012). This imbalance results in VLMs that are poorly equipped to learn or attend to negation semantics.

To mitigate this limitation, we introduce a **c**hain-**o**f-thought with **V**QA **a**lignment for **n**egation **d**etection dataset (COVAND). It is a negation-focused training dataset constructed via chain-of-thought (CoT) reasoning and VQA-based caption alignment. To construct COVAND, we first extract both present and absent attributes from object regions. For each region, we then generate matched positive and negative captions using a CoT approach, followed by semantic verification using a VQA module. This process ensures each caption precisely reflects the presence or absence of key attributes, resulting in high-quality negation data pairs. As a result, our dataset provides a rich resource with 9.29% of negation words, a frequency 100× higher than that of typical datasets.

In addition to data-related factors, we observe that negation tokens receive notably lower attention weights, suggesting that current VLM detectors architecturally ignore or undervalue negation cues, as shown in Figure 1c. To counteract the low attention given to negation cues, the core of our method is NEGTOME, our novel text token merging module. It is designed to solve a key problem where standard tokenization often fragments phrases, separating negation cues (e.g., "not") from the attributes they modify (e.g., "lying"). NEGTOME addresses this by first merging these fragmented tokens into a single, coherent phrase. Through this binding, the negated concept of "not lying" can be learned as semantically distinct from "lying". This step strengthens the role of the attribute by ensuring it is always interpreted within its negated context. Crucially, this merged representation is enhanced with a negation-aware boost, explicitly amplifying the negated signal to ensure its polarity is preserved for downstream fusion. To our knowledge, this is the first work to employ a boosted token merging strategy for preserving semantic polarity in VLM-based detection.

To ensure the model effectively uses this enhanced text representation, we combine NEGTOME with a highly targeted application of Low-Rank Adaptation (LoRA). Our layer-wise attention analysis revealed that the negation signal dissipates before reaching the final decision-making blocks. Therefore, we apply LoRA to the deep cross-attention layers, the core of multimodal compositional understanding (Laurençon et al., 2024; Hertz et al., 2022). Together, this strategy modifies less than 0.1% of the model's parameters yet achieves a significant improvement in negation comprehension.

Our approach achieves state-of-the-art performance with 6.6 mAP on $D^3$ dataset, with 7.2 mAP improvement specifically on the challenging absence subset. In particular, our method not only increases the NMS-AP metric by 10.8 mAP but also reduces the false positive rate by 19.1%, demonstrating its enhanced ability to distinguish between contradictory queries. Importantly, these results are consistently observed across multiple distinct evaluation datasets, despite the model being

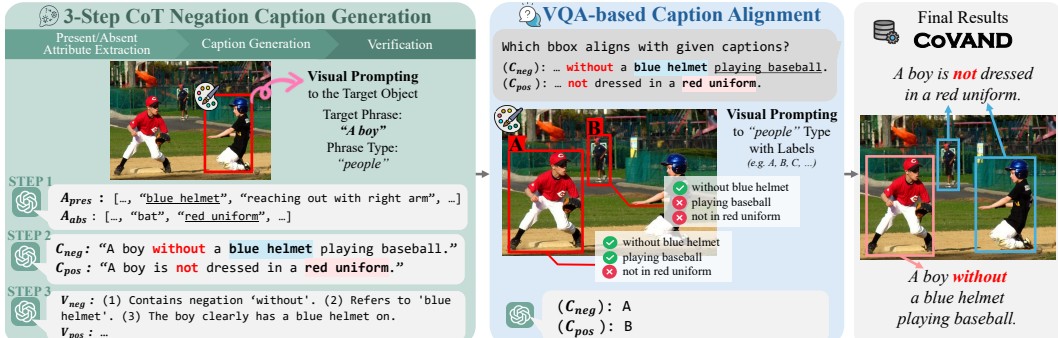

Figure 2: **Dataset Generation Pipeline of the CoVAND.** Our method first generates negation-focused captions for visually prompted regions using a three-step CoT process, then aligns each caption with the correct bounding box via VQA-based reasoning to ensure semantic correspondence.

trained solely on CoVAND. This highlights the strength of our approach and its superior generalization capability to unseen data and negation patterns.

Our work represents an initial yet substantial step toward robust negation understanding with the following key contributions:

- Our work presents CoVAND, a systematically generated dataset focusing on negation, to bridge a critical gap within existing multimodal benchmarks.

- We propose a novel adaptation recipe with NEGToME, our text token merging module that introduces a negation-aware boost to preserve semantic polarity.

- We achieve consistent gains across benchmarks, including +7.2 mAP on $D^3$ absence subset and +10.8 mAP on the NMS-AP metric in OVDEval's negation subset, demonstrating effective generalization to real-world negation scenarios.

## 2 CoVAND: DATASET GENERATION

To address the scarcity of negation data, we present CoVAND, a region-grounded negation dataset constructed through a multi-stage pipeline. As shown in Figure 2, the curation process consists of CoT caption generation followed by VQA-based alignment. This pipeline generates new high-quality captions that cover not only existence but also diverse attribute-based negations. In this way, CoVAND provides fine-grained, compositional supervision that trains detectors more robustly than only injecting templated or caption-level negations (Alhamoud et al., 2025; Park et al., 2025).

### 2.1 VISUAL PROMPTING WITH BOUNDING BOXES

Before caption generation, we apply visual prompting (Cai et al., 2024) to overlay a marker on the image. The marker specifies the region to describe and directs the CoT model's attention to that area. We apply this technique to bounding boxes in the Flickr30k Entities dataset (Plummer et al., 2015). For each image, we randomly choose two boxes linked to meaningful objects and exclude any box that spans a large background area to avoid ambiguity. Each selected region is then highlighted with a red bounding box and serves as an input image for region-grounded caption generation.

### 2.2 THREE-STEP CHAIN-OF-THOUGHT CAPTION GENERATION

We generate region-grounded paired negation captions through a three-step CoT process using GPT-4o (Hurst et al., 2024). We provide an explicit sequence that ensures consistent quality, rather than leaving it to the model's decision. The design follows the multi-step reasoning strategy of LLMs, where a complex visual query is split into ordered subtasks that improve factual accuracy and transparency. The input prompt for caption generation shows the image with a red bounding box, a target phrase such as "a boy" in "person" type. These cues fix the subject within the highlighted area and guide each reasoning step. The three steps are detailed below.

**Step 1: Present and Absent Attribute Extraction.** For each visually prompted region, we extract two sets of attributes: (1) *Present Attributes* ($A_{pres}$), consisting of attributes visibly present within the bounding box (e.g., colors, actions, relationships, actions, etc.), and (2) *Absent Attributes* ($A_{abs}$), representing relevant but missing attributes that could reasonably be expected. This rich attribute pool is the key novelty that lets our pipeline create attribute-level negations, which are far beyond the object-level attributes used in prior approaches (Alhamoud et al., 2025).

**Step 2: Negative and Positive Caption Generation.** We generate two types of paired captions using the extracted attributes:

- *Negative Caption* ($C_{neg}$): Incorrectly describes an attribute in $A_{pres}$ as absent (e.g., *"A man without a hat"* when "hat" $\in A_{pres}$).
- *Positive Caption* ($C_{pos}$): Correctly describes an attribute in $A_{abs}$ as absent (e.g., *"A woman without a red hoodie"* when "red hoodie" $\in A_{abs}$).

Each caption includes negation cues such as "no", "not", "never", "without", the prefix "un-", or the contraction "n't". The cue list is open to keep language natural and diverse.

**Step 3: Verification.** To ensure semantic consistency, we verify that $C_{pos}$ accurately describes the region while $C_{neg}$ contradicts it by asking GPT-4o. We also check whether generated captions contain negation words and attributes from step 1. If the pair fails on the test, it discards invalid captions and repeats caption generation until a valid pair appears or the retry limit is reached. This iterative guard preserves semantic integrity and keeps the quality of the overall dataset.

## 2.3 VQA-BASED CAPTION ALIGNMENT

The CoT stage produces a positive caption $C_{\text{pos}}$ and a negative caption $C_{\text{neg}}$ for each randomly chosen target box. However, label noise may still occur since another object of the same phrase type can also fit the captions. In Figure 2, for example, a person marked with "A" in the image could satisfy $C_{neg}$, even though it is not the designated target, which causes label noise. To eliminate this ambiguity, we add a dedicated region-level VQA alignment step.

First, we draw alphabetical labels on every box that shares the phrase type of the target. The target box stays unlabelled because it has already passed the in-context verification step. To determine the final alignment, we ask a VQA model two separate questions: "`Which labelled box aligns with` $C_{\text{pos}}/C_{\text{neg}}$?". Then, the VQA model simply answers with overlaid letters on the input images. While prior work used VQA for coarse, image-level validation (Park et al., 2025), their approach fails to resolve which specific instance a caption refers to. Our region-level alignment stage solves this ambiguity by requiring the VQA model to match each caption to a specific, visually-labeled bounding box, thereby delivering a more region-level ground truth.

Through this multi-stage process combining CoT reasoning and VQA alignment, COVAND provides rich training signals for negation understanding. We generate 91,110 captions with 23,876 images. In particular, our dataset exhibits approximately 9.29% negation word frequency, significantly higher than existing datasets like Flickr30k (0.04%). Detailed examples in Appendix A.

## 3 FINE-TUNING WITH NEGATION-SENSITIVE TEXT TOKEN MERGING

Our method addresses the two root causes of negation blindness: token fragmentation and low attention on negation cues. We propose a lightweight adaptation recipe that integrates our novel text token merging module, NEGTOME, with a targeted application of LoRA as in Figure 3.

### 3.1 NEGATION LORA ADAPTER

We apply LoRA following (Hu et al., 2022) with two key enhancements for vision-language fusion. Given frozen base weights $W_q, W_v \in \mathbb{R}^{d \times d}$ in cross-attention layers, we inject parallel adapters with an activation layer. Let $\sigma(\cdot)$ denote ReLU (Agarap, 2018) and let $A_q, A_v \in \mathbb{R}^{r \times d}$ and $B_q, B_v \in \mathbb{R}^{d \times r}$ be the trainable low-rank matrices. For an input $x \in \mathbb{R}^d$ we obtain

$$q = W_q x + \alpha B_q \sigma(A_q x), \quad v = W_v x + \alpha B_v \sigma(A_v x), \tag{1}$$

where $W_q, W_v \in \mathbb{R}^{d \times d}$ are the frozen base weights and $\alpha$ scales the LoRA update.

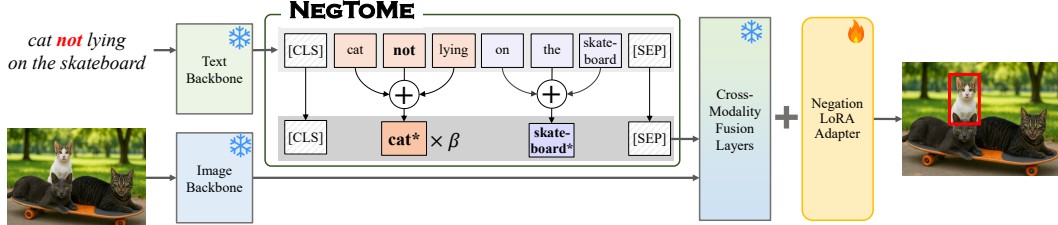

Figure 3: **Overview of Training Pipeline.** The input image and captions of CoVAND are encoded by frozen backbones. NegToMe assigns higher importance to negation cues in the text, and the LoRA adapter enables accurate localization of objects described by negated queries.

## 3.2 NegToMe: Semantic Text Token Merging for Negation Understanding

**Motivation.** While fine-tuning with negation-rich data can partially alleviate affirmative bias, it does not address a more fundamental flaw embedded in the model's tokenization process. Standard tokenizers inherently fragment phrases, separating negation cues (e.g., "not") from the words they modify (e.g., "lying"). This structural separation effectively causes the model to treat the phrase "not lying" as semantically equivalent to "lying", as the attention weight of the isolated negation tends to be ignored. To rectify this intrinsic information loss, we introduce NegToMe. It moves beyond data-level fixes to structurally ensure that a negated concept like "cat not lying" is represented as a single semantic unit, fundamentally distinct from {"cat", "not", "lying"}.

**Text Token Merging.** The caption is first split into sub-tokens $\mathcal{T} = \{t_1, \ldots, t_n\}$ by a standard tokenizer. To merge the tokens, an off-the-shelf parser then groups these tokens into disjoint phrase sets $\mathcal{P} = \{\mathcal{P}_1, \ldots, \mathcal{P}_m\}$ where $m < n$. For every phrase $\mathcal{P}_i \subseteq \mathcal{T}$, we compute one representative embedding by taking the normalized weighted average using fixed importance weights $\gamma_j$ of the sub-token vectors inside the phrase and replacing the original vectors with this average.

**Negation-aware Boost.** After merging, let $\mathcal{P}_{\text{neg}}$ be the phrase containing a cue (not, no, without, un-, etc.), and $\mathcal{I}_{\text{neg}} = \{ j \mid t_j \in \mathcal{P}_{\text{neg}} \}$ its index set. We assign a larger weight to the negation cue:

$$\bar{t}_{\text{neg}} = \frac{\sum_{j \in \mathcal{I}_{\text{neg}}} \gamma_j\, t_j}{\sum_{j \in \mathcal{I}_{\text{neg}}} \gamma_j}, \qquad \gamma_j = \begin{cases} \beta & \text{if } t_j \text{ is the negation cue,} \\ 1 & \text{otherwise,} \end{cases} \qquad \beta > 1. \tag{2}$$

The negation boosting factor $\beta$ amplifies the cue so that the merged embedding explicitly retains the negated meaning, improving polarity reasoning without increasing sequence length.

**Effect of Negation Boost on Representations.** Suppose the encoder maps a caption of $n$ sub-tokens to vectors $h_1, \ldots, h_n \in \mathbb{R}^d$. We write $h_c$ for the vector of the negation cue (e.g. "*not*") and $h_p$ for the vector of the predicate it modifies (e.g. "*moving*"). With vanilla mean pooling, the sentence embedding is $\bar{h} = \frac{1}{n} \sum_{i=1}^{n} h_i$, so the cue contributes only $s_{\text{single}} = \langle v, h_c \rangle / n$ to any linear probe $v \in \mathbb{R}^d$. After applying NegToMe, the merged representation of the negated phrase becomes $h_{\text{neg}} = \frac{\beta\, h_c + h_p}{\beta + 1}$ and the pooled vector gives $s_{\text{merge}} \geq \frac{\beta}{\beta+1} \langle v, h_c \rangle / m$, Hence

$$\frac{s_{\text{merge}}}{s_{\text{single}}} \geq \frac{\beta}{\beta + 1} \cdot \frac{n}{m}, \qquad 1 \leq m < n, \tag{3}$$

so the cue's influence is amplified by at least the factor $\frac{\beta}{\beta+1} \cdot \frac{n}{m}$. This gain aligns with the larger attention weights observed in Figure 1c and Figure S18, and experimentally show higher mAP.

## 4 Experiments

### 4.1 Experimental Setups

**Datasets.** DOD requires resolving compositional descriptions as in Figure 4a. To rigorously assess our model's ability to overcome the affirmative bias inherent in VLMs, we select two benchmarks

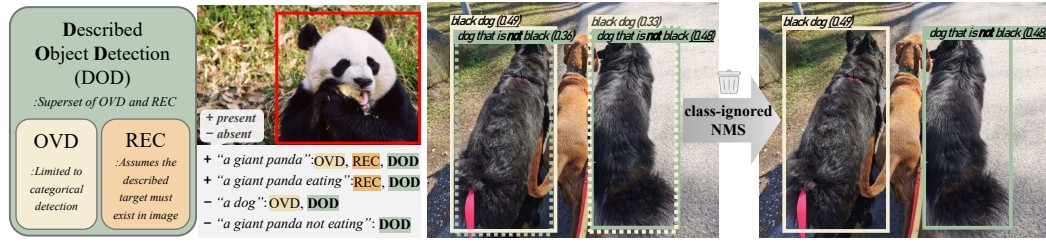

(a) Definition of DOD          (b) Process of NMS-AP score (Yao et al., 2024).

Figure 4: **Definition of Task and Metric.**

specifically designed to challenge negation understanding. We evaluate our method on two challenging DOD benchmarks for negation detection in VLMs. *Described Object Detection* ($D^3$) (Xie et al., 2023) introduces three evaluation protocols. *Pres* is a subset of 316 presence descriptions, *ABS* is 106 absence descriptions, and *Full* is an evaluation across all 422 descriptions. For *OVDEval Negation Subset* (Yao et al., 2024), we report both standard AP and the NMS-AP. The standard AP score can be misleadingly inflated when a model, confused by fragmented tokens, predicts overlapping boxes for contradictory pairs like *"black dog"* and *"dog that is **not** black"*. In contrast, NMS-AP (Yao et al., 2024) applies stricter filtering by removing overlapping predictions on contradictory pairs with IoU>0.5, effectively penalizing affirmative bias and accurately measuring negation understanding (Figure 4b). Additionally, we employ a practical yet challenging evaluation by performing class-ignored NMS separately after predicting each caption individually. (see the Appendix F.1.)

**Implementation Details.** We implement parameter-efficient fine-tuning through LoRA (Hu et al., 2022) applied to the deep cross-attention layers in the vision-language fusion module with $r = 4$. VLM-based detectors are trained for $5,000$ iterations with a batch size of $24$ for the Grounding DINO model, and $6,000$ iterations with a batch size of $4$ for the APE-Ti model. Training is conducted on two NVIDIA A6000 GPUs with mixed precision with a learning rate of $5 \times 10^{-4}$. Qwen-2.5-VL (Bai et al., 2025) is trained for 1 epoch batch size of 32 with a learning rate of $5 \times 10^{-5}$. All models are only trained with the COVAND dataset using the AdamW optimizer (Loshchilov & Hutter, 2017), freezing all backbone parameters except the LoRA layers. For NEGTOME, we use spaCy for the parser and set the negation boost factor $\beta = 2.0$. More details in the Appendix B.

## 4.2 EXPERIMENTAL RESULTS

**Quantitative Results.** As shown in Table 1, even powerful Multimodal Large Language Models (MLLMs) struggle with the $D^3$ benchmark. SoTA models like SPHINX-7B (Lin et al., 2023) and Qwen-2.5-VL-3B (Bai et al., 2025) achieve low performance on the full set (10.6 and 18.6 mAP,

Table 1: **Evaluation on the $D^3$ benchmarks.** Descriptions categorized by length; *S* for 1-3, *M* for 4-6, *L* for 7-9, and *XL* for 10+ words. *Pres* refers to present and *Abs* refers to absence subset.

| Method | Architecture | | | $D^3$ (default) | | | $D^3$ (by length of texts) | | | |
|---|---|---|---|---|---|---|---|---|---|---|
| | Backbone | Text Encoder | Detection Head | Full | Pres | Abs | S | M | L | XL |
| OFA-L | ResNet-101+ViT | BART | Seq2Seq | 4.2 | 4.1 | 4.6 | 4.9 | 5.4 | 3.0 | 2.1 |
| OWL-ViT-L | ViT-L | CLIP | OWL-ViT | 9.6 | 10.7 | 6.4 | 20.7 | 9.4 | 6.0 | 5.3 |
| SPHINX-7B | CLIP,DINO-v2, Q-Former | LLaMA-2 | – | 10.6 | 11.4 | 7.9 | 16.8 | 13.8 | 5.6 | 3.1 |
| OFA-DOD | ResNet-101+ViT | BART | Seq2Seq | 21.6 | 23.7 | 15.4 | 23.6 | 22.6 | 20.5 | 18.4 |
| GLIP-T | | | | 19.1 | 18.3 | 21.5 | 22.4 | 22.0 | 16.6 | 10.6 |
| + GEN | Swin-T | BERT | DyHead | 21.4 | 20.6 | 23.7 | 28.1 | 24.5 | 17.4 | 11.5 |
| + W2S | | | | 26.0 | 25.6 | 27.1 | - | - | - | - |
| FIBER-B | | | | 22.7 | 21.5 | 26.0 | 30.1 | 25.9 | 17.9 | 13.1 |
| + GEN | Swin-B | RoBERTa-B | DyHead | 26.0 | 25.2 | 28.1 | 35.5 | 29.7 | 20.5 | 14.2 |
| + W2S | | | | 26.5 | 26.0 | 27.7 | - | - | - | - |
| G-DINO-B | | | | 20.7 | 20.1 | 22.5 | 22.6 | 22.5 | 18.9 | 16.5 |
| + Ours | Swin-B | BERT | DINO | 27.3 | 26.4 | 29.7 | 29.9 | 29.5 | 25.2 | 21.3 |
| (↑ Δ) | | | | (+6.6) | (+6.3) | (+7.2) | (+7.3) | (+7.0) | (+6.3) | (+4.8) |
| APE-Ti | | | | 29.1 | 29.9 | 26.9 | 31.1 | 31.9 | 27.4 | 21.4 |
| + Ours | ViT-Ti | CLIP | DETA | 32.5 | 32.9 | 31.5 | 33.2 | 35.3 | 31.3 | 25.4 |
| (↑ Δ) | | | | (+3.4) | (+3.0) | (+4.6) | (+2.1) | (+3.4) | (+3.9) | (+4.0) |
| Qwen-2.5-VL-3B | | | | 18.6 | 18.5 | 19.2 | 18.2 | 20.7 | 17.0 | 16.0 |
| + Ours | ViT-H | Qwen-2.5 | – | 22.2 | 22.8 | 20.6 | 19.8 | 25.8 | 20.2 | 17.8 |
| (↑ Δ) | | | | (+3.6) | (+4.3) | (+1.4) | (+1.6) | (+5.1) | (+3.2) | (+1.8) |

respectively), and their slow inference makes them impractical for many detection scenarios. In contrast, our lightweight adaptation recipe significantly boosts the performance of strong detector baselines. When applied to Grounding-DINO, our method improves the overall mAP by +6.6 points, with a notable gain of **+7.2 mAP** on the challenging absence subset. This performance gain is direct evidence of a more robust understanding of semantic polarity. Baseline models often generate false positives because they fail to distinguish between conceptually opposite phrases like "*with a hat*" and "*without a hat*". As a specific absence scenario, when prompted with "*a person without a hat*" in an image where everyone is wearing one, they would incorrectly detect a person. Our tokenizer modification, NEGTOME, resolves this by forcing the model to process the negated phrase as a single semantic unit with distinct polarity, enabling it to correctly reject such invalid instances. Similarly, on APE-Ti, we achieve a +4.6 mAP improvement on the absence subset, demonstrating an enhanced ability to reject non-existent objects. Notably, these gains are comparable to computationally expensive, large-scale fine-tuning methods (Zhao et al., 2024a; Park et al., 2024b) while updating less than 0.1% of the model's parameters only with our COVAND dataset. The improvements are also consistent across all description lengths, validating the robustness of our approach. Furthermore, preliminary experiments demonstrate the generalizability of our method to MLLMs, with an improvement of +3.6 mAP on Qwen-2.5-VL-3B.

Even powerful SoTA MLLMs struggle on the challenging OVDEval-Negation subset, demonstrating that simply applying a large-scale model is not a sufficient solution for negation. Notably, as shown in Table 2, the powerful Qwen-2.5-VL-7B underperforms the much smaller Grounding-DINO baseline, highlighting the difficulty of the task. In contrast, our lightweight adaptation recipe yields significant performance gains across all tested architectures, particularly on the stricter NMS-AP metric. Our method boosts the Grounding-DINO by a substantial **+10.8** mAP in NMS-AP and improves the Qwen-2.5-VL-3B by +7.3 in mAP and +3.8 in NMS-AP. For the MLLM, the substantial AP gain is significant because

Table 2: **Results on OVDEval-Negation**. † means reproduced AP.

|  | AP | NMS-AP |
| --- | --- | --- |
| G-DINO-B† | 54.0 | 36.8 |
| + Ours | 57.2 | 47.6 |
| (↑ Δ) | (+3.2) | (+10.8) |
| APE-Ti | 50.5 | 32.3 |
| + Ours | 54.1 | 33.5 |
| (↑ Δ) | (+3.6) | (+1.2) |
| Qwen-2.5-VL-7B | 37.8 | 35.9 |
| Qwen-2.5-VL-3B | 34.6 | 31.3 |
| + Ours | 41.9 | 35.1 |
| (↑ Δ) | (+7.3) | (+3.8) |

it enhances both negation reasoning and foundational localization, a typical weakness of such models. Further results, including a detailed comparison with two-stage post-hoc VQA with MLLM and a full evaluation across all OVDEval subsets, are available in Appendix E and F.

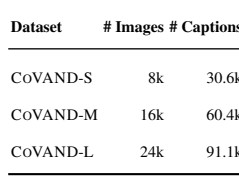

| Dataset | # Images | # Captions |
| --- | --- | --- |
| COVAND-S | 8k | 30.6k |
| COVAND-M | 16k | 60.4k |
| COVAND-L | 24k | 91.1k |

(a) COVAND splits.

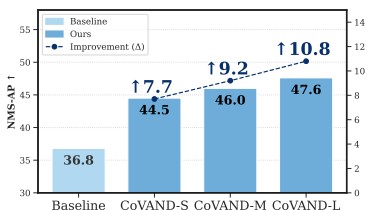 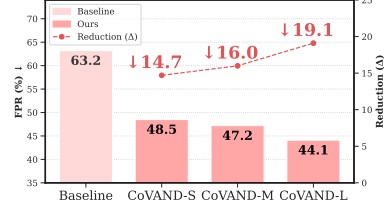

(b) Dataset Scalability. NMS-AP (left) and FPR (right).

Figure 5: **Dataset Statistics and Performance Scaling.** (a) Statistics for our three COVAND splits. (b) Bar plots with **blue** refer to NMS-AP and **pink** refer to FPR (lower is better).

**Dataset Scalability.** Figure 5 presents our scalability analysis of the dataset on the OVDEval-Negation subset. We observe a consistent improvement as we scale the COVAND dataset from small to large. Specifically, NMS-AP improves from 44.5 to 47.6, while the FPR decreases from 48.5% to 44.1%, which is a total reduction of **19.1** points from the baseline. This trend of simultaneously improving NMS-AP, a metric that penalizes contradictory predictions, while lowering FPR, which measures the failure to reject absent objects, shows the effectiveness of our approach.

**Qualitative Results.** Figure 6 presents qualitative results from the OVDEval dataset comparing our fine-tuned Grounding DINO model against the baseline. The baseline model often exhibits a strong affirmative bias, frequently collapsing contradictory captions into the same prediction. Our model, however, successfully handles these complexities across various patterns. For instance, it accurately

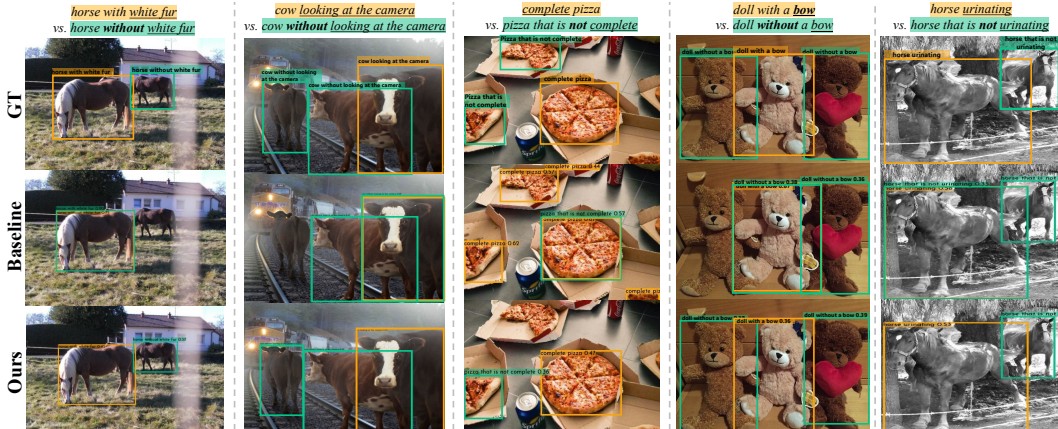

Figure 6: **Qualitative Comparison on the OVDEval Negation Subset.** Our model correctly distinguishes the polarity of contradictory caption pairs, overcoming the baseline's affirmative bias.

Table 3: **Ablation Study.** Best in `blue` and worst in `red`. LoRA adapters are inserted at three fusion-block depths: `shallow` (blocks 0–2), `strided` (1, 3, 5), and `deep` (3–5).

| Training Data | Settings LoRA Placement | NEGTOME | $\beta$ | OVDEval (Negation Subset) AP | NMS-AP | AR | NMS-AR | ↓FPR | D³ Full | Pres | Abs | ↓FPR |
|---|---|---|---|---|---|---|---|---|---|---|---|---|
| **Pretrained Weight** | | | | 54.0 | 36.8 | 20.5 | 14.7 | 63.2 | 20.7 | 20.1 | 22.5 | 67.2 |
| Flickr30k | shallow | ✗ | – | 55.9 | 38.5 | 21.7 | 15.2 | 61.3 | 18.4 | 18.2 | 23.0 | 66.5 |
| Flickr30k | strided | ✗ | – | 54.8 | 36.5 | 20.5 | 14.1 | 62.6 | 20.9 | 19.9 | 24.0 | 68.2 |
| Flickr30k | deep | ✗ | – | 53.7 | 31.8 | 20.7 | 12.8 | 59.9 | 22.0 | 21.0 | 24.8 | 67.8 |
| COVAND-S | shallow | ✗ | – | 46.8 | 31.5 | 21.9 | 14.8 | 56.0 | 18.5 | 17.6 | 21.0 | 63.9 |
| COVAND-S | strided | ✗ | – | 52.8 | 43.9 | 20.0 | 17.1 | 49.0 | 20.1 | 19.2 | 22.9 | 63.4 |
| COVAND-S | deep | ✗ | – | 55.4 | 41.8 | 21.4 | 18.0 | 48.6 | 24.2 | 23.0 | 27.0 | 64.0 |
| COVAND-S | deep | ✔ | 1.0 | 57.8 | 43.8 | 24.0 | 19.6 | 50.8 | 25.7 | 25.1 | 27.3 | 63.7 |
| COVAND-S | deep | ✔ | 2.0 | 58.7 | 44.5 | 24.1 | 19.2 | 48.5 | 26.2 | 25.4 | 28.2 | 63.3 |

identifies the "*cow without looking at the camera*" and the "*horse that is not urinating*", proving it can ground negation in complex contexts. Moreover, for "*banana that is not unpeeled*", it correctly identifies the peeled banana by resolving the "not" + "un-" double negative as in Figure 1a. Our model sometimes fails to detect every target instance, for example "*pizza that is not complete*", its predictions are a marked improvement over the baseline, which provides completely unreliable detections for both queries. Together, these examples show that our method achieves a more compositional understanding of negation. Further qualitative results on OVDEval and $D^3$ are presented in Figure S23–S24 and Figure S25–S27, respectively.

## 4.3 ABLATION STUDY

Our ablation study, summarized in Table 3, reveals the impact of each component, with attention diagnostics in Figure S18 in the Appendix providing a clear mechanism for the improvements. Placing LoRA adapters in the `deep` fusion blocks consistently outperforms `shallow`. This is because `deep` placement maintains elevated attention on negation tokens in the later blocks where decisions are formed, whereas the effect of `shallow` placement dissipates too early. Furthermore, training with COVAND dataset yields substantial gains over generic captions, demonstrating its value for both accuracy and generalization. Finally, adding NEGTOME with its negation boost factor provides large gains, such as a **+2.7** improvement in NMS-AP. This trend is mirrored on the $D^3$ benchmark. While using our COVAND dataset alone yields a +2.2 mAP improvement over the baseline, NEGTOME adds a further +2.0 mAP on top. This near-equal contribution highlights that our token merging strategy is as impactful as the dataset itself. The attention analysis further confirms that NEGTOME directly causes this improvement by increasing attention to the negated phrase. Together, these results motivate our final design that locates adaptation late in the fusion stack and explicitly increases negation cues to counteract affirmative bias.

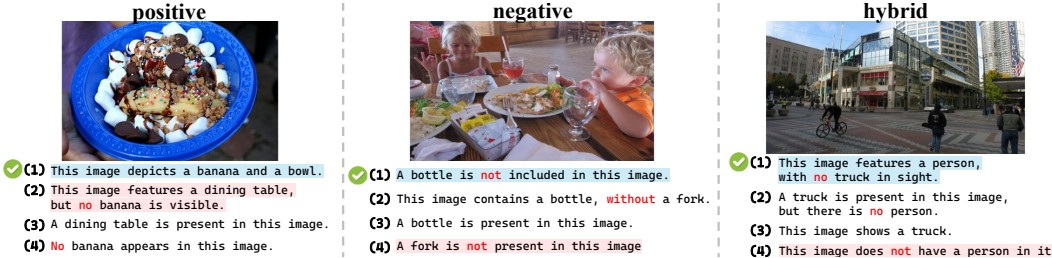

**Figure 7: Qualitative Comparison on the NegBench MCQ Benchmark.** Captions with green checkmark ✅ is **GT**, pink refer to **Baseline**, and blue refer to **Ours**.

## 4.4 ZERO-SHOT DOWNSTREAM EVALUATION OF SEMANTIC COMPREHENSION.

To verify our method achieves a semantic understanding of negation that generalizes beyond detection, we evaluate it on the NegBench COCO subset of Multiple Choice Question (MCQ) benchmark (Alhamoud et al., 2025). This task requires the model to select the most accurate caption for an image from four options. These options include three subsets: 'Positive' correctly affirming present objects (e.g., "*A and B*"), 'Negative' correctly negating absent ones (e.g., "*not B*"), and 'Hybrid' that combine both types (e.g., "*A but not B*"). In a zero-shot setting, we select the caption that produces the highest max-logit score when grounded in the image. As shown in Table 4, our method improves accuracy over the baseline with a +10.86% improvement. This result provides strong evidence that our approach enhances a robust understanding of negation. We present qualitative examples in Figure 7 and in Appendix H.

Table 4: **Results on the NegBench Multiple Choice Question (MCQ) benchmark.**

| Model | Overall Acc. | Positive | Negative | Hybrid |
|---|---|---|---|---|
| CLIP-OpenAI | 16.27 % | — | — | — |
| NegCLIP | 10.21 % | — | — | — |
| G-DINO-B | 21.69 % | 27.36 % | 13.37 % | 23.71 % |
| + Ours | **32.55** % | **46.85** % | **23.37** % | **26.64** % |
| (↑ Δ) | (+10.86) | (+19.49) | (+10.00) | (+2.93) |

## 4.5 ZERO-SHOT GENERALIZATION ON BIOMEDICAL DOMAIN

To validate that our method learns a robust negation mechanism rather than merely memorizing the training data, we conducted a zero-shot evaluation on the biomedical domain using the FG-CXR dataset. This domain presents significant generalization challenges due to its distinct visual features (e.g., grayscale X-rays) and unique negation taxonomy.

Table 5: **Zero-shot Results on FG-CXR.**

| Method | Accuracy |
|---|---|
| Baseline (G-DINO-B) | 54.86% |
| **+ Ours (NEGTOME)** | **62.55%** |

We formulated a zero-shot binary discrimination task by generating hard negative contradictions for each GT diagnosis through rule-based polarity flipping (e.g., presence vs. absence of disease). The model was evaluated on its ability to assign a higher matching score to the GT caption than to the hard negative. While the baseline model (G-DINO-B) struggled with a near-random accuracy of 54.86%, our NEGTOME method achieved 62.55% (+7.69%). Since our model was never exposed to medical data during fine-tuning, this substantial improvement demonstrates that NEGTOME effectively structuralizes the binding between negation cues and their targets, enabling robust generalization to entirely unseen domains. Details are in I.

# 5 RELATED WORK

## 5.1 OBJECT DETECTION

OVD extends classical detectors to arbitrary text labels (Zareian et al., 2021; Yao et al., 2022; Kim et al., 2023a; Chen et al., 2025d). Methods such as GLIP (Li et al., 2022), FIBER (Dou et al., 2022), and APE (Shen et al., 2024) fuse language either in the detection head, in the backbone, or in a task-general prompt module, and achieve strong zero-shot performance. REC adds compositional phrases. Grounding DINO (Liu et al., 2024b) proposes DETR-style decoders that localize the described object without category supervision. Despite this progress, REC models still assume the target exists and therefore struggle to reject absent or negated descriptions. DOD (Xie et al.,

2023) generalizes OVD and REC by requiring the detector to decide both existence and location. Benchmarks such as $D^3$ and OVDEval (Yao et al., 2024) reveal a low in accuracy on absence or negation subsets. It confirms that current VLMs often have an affirmative bias on negation cues. MLLM (Lin et al., 2023; Bai et al., 2025) have recently been applied to DOD, but their accuracy fails to surpass that of VLM-based detectors, their performance on negation remains low, and their inference speed is incompatible with real-time detection scenarios. We tackle negation explicitly by (1) introducing CoVAND, a high-coverage negation dataset, and (2) proposing a lightweight LoRA and NEGTOME recipe that plugs into VLM decoders. This design yields higher mAP and lower FPR on both $D^3$ and OVDEval, thereby overcoming the limitations of prior approaches (Zhao et al., 2024a; Park et al., 2024b).

## 5.2 NEGATION UNDERSTANDING IN VISION-LANGUAGE MODELS

CLIP-based studies such as NegBench (Alhamoud et al., 2025) reveal the affirmative bais that state-of-the-art VLMs often treat *"dog"* and *"not dog"* identically; subsequent fixes like Negation-CLIP (Park et al., 2025) simply augment pre-training with template-level negation pairs and thus miss context-dependent or region-grounded cases. We instead build a fine-grained dataset with CoT reasoning and VQA alignment, producing positive and negative caption pairs that are grounded to target boxes, and show that this richer supervision transfers to multiple architectures beyond CLIP.

## 5.3 TEXT TOKEN-LEVEL MERGING

Token Merging (ToMe) (Bolya et al., 2022) merges similar image tokens to accelerate inference without sacrificing accuracy. It is a technique originally proposed for Vision Transformers (ViTs), where similar image tokens are merged to accelerate inference without sacrificing accuracy. ToMe is extended to diffusion and grounding models, where token merging based on semantic phrase is introduced to mitigate the loss of modifier information (Hu et al., 2024; Li et al., 2024b). In the context of OVD, there have been attempts to merge image tokens (Su et al., 2024; Norouzi et al., 2024), but the merging of text tokens has been unexplored. Previous studies on text token merging have primarily focused on diffusion models, particularly in text-to-image generation (Hu et al., 2024). In this work, we are the first to explore text token merging in detection models and empirically demonstrate its feasibility and effectiveness.

## 6 CONCLUSION

This work presents a comprehensive solution to the affirmative bias that hinders negation understanding in VLMs by addressing its two root causes. To resolve data scarcity, we introduce CoVAND, a systematic pipeline using CoT reasoning and VQA-based alignment to generate high-quality, instance-grounded negation data. To counteract the model's architectural tendency to ignore negation cues, we propose NEGTOME, a novel module that, to our knowledge, is the first to use a negation-aware boost to preserve semantic polarity in detection tasks. Our parameter-efficient recipe integrates these contributions to achieve substantial gains on challenging negation benchmarks and demonstrate strong generalization across VLM-based detectors and MLLMs, marking a significant step towards VLMs that can understand not only what is present, but also what is absent.

## ACKNOWLEDGMENTS

This research was supported by the Basic Science Research Program through the National Research Foundation of Korea (NRF) funded by the MSIP (No. RS-2025-00520207); the Institute of Information & Communications Technology Planning & Evaluation (IITP) grants funded by the Korea government (MSIT) (Nos. 2022-0-00680, 2022-0-01045, RS-2024-00457882, RS-2025-02217259, RS-2019-II190075), including the National AI Research Lab Project and the Artificial Intelligence Graduate School Support Program (KAIST); and the Korea Evaluation Institute of Industrial Technology (KEIT) grants funded by the Korea government (MOTIE) (Nos. 2022-0-00680, 2022-0-01045, RS-2025-02217259).

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

## SUPPLEMENTARY MATERIALS

We provide supplementary materials in the following order:

- Section A: **CoVAND Details** describing our negation-focused dataset construction pipeline, including the three-step Chain-of-Thought prompt design, VQA-based caption alignment, negation cue distribution, and a human-in-the-loop data error analysis.
- Section B: **Implementation Details** of all backbone architectures and our LoRA placement strategy, together with additional attention visualizations.
- Section C: **Extended Related Work** covering CoT-based dataset construction, visual grounding and region-level alignment, parameter-efficient fine-tuning for VLMs, compositional reasoning, and bias mitigation.
- Section D: **Additional Ablations: Negation- and Noun-Only Boosting**, which compare simple token-level boosting and attention-bias variants against our NEGTOME.
- Section E: **Comparison with Post-hoc VQA Methods** analyzing two-stage detector+VQA pipelines and their accuracy–latency trade-offs.
- Section F: **Evaluation on Full OVDEval Subsets** reporting results on all OVDEval subset.
- Section G: **Analysis on RPN-based Detectors** contrasting RPN-based and DETR-style architectures under negation.
- Section H: **Zero-shot Downstream NegBench MCQ** presenting a detailed breakdown of the multiple-choice subsets and characteristic error patterns.
- Section I: **Zero-shot Generalization on the Biomedical Domain** evaluating our method on the FG-CXR chest X-ray dataset and analyzing cross-domain.
- Section J: **Qualitative Results** displaying additional examples on OVDEval and $D^3$, as well as representative failure cases on complex negation and event-level reasoning.
- Section K: **Declarations** summarizing LLM usage, ethics, and reproducibility.

## A DETAILS ON CoVAND

### A.1 PROMPT FOR THREE-STEP CoT CAPTION GENERATION

We employ a systematic three-step CoT reasoning approach using GPT-4o (Hurst et al., 2024) to generate high-quality negation-focused captions. As shown in Figure S11, the prompt structure is carefully designed to elicit temporally coherent reasoning that produces semantically valid negation captions grounded in the visual content.

Our prompt begins by informing the model that it will be provided with an image containing a highlighted bounding box, along with a target phrase describing the main subject in the region. The model is then guided through three distinct reasoning steps:

### A.1.1 STEP 1: ATTRIBUTE EXTRACTION

The model first generates two comprehensive lists of attributes:

- **Present Attribute** ($A_{pres}$): At least three attributes or keyword items clearly visible within the bounded region.
- **Absent Attribute** ($A_{abs}$): At least three attributes or keyword items that are contextually relevant but clearly not present in the bounded region.

### A.1.2 STEP 2: CAPTION GENERATION

Using the attributes from Step 1, the model produces two types of captions:

- **Negative Caption** ($C_{neg}$): Creates a factually incorrect statement by falsely claiming an existing attribute is absent. This caption must contain a negation expression (e.g., "no", "not", "without") coupled with an attribute from the existing contents list.

- **Positive Caption** ($C_{pos}$): Creates a factually correct statement by accurately describing an absent attribute as absent. This caption pairs a negation expression with an attribute from the absent contents list.

This approach yields contrastive pairs where the negative caption contradicts the visual evidence while the positive caption aligns with it, creating training data that specifically targets negation understanding.

### A.1.3   STEP 3: SEMANTIC VERIFICATION

For quality assurance, each generated caption undergoes verification:

- **Negative Verification**: Confirms the caption (1) contains a negation expression, (2) references an existing attribute from Step 1, and (3) factually mismatches the actual content of the bounded region.
- **Positive Verification**: Confirms the caption (1) contains a negation expression, (2) references an absent attribute from Step 1, and (3) correctly describes the absence of the attribute in a way relevant to the context.

This verification step ensures semantic integrity and prevents generation artifacts by applying explicit logical checks. If either caption fails verification, the process iteratively regenerates captions until valid pairs are produced or the retry limit is reached.

The prompt enforces concise, natural language expressions with a single-sentence structure. As examples in Figure S12 and Figure S13, it requires the model to focus exclusively on the bounded region, preventing semantic drift to other parts of the image. The entire process outputs a structured JSON format containing the attribute lists, caption pairs, and verification rationales, facilitating downstream dataset creation and quality control processes.

### A.2   VQA-BASED CAPTION ALIGNMENT

To address a critical challenge in negation-aware detection, ensuring generated captions reference exclusively the intended bounding box rather than other visually similar regions, we implement a structured verification pipeline with VQA alignment.

First, we apply alphabetical region labeling to all bounding boxes that share the target phrase type (e.g., "person") by assigning distinct markers (`A, B, C, ...`) to each instance. The originally prompted region remains unlabeled to avoid biasing the verification process. As shown in Figure S14, our visual prompting approach carefully considers label placement to maintain visual clarity. When labeling multiple instances of the same type (e.g., multiple "person" boxes), we position alphabetical markers outside the top-left corner of each bounding box to avoid occluding the object itself. This placement strategy preserves the visual integrity of the object while providing clear reference points for the VQA model. In cases where objects appear near image boundaries, we adaptively place labels inside the top-left corner of the bounding box to ensure they remain visible within the frame. This adaptive positioning is crucial for maintaining consistent label visibility across diverse image compositions.

Then, for each caption pair $(C_{pos}, C_{neg})$, we query a multimodal VQA model with two precisely formulated questions as in Figure S15. The VQA model analyzes the image and captions to produce structured JSON responses specifying matching box labels. A valid alignment requires that $C_{pos}$ matches *exactly* the original unlabeled region, while $C_{neg}$ either matches no regions (``None'') or incorrectly matches another box. This process effectively eliminates label noises: false negatives, where $C_{neg}$ accidentally describes another instance, and ambiguous groundings, where captions generically describe multiple regions.

Figure S16 showcases several successful examples from our complete caption generation pipeline. In these examples, we can observe how the three-step CoT process first generates attribute-based negative and positive captions for the target region, followed by the VQA alignment step that verifies caption-region correspondence. Despite the effectiveness of our approach, we encountered certain limitations in complex scenes, as illustrated in Figure S17. When multiple instances of the same type are densely clustered, the visual prompting can become ambiguous, making it difficult for the

VQA model to determine precise correspondences. To maintain dataset quality, we implemented a filtering mechanism that excludes images containing more than five instances of the same type from the caption generation process. This threshold was empirically determined to balance the diversity of the dataset with the precision of the annotation, ensuring that our training data provides unambiguous supervision signals for understanding the meaning of negations.

## A.3 DATASET DISTRIBUTION

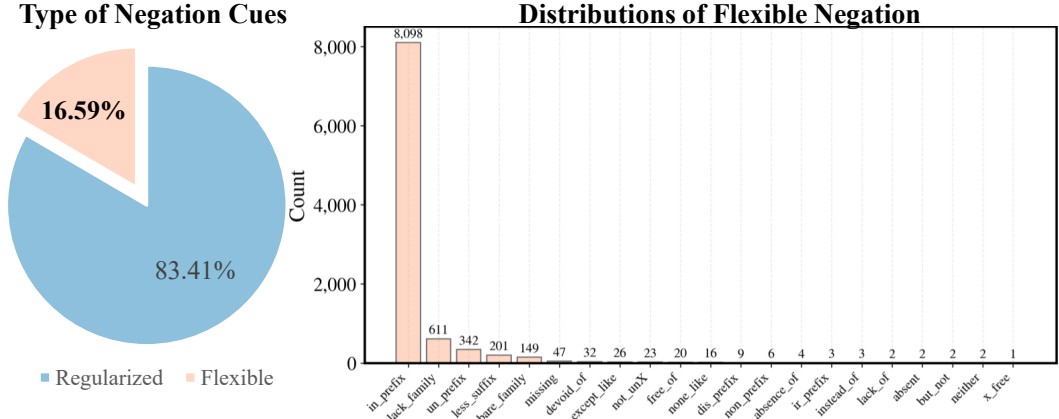

Figure S8: **Distributions of Negation Type.** Analyzing all 48,761 captions in CoVAND and identified 57,874 negation instances. Following standard linguistic taxonomies, we categorized them into Regularized (explicit syntactic markers) and Flexible (lexical/morphological) cues. Regularized Cues means high-frequency surface markers, including not, no, without, never, and contractions like n't. Flexible Cues means a diverse long-tail of expressions including lack, un-, in-/im-/ir-, dis-, non-, and -less.

We separate surface **regularized cues** versus **flexible cues** as below:

- **Regularized cues** are short, high-frequency surface markers: `not`, `without`, `no`, `never`, and clitic contractions `n't`.

- **Flexible cues** cover lexical and morphological forms that naturally occur in open text: *lack*-family (`lack`, `lacks`, `lacking`, `lack of`), `devoid of`, `absence of`/`absent`, coordinations (`neither`/`nor`, `but not`, `rather than`, `instead of`), and productive morphology such as negative prefixes/suffixes (`in-`/`im-`/`il-`/`ir-`, `un-`, `dis-`, `non-`), and `-less`, as well as `X-free`/`free of`.

We analyze all 48,761 captions (24,381 *positive* vs. 24,380 *negative*). Across all captions, we detect 57,874 negation cues in total: **48,275 regularized** (83.41%) and **9,599 flexible** (16.59%).

Prior analyses of negation in natural language understanding corpora show that explicit markers such as `not`, `no`, and `n't` account for the large majority of negation instances. Hossain et al. (2022) found that syntactic negation (regularized cues) constitutes 88.6% in CommonsenseQA (Talmor et al., 2019) and 71.9% in SST-2 (Socher et al., 2013), compared to morphological negation (11.4% and 28.1%, respectively) as in (Hossain et al., 2022). While these figures come from specific NLU datasets rather than unrestricted natural language, they suggest that regularized forms are prevalent in realistic language tasks. Thus, our dataset's 83.41% regularized distribution is not solely an artifact of GPT-4o's generation bias, but rather aligns with patterns observed in existing negation-annotated corpora for downstream applications.

**Flexible forms provide meaningful diversity.** While regularized cues dominate, the presence of 9,599 flexible cues (16.59%) ensures the dataset includes a non-trivial variety of negation expressions. This diversity is essential for evaluating whether models generalize beyond high-frequency patterns. Flexible negations, though less common, are critical in compositional reasoning tasks such as DOD, where attribute-level and relational negations often require nuanced understanding. By

including both regularized and flexible forms, CoVAND provides a more comprehensive training signal than datasets relying solely on template-based augmentation.

**Future work: Mitigating prompt bias for richer flexibility.** We acknowledge that the current distribution may still reflect prompt-design bias inherent to GPT-4o's training data. To further enhance the diversity of flexible negation forms, future iterations of CoVAND could employ targeted prompt engineering strategies—such as explicitly requesting diverse negation structures (e.g., "describe the absence using lexical negation such as *lack* or *devoid of*")—or post-hoc augmentation techniques to rewrite regularized negations into their flexible counterparts. Such refinements could yield a more balanced distribution while preserving the semantic integrity established by our CoT and VQA pipeline.

### A.4 DATA ERROR CASE ANALYSIS

To quantify residual annotation errors in CoVAND, we perform a two–stage *human-in-the-loop* audit combining an independent multimodal language model with manual inspection.

**Stage 1: Automated cross–model audit.** We first randomly sample 1,000 image–caption pairs from the training split of CoVAND. Each sample consists of an image, a target bounding box, and a pair of captions describing the same region: a *negative* caption (hard negative, e.g., "a boy without a helmet") and a *positive* caption (true description, e.g., "a boy without a backpack"), together with the key attribute mentioned in the caption ("helmet", "backpack", *etc.*).

For each sample, we generate a visual prompt by overlaying a red rectangle on the target bounding box and feed the resulting image, the caption, and the attribute to an off-the-shelf multimodal LLM, Gemini-2.5 (Comanici et al., 2025), that is architecturally and training-wise independent from the model used in our data generation pipeline. The model is instructed to act as an objective judge and to return a structured JSON answer indicating whether the attribute is visually *present* in the red box:

```
{
  "is_attribute_present": boolean,
  "reasoning": "short explanation"
}
```

We then apply a deterministic decision rule to compare the dataset label with the model's prediction:

- For a negative caption (intended hard negative of the form "$X$ without $Y$"), the example is considered *valid* if the attribute $Y$ is in fact present in the region (is_attribute_present = true).
- For a positive caption (intended true caption of the form "$X$ without $Y$"), the example is considered *valid* if the attribute $Y$ is indeed absent (is_attribute_present = false).

For each caption, we log the full record (image path, bounding box, caption type, attribute, model verdict, and free-form reasoning) in a JSON file for subsequent human analysis.

Over 1,000 sampled pairs, the independent model disagrees with the CoVAND label in 78 cases (7.8%). These disagreements define the pool of potentially erroneous annotations.

**Stage 2: Manual verification of disagreements.** In the second stage, we load the logged results and focus on the disagreement set. A lightweight visualization tool displays, for each case, the original image with the red bounding box plus textual metadata (caption, attribute, model verdict, and reasoning). Annotators then categorize each disagreement as either:

- **Dataset error:** the CoVAND label is incorrect and the independent model's judgment is correct, or
- **Model error:** the CoVAND label is correct and the independent model misinterprets the visual evidence.

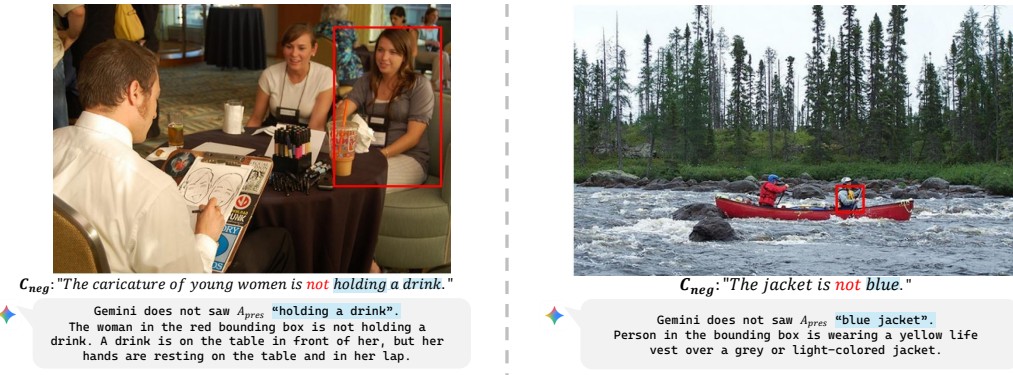

Figure S9: **Representative dataset errors for negative captions.** Each panel shows an image with the target region highlighted and the corresponding hard-negative caption ("$X$ without $Y$"). Many mislabels arise in visually subtle cases (e.g. barely visible or skin-colored attributes) where the "absent" attribute $Y$ is in fact present but difficult to perceive even for humans.

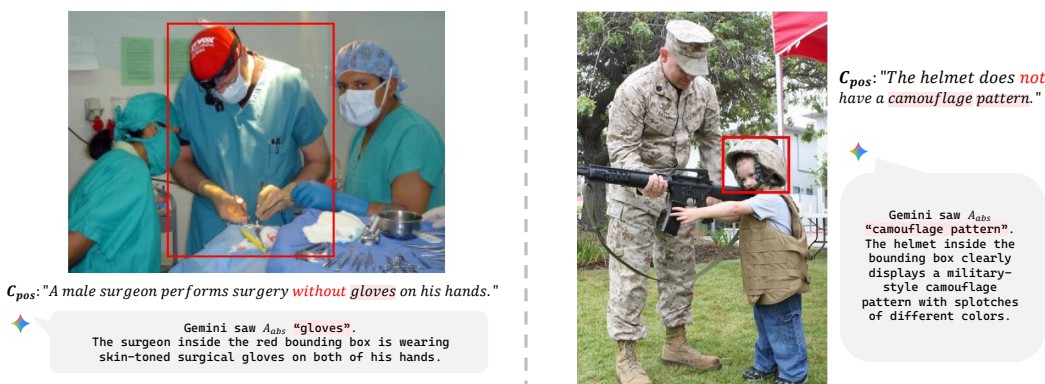

Figure S10: **Representative dataset errors for positive captions.** Examples where the caption intends to describe a true absence of an attribute, but small objects, occlusions, or cluttered scenes make the decision borderline. These cases illustrate that the residual annotation noise in COVAND is dominated by inherently ambiguous instances.

Among the 78 disagreements, 23 are judged as true dataset errors and 55 as model errors. This yields an estimated annotation error rate of 2.3% on the audited sample, corresponding to 97.7% factual accuracy for COVAND. Most errors occur in visually ambiguous situations, such as fine-grained appearance attributes or partially occluded objects, rather than systematic failures of the generation pipeline.

**Qualitative patterns.** Fig. S9 and Fig. S10 illustrate typical error cases discovered by this audit. For negative captions, errors often arise when the "absent" attribute is present only in a subtle or non-prototypical form (e.g., skin-toned medical gloves that are hard to distinguish from bare hands), or when the bounding box tightly crops out context that would disambiguate the attribute. For positive captions, errors typically involve borderline cases where small accessories, distant objects, or strong reflections make it difficult to determine whether the attribute is truly absent in the marked region.

Overall, this analysis indicates that the remaining noise in COVAND is both quantitatively small and qualitatively concentrated in genuinely ambiguous instances, rather than reflecting a systematic self-consistency bias of the underlying generation procedure.

```
You are provided with an image in which the target object "<TARGET_PHRASE>" is highlighted using a red contoured
bounding box.  You are a vision–language model with advanced chain-of-thought reasoning.  You must produce both
negative and positive captions referencing the same main subject, "<TARGET_PHRASE>".

Step 1) Summarize the highlighted bbox existing/missing contents (color, action, location, relationship, shape,
texture, etc.):

    [Existing Contents] Provide at least 3 short attribute or keyword items that describe SHOWN within the red
    bounding box.
    - All contents should be CLEARLY CHECKED in image.
    - Example: If the region corresponds to 'woman', you could include items like ['running at left lane', 'brown
            hair', 'blue shirt', 'jumping', 'holding a bat'].
    [Absent Contents] Provide at least 3 short attribute or keyword items that describe NOT in the red bounding box.
    - All contents should be CLEARLY MISSING in image, but somewhat relvant to the situation.
    - Example: If the region corresponds to 'A woman in a blue shirt rides a bicycle', you could include items like
            ['helmet', 'glasses', 'red hoodie'], if all items are not in the image.

Step 2) For selected content items from step 1, produce exactly ONE negative caption and ONE positive caption with
negation expressions (e.g. 'no', 'not', 'never', 'without', 'un-', ...). Each caption should be about the bounding
box's main subject ("<TARGET_PHRASE>" in the red bbox) as the focus.

    [Negative caption]: Caption that mismatched with the target region by combining negation expression and existing
    content item.
    (1) Must contain a negation expression with Existing Contents.
    (2) Keep it a single sentence or phrase, but it can be descriptive on target region.
    (3) Example: If existing contents are ['man', 'blue shirt', 'hat'] -> select 'hat'
                => 'A man without hat on his head.' ('hat' with 'without')
                If existing contents are ['plate', 'on the top', 'black', 'near the woman']
                => select 'near the woman' => 'A black plate is not located near the woman.'

    [Positive caption]: Caption that match with target region containing absent concepts with negation expressions.
    (1) Must contain a negation expression with Absent Contents.
    (2) Keep it a single sentence or phrase, which is actually present or relevant.
    (3) Example: If absent contents are ['helmet', 'glasses', 'red hoodie'] => select 'red hoodie',
                you could say 'A woman without a red hoodie rides a bicycle.'

Step 3) Provide verification for each caption:

- After each negative or positive caption, include a short 'verification' string that clarifies why it is truly
  negative or positive, focusing on the use of the negation.
- Negative check: (1) Does it contain a negation expression? (2) Does it contain the existing item from Step 1? (3)
  Does it mismatch with the bounding box contents?
- Positive check: (1) Does it contain a negation expression? (2) Does it contain the absent item from Step 1? (3)
  Is that negation absent from the bounding box, but thematically relevant?

IMPORTANT:
- Keep each caption to one sentence. Natural, fluent English with a bit of descriptive detail is encouraged.
- Your bbox_contents and subsequent captions should provide unique or distinguishing details specifically about the
  object in the target region, ensuring that they do not unintentionally refer to objects or attributes that lie
  outside of this indicated region.
- Return your final answer in a JSON structure with the following schema:

{
        "steps": [ { "explanation": "...", "output": "..." }, ... ],
        "bbox_contents": {      "existing": [ ... ],      "absent":   [ ... ]  },
        "pairs": [
                {
                "content_item": {
                        "existing": "<one existing item>",
                        "absent":   "<one absent item>"
                },
                "negative_caption": "...",
                "negative_verification": "...",
                "positive_caption": "...",
                "positive_verification": "..."
                }
        ]
}

You should reveal your chain-of-thought in steps[1,2,3], but keep it concise and do not mention about visual prompt
in the final output sentences. Please identify at least 3 existing/missing items (other than the main subject) in
that region, then select one for generating negative/positive caption pairs with verification. Use the JSON schema
described above.
```

Figure S11: **Prompt for Three-step CoT Negation Caption Generation.** Our prompt guides the model to systematically (1) extract present and absent attributes from visually highlighted regions, (2) generate complementary negative and positive captions with explicit negation markers, and (3) verify semantic alignment through logical validation.

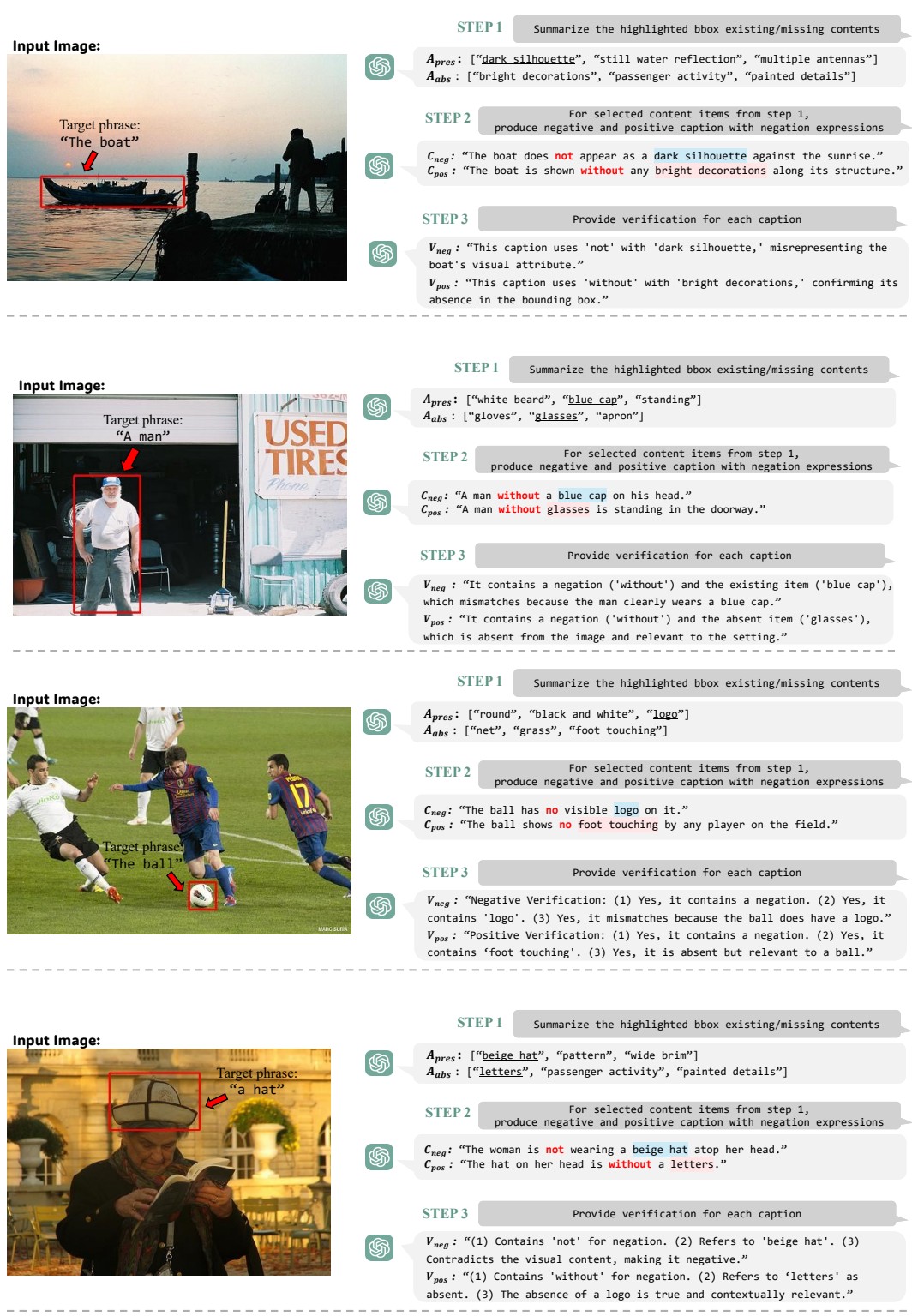

Figure S12: **Examples of CoVAND with 3-step CoT Caption Generation (1).** Example images and corresponding captions. Text with blue is present attribute($A_{pres}$) and pink is absent attribute($A_{abs}$). In detail, <negation word>+<$A_{pres}$> can generate negative caption($C_{neg}$) and <negation word>+<$A_{abs}$> can generate positive caption($C_{pos}$).

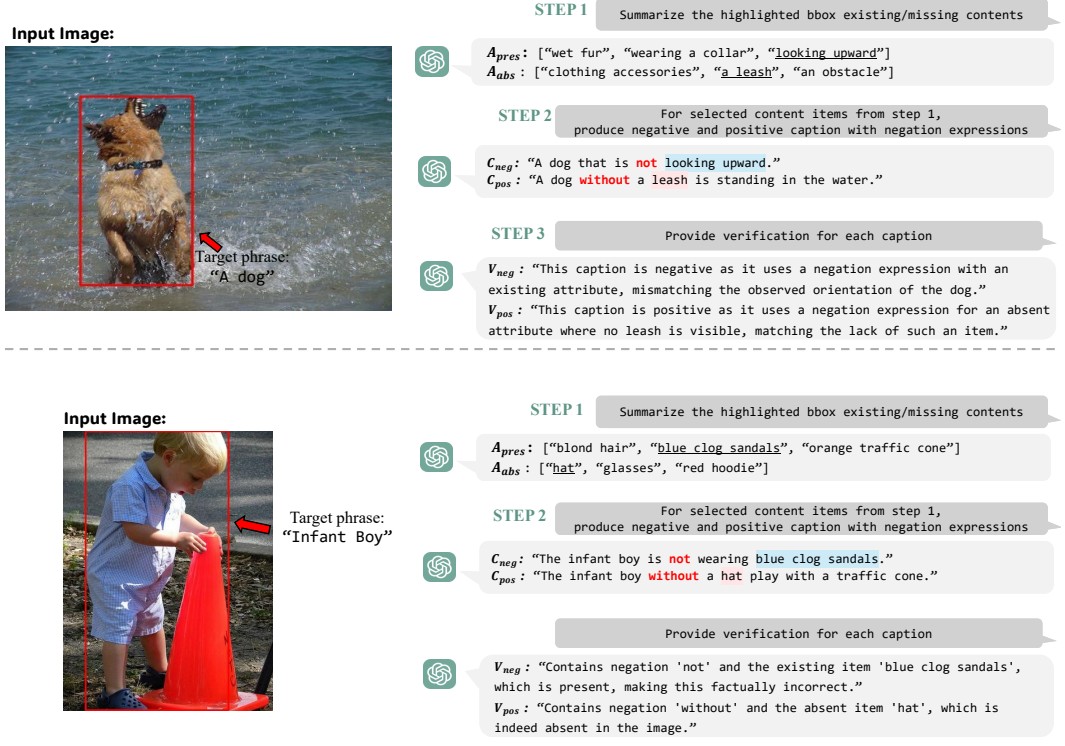

**STEP 1** Summarize the highlighted bbox existing/missing contents

$A_{pres}$: ["wet fur", "wearing a collar", "looking upward"]
$A_{abs}$: ["clothing accessories", "a leash", "an obstacle"]

**STEP 2** For selected content items from step 1, produce negative and positive caption with negation expressions

$C_{neg}$: "A dog that is **not** looking upward."
$C_{pos}$: "A dog **without** a leash is standing in the water."

**STEP 3** Provide verification for each caption

$V_{neg}$: "This caption is negative as it uses a negation expression with an existing attribute, mismatching the observed orientation of the dog."
$V_{pos}$: "This caption is positive as it uses a negation expression for an absent attribute where no leash is visible, matching the lack of such an item."

**Input Image:**

Target phrase: "A dog"

**STEP 1** Summarize the highlighted bbox existing/missing contents

$A_{pres}$: ["blond hair", "blue clog sandals", "orange traffic cone"]
$A_{abs}$: ["hat", "glasses", "red hoodie"]

**STEP 2** For selected content items from step 1, produce negative and positive caption with negation expressions

$C_{neg}$: "The infant boy is **not** wearing blue clog sandals."
$C_{pos}$: "The infant boy **without** a hat play with a traffic cone."

Provide verification for each caption

$V_{neg}$: "Contains negation 'not' and the existing item 'blue clog sandals', which is present, making this factually incorrect."
$V_{pos}$: "Contains negation 'without' and the absent item 'hat', which is indeed absent in the image."

**Input Image:**

Target phrase: "Infant Boy"

Figure S13: **Examples of CoVAND with 3-step CoT Caption Generation (2).** Example images and corresponding captions. Text with blue is present attribute($A_{pres}$) and pink is absent attribute($A_{abs}$). In detail, <negation word>+<$A_{pres}$> can generate negative caption($C_{neg}$) and <negation word>+<$A_{abs}$> can generate positive caption($C_{pos}$).

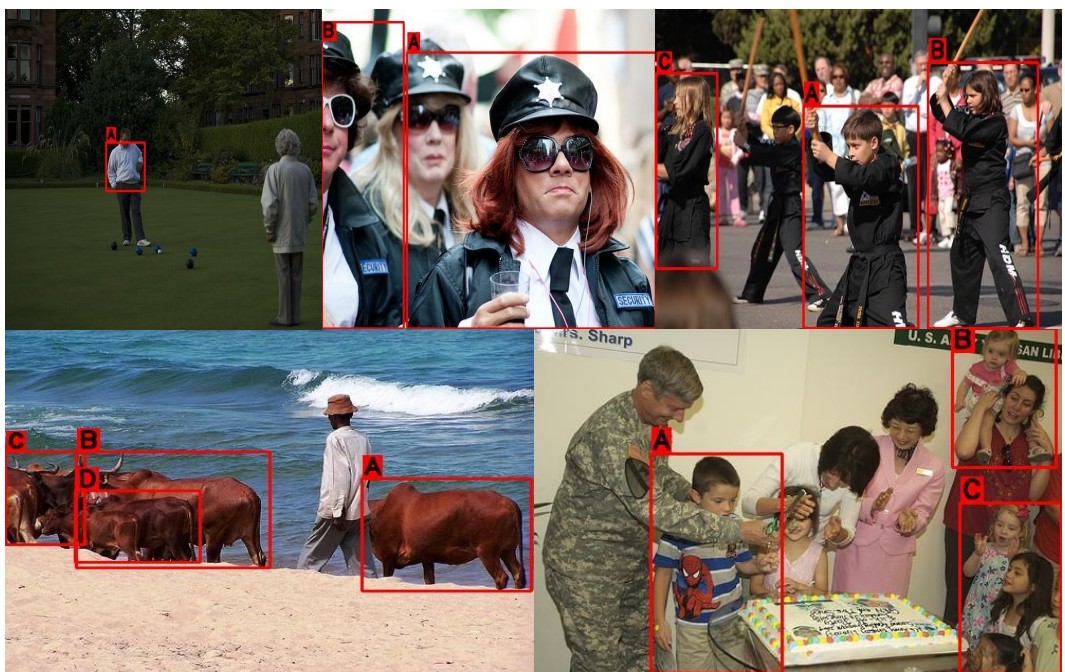

Figure S14: **Examples of Visual Prompt on VQA Alignments.** We apply alphabetical region labeling to all bounding boxes that share the target phrase type by assigning distinct markers (A, B, C, ...) to each instance with red bounding boxes.

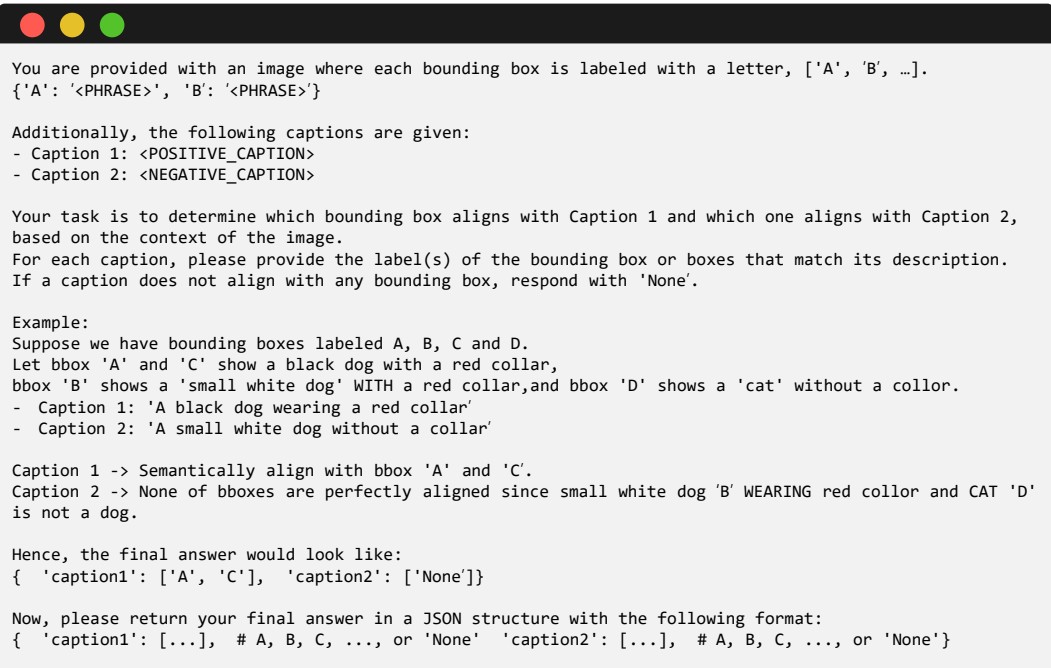

```
You are provided with an image where each bounding box is labeled with a letter, ['A', 'B', …].
{'A': '<PHRASE>', 'B': '<PHRASE>'}

Additionally, the following captions are given:
- Caption 1: <POSITIVE_CAPTION>
- Caption 2: <NEGATIVE_CAPTION>

Your task is to determine which bounding box aligns with Caption 1 and which one aligns with Caption 2,
based on the context of the image.
For each caption, please provide the label(s) of the bounding box or boxes that match its description.
If a caption does not align with any bounding box, respond with 'None'.

Example:
Suppose we have bounding boxes labeled A, B, C and D.
Let bbox 'A' and 'C' show a black dog with a red collar,
bbox 'B' shows a 'small white dog' WITH a red collar,and bbox 'D' shows a 'cat' without a collor.
- Caption 1: 'A black dog wearing a red collar'
- Caption 2: 'A small white dog without a collar'

Caption 1 -> Semantically align with bbox 'A' and 'C'.
Caption 2 -> None of bboxes are perfectly aligned since small white dog 'B' WEARING red collor and CAT 'D'
is not a dog.

Hence, the final answer would look like:
{ 'caption1': ['A', 'C'],  'caption2': ['None']}

Now, please return your final answer in a JSON structure with the following format:
{ 'caption1': [...],  # A, B, C, ..., or 'None'  'caption2': [...],  # A, B, C, ..., or 'None'}
```

Figure S15: **Prompt for VQA Alignment.** Our alignment process with (1) labeling all candidate bounding boxes with alphabetical markers, and (2) querying the VQA model to determine precise correspondences between generated captions and visually annotated regions.

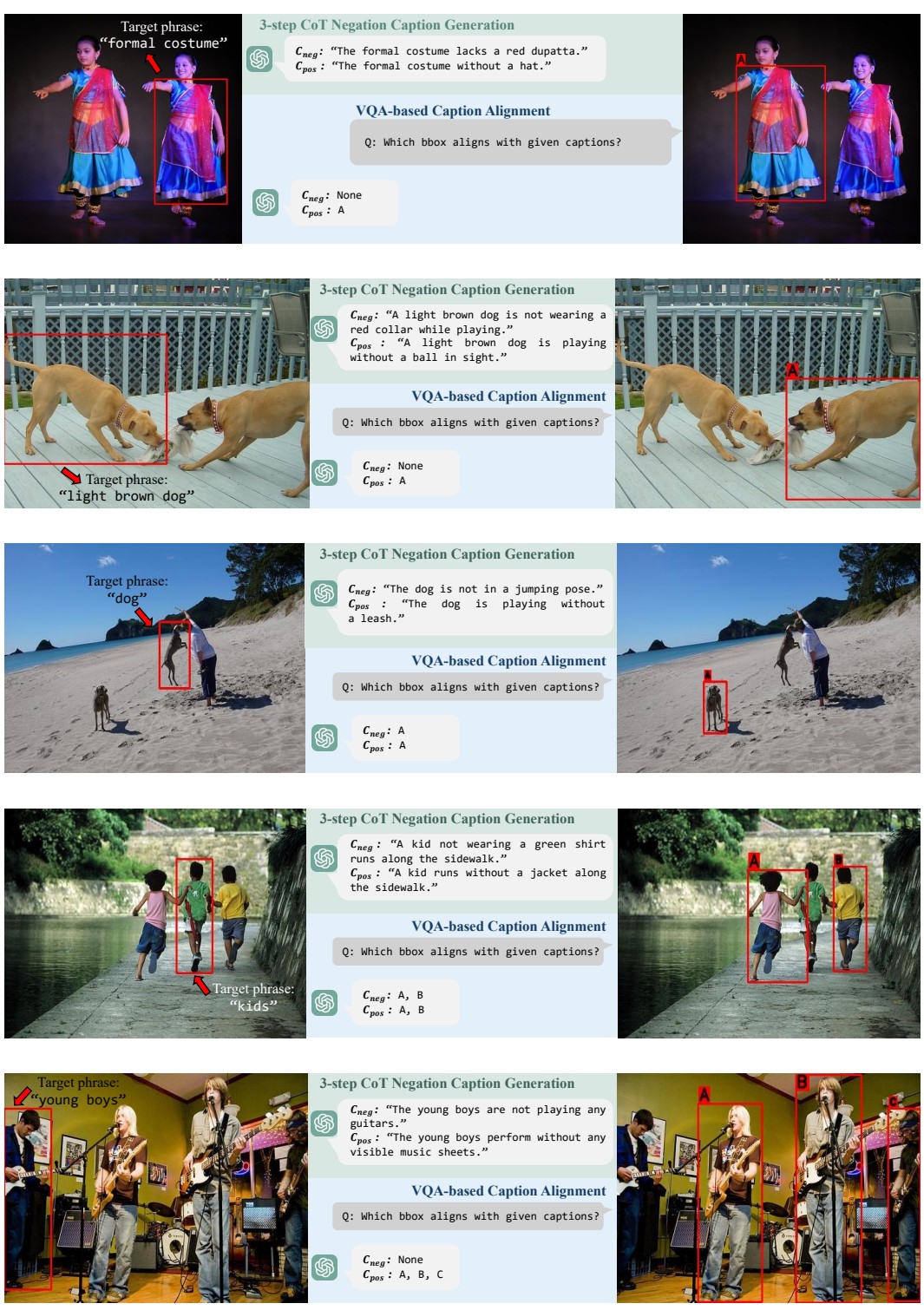

Figure S16: **Examples of CoVAND.** Example images for the 3-step CoT Negation Caption Generation and the VQA alignment are needed. The VQA alignment step is only executed when there are multiple instances with the same phrase type.

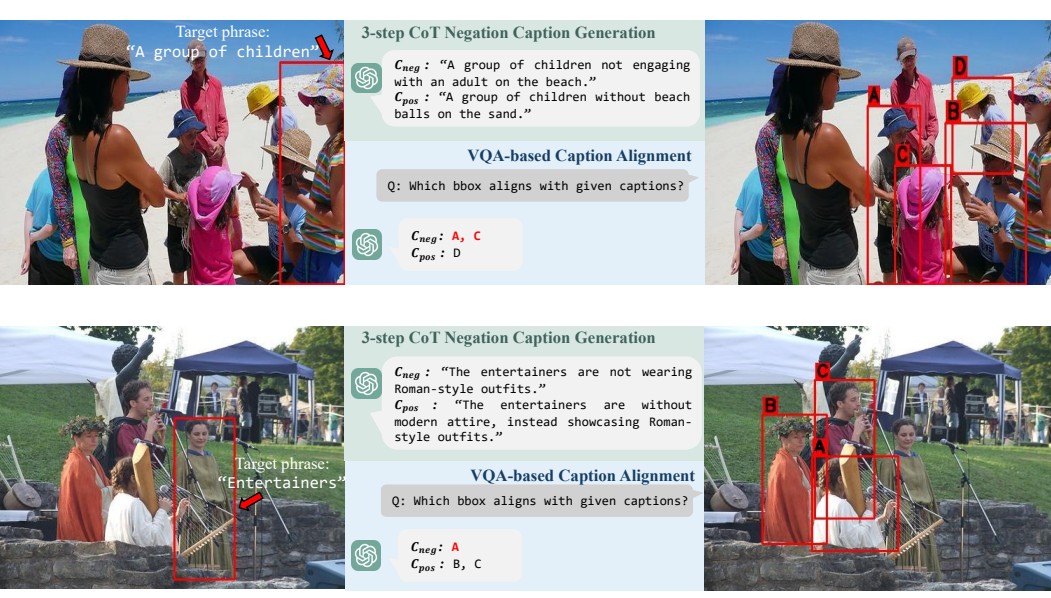

Figure S17: **Error on CoVAND.** VQA alignment occasionally fails when instances are densely clustered, making it difficult to determine which instance each visual prompt references.

## B    IMPLEMENTATION DETAILS

### B.1    GROUNDING DINO MODEL

Our implementation is built upon the Grounding DINO architecture (Liu et al., 2024b; Zuwei Long, 2023), which employs a dual-encoder-single-decoder design for vision-language understanding. For efficient fine-tuning towards negation understanding, we apply LoRA (Hu et al., 2022) to specific layers of the cross-modality decoder. The Grounding DINO consists of several key components:

- An image backbone (Swin Transformer (Liu et al., 2021)) for visual feature extraction
- A text backbone (BERT (Devlin et al., 2019)) for textual feature encoding
- A feature enhancer with self-attention and cross-attention mechanisms
- A language-guided query selection module that initializes query embeddings
- A cross-modality decoder that refines object detection based on both visual and text

We implement parameter-efficient fine-tuning by applying LoRA to `deep` layers (the final three cross-attention layers in the cross-modality decoder). This strategic placement allows us to modify how the model integrates negation cues from text with visual features while preserving pre-trained knowledge in earlier layers. Specifically, we insert LoRA only into the query ($Q$) and value ($V$) projections of the **text cross-attention**; the image deformable cross-attention and the self-attention blocks remain unchanged. The addition of ReLU activation between the down-projection and up-projection matrices, similar to (Chen et al., 2022), enhances the model's ability to capture non-linear relationships between negation cues and visual features. In Grounding DINO's cross-attention, the interactions operate as follows:

- **Image Cross-Attention**:
  - Query ($Q$): the updated cross-modality query from the preceding self-attention layer
  - Key ($K$) and Value ($V$): the image features processed through the feature enhancer
- **Text Cross-Attention**:
  - Query ($Q$): the output from the image cross-attention layer
  - Key ($K$) and Value ($V$): text features encoding language information

Figure S18 reveals critical insights into the optimal placement of LoRA modules (Boenisch et al., 2025) for negation understanding. The baseline model (Figure S18a) shows a strong bias toward Special tokens across all decoder blocks, with negation cues receiving minimal attention. When we apply LoRA to `shallow` blocks (Figure S18b), negation tokens initially receive higher attention weights in blocks 0-2, but this effect rapidly diminishes in the later blocks where attention to negation drops.

In contrast, when we apply LoRA to `deep` blocks (Figure S18c), the model maintains consistent attention to negation tokens through blocks. This pattern persists through the final detection heads, explaining the superior negation-aware detection performance. Some works (Gao et al., 2025; 2024; Seputis et al., 2024) further validate our approach by demonstrating that allocation of adaptation capacity to mid-to-late transformer layers yields optimal results for complex semantic tasks.

With the addition of NEGTOME (Figure S18d), attention to negation tokens increases consistently across all blocks, with particular amplification in the final blocks where detection decisions are made. This confirms that our token merging strategy effectively preserves negation signals throughout the entire network, even in early blocks that did not receive LoRA adaptation. The combined effect creates a consistent processing path for negation cues from text encoding through to final detection, explaining the significant performance improvements observed in the OVDEval and $D^3$ benchmarks.

Together, these adaptations enable our model to effectively capture the semantics of negation by enhancing the cross-modal integration of negation cues with their corresponding visual attributes, resulting in more accurate detection under negation scenarios.

Compared with the tiny model of Grounding DINO baseline, we need merely 0.005% trainable parameters to capture negation cues effectively, as in Table S7. To keep the tiny model within the

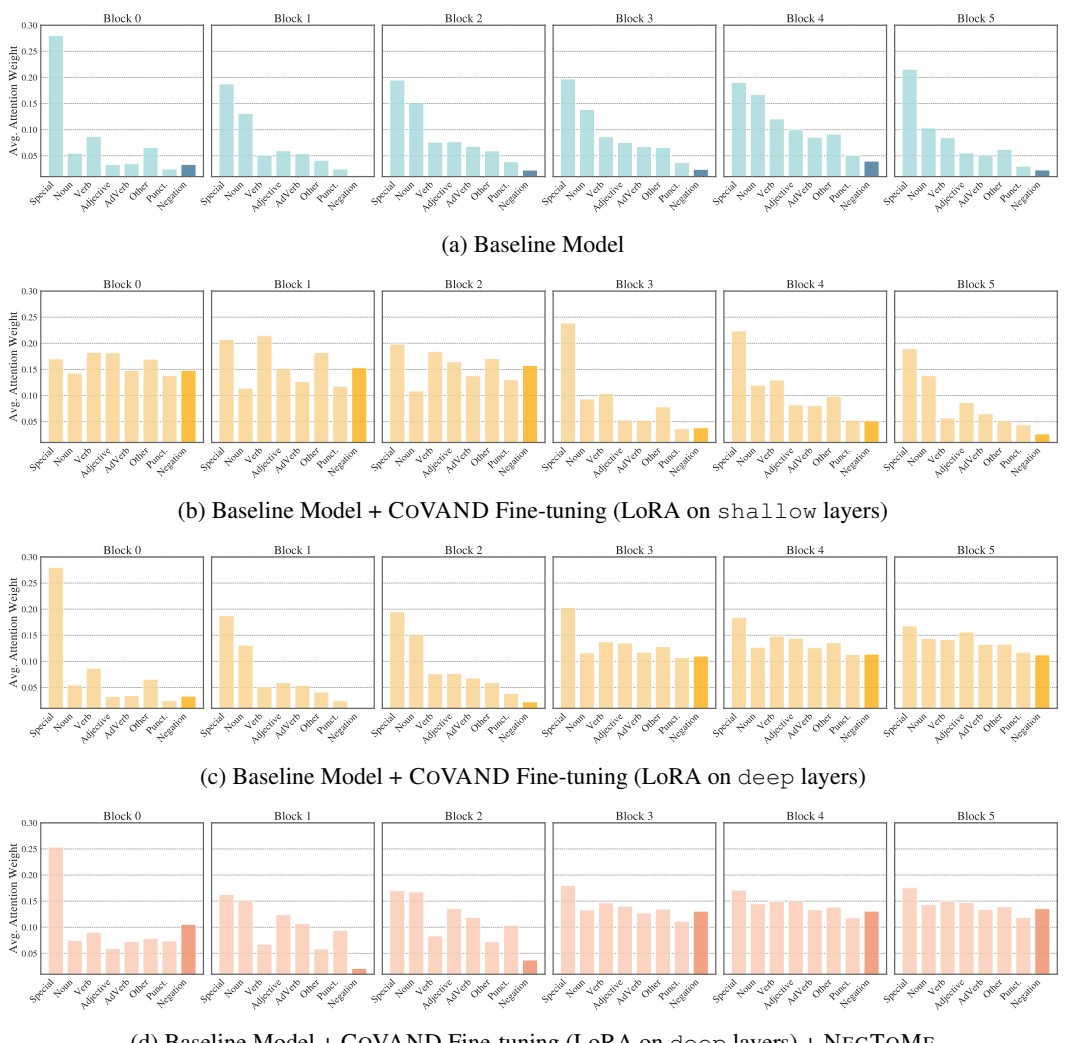

Figure S18: **Average Attention Weights by Decoder Blocks.** We only update the LoRA modules while freezing other layers for fine-tuning. Placement of LoRA, shallow means LoRA located on early decoder blocks (0-2) and deep means LoRA located on latter decoder blocks (3-5).

Table S6: **OVDEval-Negation Evaluation.** Performance on Grounding DINO tiny model.

| | AP | NMS-AP (Yao et al., 2024) | FPR |
|---|---|---|---|
| G-DINO-T (Liu et al., 2024b) | 48.5 | 22.8 | 54.0 |
| + Ours | 51.1 | 23.3 | 42.5 |
| | (+2.6) | (+0.5) | (−11.5) |

same 0.005% budget, we attach LoRA adapters to the 4 and 5 text cross-attention blocks of the decoder. Our lightweight adaptation yields a consistent performance gain: +2.6 AP and +0.5 NMS-AP, while slashing the False-Positive Rate (FPR) by **11.5%** as in Table S6. Although AP and NMS-AP improvements are moderate, they are achieved without sacrificing any metric; in fact, every reported score is on par with, or better than, the baseline, indicating that our negation-centric tuning does not degrade the detector's general ability.

Table S7: **Trainable Parameter Ratio.** The table compares the total model size with the number of LoRA-tuned parameters for each detector and backbone pair. During fine-tuning on COVAND, only the LoRA layers are trainable, with all other layers kept frozen with their pretrained weights.

| | Image Backbone | Total Param. | LoRA Param. | Ratio (%) |
|---|---|---|---|---|
| G-DINO-T (Liu et al., 2024b) | Swin-T (28.8M) | 173M | 8.2k | 0.005 |
| G-DINO-B (Liu et al., 2024b) | Swin-B (88M) | 233M | 12.3k | 0.005 |
| APE-Ti (Shen et al., 2024) | ViT-Ti (5.8M) | 771M | 129k | 0.017 |
| APE-L (Shen et al., 2024) | ViT-L (307M) | 1B | 129k | 0.012 |
| Qwen-2.5-VL-3B (Bai et al., 2025) | ViT-H (632M) | 3.8B | 77k | 0.002 |

## B.2 APE MODEL

Our implementation builds upon the APE framework (Shen et al., 2024), a universal visual perception model that unifies detection, segmentation, and grounding through instance-level region-sentence alignment. The architecture features several key innovations:

- A vision backbone (ViT-L (Dosovitskiy et al., 2020)) pretrained with EVA-CLIP (Sun et al., 2023) for visual feature extraction
- A text encoder (EVA02-CLIP (Fang et al., 2024)) processing both categorical vocabularies and free-form descriptions
- A gated cross-modality interaction module that fuses visual and text features
- A transformer decoder with deformable attention (Zhu et al., 2020) for joint reasoning

APE introduces a novel gated fusion mechanism that efficiently handles thousands of prompts per forward pass. Unlike previous approaches that directly fuse all text features (Li et al., 2022), APE implements conditional interaction paths:

$$\hat{V} = \begin{cases} V + \text{Attn}(V, P_{\text{voc}}) & \text{for vocabulary prompts} \\ \text{Attn}(V, P_{\text{sen}}) & \text{for sentence descriptions} \end{cases} \tag{4}$$

where $V$ denotes visual features and $P$ represents text embeddings. This gating strategy reduces FLOPs compared to GLIP-style fusion (Li et al., 2022). The model processes inputs at 1,024 pixel resolution using AdamW optimization (Loshchilov & Hutter, 2017) with learning rate 0.0005 and weight decay 0.05. We employ large-scale jittering augmentation (Ghiasi et al., 2021) with random scales from 0.1 to 2.0. We train APE-Ti models with four A6000 GPUs with a batch size of 4.

We apply LoRA exclusively to the encoder's cross-attention layers where visual and text features interact. This targeted adaptation modifies only 0.017% of APE's parameters as in Table S7. Despite APE-L's strong theoretical performance, its 1B parameters exceed the 48GB memory capacity of NVIDIA A6000 GPUs during training. We therefore focus on APE-Ti, which achieves 32.5 AP on $D^3$ while maintaining practical deployability.

## B.3 QWEN-2.5-VL MODEL

In addition to dedicated detectors, we test our method's generalizability on a powerful Multimodal Large Language Model (MLLM), Qwen-2.5-VL (Bai et al., 2025). Unlike dual-encoder architectures, Qwen-2.5-VL is an end-to-end model that directly processes interleaved image and text data. As detailed in its technical report, the architecture consists of three main components:

- A Vision Transformer (ViT-H) that is redesigned and trained from scratch to handle native resolution inputs. For efficiency, it incorporates windowed attention in most layers, with full self-attention only in specific blocks. The ViT architecture is also updated with RMSNorm and SwiGLU activations to align with modern LLM design principles.
- An MLP-based Vision-Language Merger that compresses spatially adjacent patch features before feeding them into the language model, enhancing computational efficiency.

- A Large Language Model decoder based on the Qwen2.5 architecture, which performs unified reasoning over the combined multimodal input and generates responses, including object coordinates for detection tasks.

For parameter-efficient fine-tuning, we again employ LoRA, strategically targeting the `deep` layers of the LLM decoder to enhance its negation reasoning without disturbing its foundational knowledge. Based on our experimental setup, LoRA adapters are specifically injected into the query (`q_proj`) and value (`v_proj`) projections of the self-attention modules within layers (15, 24, 30). This targeted placement is designed to modulate how the model integrates visual information with textual negation cues in its higher-level semantic reasoning stages. We configure the LoRA adapters with a rank $r$ of 4 and a dropout probability of 0.05.

Following the execution script, the Qwen-2.5-VL-3B model is fine-tuned for 1 epoch on our CoVAND dataset on H200 GPU. We use a learning rate of $5e-5$ with the AdamW optimizer (Loshchilov & Hutter, 2017) and a per-device batch size of 32, with 2 gradient accumulation steps, totaling an effective batch size of 64. The model is trained using bfloat16 mixed-precision. During this process, all original model parameters-including the ViT, MLP merger, and LLM backbone—are kept frozen; only the injected LoRA adapter weights are updated.

## C  EXTENDED RELATED WORK

### C.1  DATASET CONSTRUCTION WITH CHAIN-OF-THOUGHT REASONING

Chain-of-Thought (CoT) based data generation has emerged as a critical methodology for constructing high-quality datasets, particularly for tasks requiring multi-step reasoning. Early work in this domain relied heavily on manual annotation or rule-based generation, but recent advances have automated and scaled CoT-based dataset construction through Large Language Models (LLMs).

**Systematic CoT Pipeline Design**  The construction of CoT datasets requires careful consideration of quality and scale trade-offs. VideoEspresso (Han et al., 2025) demonstrates a systematic approach to automatic CoT dataset generation, employing a three-stage pipeline: (1) semantic-aware key information extraction using frame-level captioning (Kazakos et al., 2025), (2) multi-frame question-answer pair construction guided by carefully designed prompts, and (3) multimodal CoT annotation with spatial and temporal grounding (Kim et al., 2025).

This pipeline bridges the gap between manual annotation and fully automated generation by leveraging LLMs to produce reasoning chains (Lee et al., 2024; Tan et al., 2024) while maintaining consistency and factual accuracy through iterative quality filtering with an auxiliary LLM validator (Wang et al., 2025b; Chen et al., 2025e). The approach implements a redundancy removal mechanism using semantic similarity (Abbas et al., 2023) to filter out low-quality data, achieving 203,546 high-quality QA pairs with fine-grained CoT annotations. Furthermore, external validation tools (Findeis et al., 2025) are employed to ground generated annotations in visual evidence, ensuring both consistency across temporal boundaries and factual accuracy of intermediate reasoning steps.

**Information-Theoretic Framework for CoT Evaluation**  Understanding the quality of intermediate reasoning steps in CoT chains is essential for constructing reliable datasets. Recent works (Ton et al., 2024) propose an information-theoretic framework that formalizes CoT reasoning through the lens of information theory (Xiao et al., 2025), enabling the identification of failure modes without annotated CoT data. This theoretical foundation is grounded in process supervision paradigms (Lightman et al., 2023; Jia et al., 2025).

Their key contribution is the concept of information-gain at each reasoning step: a correct step should provide meaningful information toward predicting the final answer. By training a supervisor model to estimate conditional mutual information between intermediate steps and the final answer, they can quantify the contribution of each step without expensive human annotation. This approach outperforms outcome-based reward models (Lightman et al., 2023) and Math-Shepherd on detecting erroneous reasoning steps (Chen et al., 2025b).

The framework extends toward mechanistic interpretability through sparse autoencoders and activation patching (Chen et al., 2025c). These techniques reveal that CoT induces interpretable compu-

tational structures in larger models, with domain-specific causal patterns (Zhao et al., 2025). This mechanistic analysis validates that intermediate steps reflect genuine internal computation (FU et al., 2025).

The framework extends beyond simple accuracy metrics to provide step-wise interpretability (Hanna et al., 2025) and identifies unidentifiable sub-tasks—those which the model has not learned from training data (Muhamed et al., 2025). By combining information-theoretic analysis with causal structure verification, the framework creates a comprehensive evaluation pipeline grounded in both statistical and mechanistic principles.

**Synthesis for CoVAND Dataset Construction** Drawing from these insights, our CoVAND dataset construction pipeline incorporates: (1) systematic 3-step CoT prompts guided by information-theoretic principles to ensure each step contributes meaningful information, (2) multi-level quality verification using both LLM-based filtering and human review to ensure faithfulness, (3) validation through external VQA models to confirm region-level alignment accuracy, and (4) careful consideration of model scale effects by selecting sufficiently large models for CoT generation. This comprehensive approach ensures that CoVAND provides high-quality, instance-grounded negation data where the reasoning process genuinely reflects the model's internal understanding.

## C.2 VISUAL GROUNDING AND REGION-LEVEL ALIGNMENT IN VQA

Visual grounding, which is the task of localizing image regions corresponding to textual descriptions, plays a crucial role in connecting language-based reasoning to visual content. This capability is particularly important for our NEGTOME module, which grounds text tokens to image regions to implement negation-aware token merging.

**Visual Grounding Methods and Evaluation** Visual grounding has evolved from CNN-based two-stage pipelines to unified transformer-based end-to-end frameworks. TransVG (Deng et al., 2021) pioneered transformer-based visual grounding by performing intra- and inter-modality relation reasoning homogeneously. Recent advances further improved this by moving toward multi-task grounding architectures that jointly perform localization and segmentation (Dai et al., 2025) and leveraging coarse-to-fine consistency constraints. A comprehensive survey on visual grounding (Xiao et al., 2024) systematizes this evolution and identifies critical research directions, including grounding multimodal LLMs and generalized visual grounding.

A key insight is that visual grounding can be performed efficiently without fine-tuning. Contrastive Region Guidance (CRG) (Wan et al., 2024) introduces a training-free guidance method that enables open-source VLMs to respond to visual prompts by contrasting model outputs with and without region masking. CRG achieves up to 11.1% absolute accuracy improvements across diverse region-based tasks, demonstrating that pre-trained VLM representations already encode spatial information. This aligns with our design choice of using lightweight region guidance rather than expensive fine-tuning, and CRG complements our token-merging approach by providing model-driven region importance weighting.

**VQA-Based Visual Grounding** Visual Question Answering (VQA) provides a natural framework for generating region-level annotations. Some works (Shrestha et al., 2020a;b) show that visual grounding mechanisms are essential for explaining VQA predictions, bridging the gap between model confidence and spatial localization. More recent work (Reich & Schultz, 2024) uncovers the full potential of visual grounding methods in VQA, revealing that region-level understanding significantly improves multi-hop reasoning capabilities. Learning visual grounding from generative VLMs (Wang et al., 2025a) demonstrates that modern generative models can implicitly learn spatial correspondences, which our approach leverages by using model-generated region descriptions.

**Spatial vs. Semantic Matching** A fundamental distinction in visual grounding exists between spatial matching (geometric alignment) and semantic matching (concept alignment). Recent works (Chen et al., 2023; Daxberger et al., 2025) demonstrate that semantic alignment based on visual-linguistic correspondence often outperforms purely spatial approaches. Our VQA-based validation mechanism employs semantic matching by querying the model about object presence and

roles in specific regions, effectively combining spatial information (bounding boxes) with semantic understanding (question-answer consistency).

**Grounding for Instance-Level Negation**    For instance-grounded negation understanding, precise visual grounding is critical. The COVAND dataset construction process uses region-level VQA to ensure that when we assert "not X," the region boundaries are accurately localized and the absence claim is semantically verified. This region-centric approach differs from image-level negation understanding, which cannot distinguish between objects appearing in different spatial contexts.

## C.3    PARAMETER-EFFICIENT FINE-TUNING FOR VISION-LANGUAGE MODELS

Large vision-language models contain billions of parameters, making full fine-tuning computationally prohibitive. Parameter-efficient fine-tuning (PEFT) techniques enable adaptation with minimal additional parameters, a critical requirement for our NEGTOME adapter design.

**Low-Rank Adaptation (LoRA) and Variants**    LoRA (Hu et al., 2022) introduced a breakthrough approach decomposing weight updates into low-rank matrices $W = W_0 + AB^\top$, where $A \in \mathbb{R}^{d \times r}$ and $B \in \mathbb{R}^{r \times d}$ with rank $r \ll d$. This reduces trainable parameters from $O(d^2)$ to $O(dr)$. For CLIP-like models, LoRA has proven highly effective in few-shot scenarios. Low-rank few-shot adaptation of vision-language models (Zanella & Ben Ayed, 2024) empirically demonstrates that CLIP-LoRA achieves substantial improvements over prompt-learning and adapter-based approaches across 11 datasets while maintaining consistent hyperparameters across all tasks, effectively democratizing few-shot VLM adaptation without task-specific tuning.

A comprehensive PEFT survey (Han et al., 2024) provides systematic evaluation of PEFT algorithms and their computational overhead, revealing that LoRA remains competitive against more complex PEFT variants (prefix tuning, adapters, prompt tuning) for vision-language adaptation, particularly when strategically applied to specific layer types. However, the low-rank constraint limits expressivity on complex tasks. DoRA (Liu et al., 2024a), a weight-decomposed variant, decouples weight matrices into independent magnitude and direction components, emulating full fine-tuning dynamics more faithfully. DoRA consistently outperforms LoRA on vision-language tasks including visual instruction tuning (Liu et al., 2024a).

**Strategic Layer-Wise Adaptation**    Not all layers benefit equally from fine-tuning. Analysis of vision-language architectures reveals that cross-modal attention layers are critical for semantic alignment. Full-rank parameter-efficient fine-tuning (Albert et al., 2025) proposes RandLoRA, which performs full-rank updates via learned linear combinations of low-rank random matrices. RandLoRA significantly reduces the performance gap between LoRA and standard fine-tuning, particularly on vision-language tasks, demonstrating that when higher ranks are required, full-rank updates substantially outperform low-rank approximations.

Layer-wise learning strategies further refine adaptation. Layer-wise auto-weighting (Park et al., 2024a) employs Fisher Information Matrix (FIM) to autonomously identify which layers require preservation or concentrated adaptation, enabling more efficient non-stationary test-time adaptation. For our approach, we strategically apply LoRA to deep cross-attention layers where negation-specific reasoning is encoded, achieving a balance between expressiveness and efficiency. Additionally, data-efficient instruction tuning (Naharas et al., 2025) proves that examples with similar cross-modal attention matrices have similar gradients, informing our layer selection strategy: layers exhibiting high cross-modal attention variance are most critical for negation grounding.

**Fine-Grained Semantic Alignment**    Task-specific semantic alignment requires minimal parameters but careful design. A parameter-efficient prompt learning method (Guo et al., 2025) achieves fine-grained semantic alignment using only a few additional parameters. Their key insight is that semantic-level objectives require less parameter capacity than low-level pixel matching, particularly when combined with well-designed prompt templates and explicit semantic constraints.

For semantic gating in transformers, Value-State Gated Attention (Bu et al., 2025) introduces learnable, data-dependent gates computed from value vectors to modulate token contributions based on semantic importance. This mechanism directly informs NEGTOME's token importance weighting: by learning task-specific gates that suppress irrelevant tokens and amplify negation-bearing tokens,

we achieve semantic specialization with minimal overhead. Dynamic Mask Attention (Zhang et al., 2025) extends this by using soft-gating masks and content-aware mask generation based on value representations, enabling fine-grained token importance capture without binary keep-or-drop decisions.

**Negation-Specific Adapter Design** While most PEFT work addresses general-purpose adaptation, negation understanding requires specialized reasoning. Our NEGTOME module combines LoRA's parameter efficiency with negation-aware token merging, creating a compact yet semantically rich adaptation.

The negation-boost mechanism in token merging acts as a learned semantic gating function, allocating more computational resources to tokens representing negated concepts. This semantic specialization follows principles from hierarchical inductive transfer for continual learning (Feng et al., 2022), which demonstrates that separating base adapters (capturing general knowledge) from task-specific adapters (capturing task-specialized reasoning) prevents interference and improves performance on specialized tasks. By analogy, NEGTOME's selective token enhancement prevents general cross-modal attention from suppressing negation signals.

**Synthesis: Efficient Negation-Aware Adaptation** Our PEFT strategy for NEGTOME combines: (1) *LoRA efficiency*: strategic application to cross-modal attention layers, (2) *full-rank expressivity*: capturing complex negation-direction mappings, (3) *semantic gating*: value-state aware token importance weighting, and (4) *hierarchical structure*: base cross-modal adaptation and task-specific negation enhancement. This integrated approach ensures parameter efficiency while providing sufficient expressivity for negation-aware semantic reasoning, critical for learning instance-grounded negation in vision-language tasks.

## C.4 COMPOSITIONAL REASONING IN VISION-LANGUAGE TASKS

Compositional reasoning—understanding how simple concepts combine to form complex meanings—is fundamental to negation understanding. Negation is inherently compositional: "not red" requires first identifying "red" then applying the negation operator.

**Compositional Visual Reasoning Fundamentals** A comprehensive survey on compositional visual reasoning (Ke et al., 2025), establishing core definitions and theoretical foundations. The survey identifies key requirements for compositionality: (1) *primality*—identifying atomic concepts, (2) *compositionality*—systematically combining concepts, and (3) *systematicity*—applying compositional rules consistently across novel scenarios. The survey emphasizes that compositional reasoning provides advantages in cognitive alignment, semantic fidelity, robustness, interpretability, and data efficiency. Notably, it highlights that vision-language models struggle with compositionality due to training data bias toward positive instances and the parallel feature processing architecture, a challenge directly addressed by COVAND.

Recent analysis reveals that VLMs exhibit the "binding problem" (Campbell et al., 2024)—fundamental failures in reliably associating perceptual features with correct visual referents, particularly in multi-object scenarios. This limitation mirrors cognitive science findings on rapid feedforward processing in human brains. Addressing this binding problem through structured spatial reasoning (e.g., horizontal lines and sequential scanning prompts) yields substantial improvements across visual reasoning tasks (Izadi et al., 2025), demonstrating that explicit compositional structure enhances multi-object understanding.

**Compositional Generalization in Vision-Language Models** A critical gap exists between training and test distributions: models trained on one compositional split often fail on others, revealing weak compositional generalization. An empirical study on compositional generalization in VLMs (Li et al., 2024a) shows that current models rely primarily on linguistic priors rather than visual information, causing benchmarks to favor pure language models. To overcome this limitation, researchers propose evaluation frameworks without linguistic priors, forcing models to ground compositional reasoning in actual visual content.

Systematic compositional probing studies (Li et al., 2025c) evaluate whether RL-trained VLMs inherit compositional capabilities from LLMs. Key findings reveal: (1) RL-trained models outperform

SFT-only models on compositional generalization, (2) VLMs struggle to generalize compositionally under cross-modal and cross-task scenarios despite strong individual-task performance, and (3) enforcing models to explicitly ground visual content before reasoning (e.g., caption-before-thinking with visual grounding) yields notable gains. These findings directly motivate our explicit chain-of-thought (CoT) dataset construction: by forcing models to describe what negation means in specific regions before making predictions, we encourage true compositional understanding rather than shortcut learning.

Out-of-distribution generalization in LLMs is fundamentally tied to compositional structures internally achieved through aligned principal subspaces in self-attention layers (Song et al., 2025). This analysis suggests that models composing two self-attention layers can learn rules and generalize to novel tasks—a principle we leverage by designing NEGTOME to operate at intermediate layers where compositional negation reasoning naturally emerges.

**From Objects to Events: Temporal Compositional Reasoning**  Event-level reasoning extends compositional understanding beyond static attributes to temporal dynamics. Compositional event reasoning requires understanding object interactions and temporal sequences. Recent work on diffusion-driven video scene graph generation (Chen et al., 2025a) demonstrates that detecting complex temporal relations requires compositional reasoning about object trajectories and inter-frame dependencies, achieved through iterative refinement of spatial-temporal embeddings. Video referring object segmentation (Xu et al., 2025) decomposes referring expressions into structured event graphs with objects, relations, and temporal constraints, revealing that video-level negation requires reasoning about event-level compositional patterns.

Our 3-step CoT generation aligns with this event decomposition: each step performs a primitive operation (object detection → region analysis → negation verification), mirroring the hierarchical event reasoning paradigm. By explicitly grounding each reasoning step in concrete regions, CO-VAND teaches models to compose negation patterns systematically rather than relying on spurious correlations.

**Attribute Binding and Multi-Object Reasoning**  A central compositional challenge is correctly binding attributes to objects (the binding problem). Scene graph generation literature (Chang et al., 2021; Yang et al., 2024), identifies systematic failures in relational understanding. Video scene graph generation with unbiased training (Li et al., 2025d) addresses these failures through visual-semantic dual supervision, explicitly enforcing consistency between visual features and semantic relations. For negation, this principle translates to enforcing consistency between negated object descriptions and their spatial absence in images.

**Multimodal Compositional Generalization**  Retrieving semantically equivalent primitives across modalities enhances compositional generalization. Multi-sourced compositional generalization in VQA (Li et al., 2025b) proposes retrieval-augmented training where equivalent primitives from different modalities are aggregated to refine representations, improving generalization to novel compositions. For negation, this principle suggests that enforcing consistency between linguistic negation descriptions and visual absence patterns (via VQA-based validation) teaches models to reason about negation as a unified cross-modal compositional operator.

**Systematic Generalization in Visual Reasoning**  A benchmark for systematic generalization in visual world models (Kim et al., 2023b) evaluates whether models perform visual imagination and measure compositional generalization systematically. The benchmark reveals that even state-of-the-art models fail on out-of-distribution compositional scenarios. Our approach mirrors this systematic evaluation: by constructing CoVAND with systematic variations across objects, attributes, and negation types, we provide training data that encourages systematic rather than accidental generalization.

**Synthesis: Compositional Negation as Structured Visual Reasoning**  Our approach positions negation as a compositional operator within a broader framework of structured visual reasoning. By combining: (1) *explicit event decomposition* through 3-step CoT reasoning, (2) *visual-semantic binding enforcement* via VQA validation, (3) *spurious correlation mitigation* through systematic data variation, and (4) *out-of-distribution generalization* via consistent cross-modal mappings, CO-

VAND enables models to learn compositional negation patterns grounded in both visual evidence and semantic structure. This compositional foundation, combined with NEGTOME's semantic-aware token merging that respects the hierarchical structure of compositional reasoning, creates a framework where negation understanding emerges from systematic compositional principles rather than shallow surface-level negation cues.

## C.5 BIAS MITIGATION IN VISION-LANGUAGE MODELS

Affirmative bias—the tendency to misinterpret negations as affirmations—represents a systematic bias in VLM training data and architecture. Understanding and mitigating this bias is central to our work.

**Understanding Affirmative Bias in VLMs**  The fundamental source of affirmative bias lies in training data distribution. Most visual-linguistic corpora contain predominantly positive image-text pairs: "dog in park," "person running," etc. Negative examples are rare, creating class imbalance that models exploit through shortcut learning. DeAR (Seth et al., 2023) demonstrates that VLM bias can be partially mitigated by learning additive residual corrections to visual representations without retraining. However, their approach targets shallow social biases; negation bias requires deeper structural understanding.

Recent work reveals that VLMs exhibit social biases including gender and racial stereotypes in their generative responses (Lan et al., 2025). More critically, fair-response reliability differs from accuracy: models may achieve high accuracy while exhibiting significantly lower fairness scores. This decoupling between performance and fairness motivates our multi-level debiasing strategy, where data-level, method-level, and evaluation-level components collectively address negation bias.

**Architectural Sources of Affirmative Bias**  Bias is not merely a data problem but deeply embedded in architectural choices. Bias analysis in transformer attention heads (Yang et al., 2025) reveals that specific attention heads are responsible for encoding stereotypical biases; by identifying and masking high-bias heads, models achieve significant bias reduction without affecting language understanding. This attention-head level analysis directly informs NEGTOME's design: our selective token merging operates at the semantic level, effectively masking attention patterns that suppress negation signals.

**Shortcut Learning and Spurious Correlation**  Models exploit spurious correlations—features coincidentally correlated with labels in training data—to achieve high training accuracy while generalizing poorly to test data. Shortcut learning in LLMs (Du et al., 2023) identifies that models learn to associate demographic attributes with spurious features rather than learning genuine causal relationships. For negation, this manifests as: models may learn that "absence of red" correlates with "dark backgrounds" rather than learning true negation semantics.

ShortcutProbe (Zheng et al., 2025) introduces a post-hoc framework for identifying and mitigating spurious biases via prediction shortcuts in the model's latent space, without requiring group labels. The key insight is that spurious correlations create non-generalizable prediction shortcuts that fail under distribution shift. Our COVAND addresses this by systematically varying objects, attributes, and spatial configurations, forcing models to learn generalizable negation patterns rather than spurious shortcuts. Furthermore, generative classifiers naturally avoid shortcut learning (Li et al., 2025a) by modeling all features rather than mainly spurious ones, suggesting an alternative avenue for robust negation understanding.

**Synthetic Data Augmentation and Its Pitfalls**  While data augmentation seems a natural solution, naive approaches can inherit and amplify existing biases. Decoupling augmentation bias in prompt learning for VLMs (Gerych et al., 2024) demonstrates that random negation (e.g., "not red" generated from "red") can create inconsistent training signals due to augmentation distribution mismatch. COVAND addresses this by employing LLMs to generate semantically coherent negation examples with multi-step chain-of-thought reasoning verification, ensuring both linguistic consistency and visual grounding.

Handling imbalanced pseudolabels in VLMs (Pang et al., 2025) reveals that models exhibit class-preference biases when generating pseudolabels for unlabeled data. Their proposed solution com-

bines concept alignment with confusion-aware calibrated margins. Our CoVAND-derived dataset similarly enforces consistency through VQA validation and avoids concept confusion via systematic negation type variation.

**Bias Inheritance in LLM-Based Data Augmentation**     LLM-based augmentation introduces new challenges: models inherit biases from their training distribution even when generating synthetic data (Li et al., 2025a). When an LLM generates "not sitting," it may unconsciously preserve biases about typical sitting scenarios and demographic associations. Our solution combines LLM generation with human review and VQA validation, creating a hybrid pipeline resistant to bias inheritance. This multi-stage validation mirrors fairness-aware domain adaptation for VLMs (Pang et al., 2025), which uses attribute-aware strategies to dynamically adapt to diverse demographics for equitable outcomes.

**Negation-Aware Test-Time Adaptation**     Beyond training-time mitigation, test-time debiasing offers complementary benefits. BEND-VLM (Gerych et al., 2024) proposes nonlinear, fine-tuning-free debiasing that tailors debiasing to each unique input without prior knowledge of the test set. While their approach targets social biases, the principle of input-specific bias correction directly applies to negation: negation scopes and interpretations vary contextually, so adaptive, query-specific debiasing enhances robustness.

Debiasing VLMs using backdoor learning (Pang et al., 2025) proposes learning backdoor patterns that trigger debiased representations, enabling inference-time debiasing without model modification. This technique complements our approach: while NEGTOME merges tokens based on learned negation patterns, backdoor triggers could provide additional control signals for negation-aware inference.

**Synthesis: Multi-Level Affirmative Bias Mitigation**     Our comprehensive approach targets affirmative bias across multiple levels:

1. **Data-level**: Systematic CoVAND dataset construction with multi-modal chain-of-thought reasoning and VQA validation, explicitly addressing spurious correlations and shortcut learning through systematic data variation (Zheng et al., 2025; Gerych et al., 2024).

2. **Architecture-level**: NEGTOME's negation-aware token merging acts as a selective attention-head debiaser, amplifying negation-bearing tokens while suppressing spurious features. This semantic gating follows attention-head masking principles, but operates at the token level with learned importance weighting.

3. **Method-level**: Strategic LoRA application to cross-modal attention layers (Sections C.3) provides fine-grained control over negation reasoning without full model retraining, mitigating catastrophic forgetting associated with fine-tuning-based debiasing.

4. **Evaluation-level**: Instance-grounded metrics that measure negation understanding separately from general object detection, preventing bias amplification in evaluation and enabling fair assessment of minority negation patterns.

By combining these complementary strategies, we provide stronger affirmative bias mitigation than any single technique, addressing shortcut learning, spurious correlations, and architectural bias simultaneously. Furthermore, this multi-level approach generalizes beyond negation to other linguistic phenomena requiring fine-grained compositional understanding.

## D     ADDITIONAL ABLATIONS: NEGATION- AND NOUN-ONLY BOOSTING

This section provides implementation details and empirical analysis of negation- and noun-only boosting variants that test whether simply increasing the emphasis on certain tokens (without structural merging) is sufficient to improve negation understanding in VLM-based detectors.

**Implementation details.**     Given a caption string $x$, we first run a spaCy parser to obtain a token sequence $d = (w_1, \ldots, w_{L_{\text{spa}}})$ and phrase-level groupings $P_1, \ldots, P_M$ as described in the main paper. We then tokenize the same string with the model's text tokenizer (BERT/CLIP) to obtain

subword tokens $(u_1, \ldots, u_{L_{\text{hf}}})$ and use the alignment procedure to map each spaCy token to its set of subword indices.

For each phrase $P_i$ we obtain an index set $\mathcal{I}_i \subseteq \{1, \ldots, L_{\text{hf}}\}$ of subword positions and compute a merged embedding $\bar{t}_i$ using the normalized weighted-average *merging operator*:

$$\bar{t}_i = \frac{\sum_{j \in \mathcal{I}_i} \gamma_j t_j}{\sum_{j \in \mathcal{I}_i} \gamma_j}, \tag{5}$$

where $t_j$ is the original text embedding at subword index $j$ and $\gamma_j > 0$ is an importance weight that depends on the phrase type.

In all variants, the phrase replacement is applied once per phrase: the original subword embeddings $\{t_j\}_{j \in \mathcal{I}_i}$ are removed and replaced by a single merged embedding $\bar{t}_i$ at that position. The different ablations differ only in how we choose the weights $\gamma_j$ and which tokens are boosted by a multiplicative factor $\beta > 1$:

- **Noun-only Boost.** We identify head nouns inside each phrase $P_i$ using spaCy dependency tags (see `collect_noun_spacy_indices` and `spacy_indices_to_hf_indices` in the code), map them to subword indices, and form a set $\mathcal{B}_{\text{noun}}$ of boosted positions. For all phrases we use the merging operator in Eq. equation 5, with

$$\gamma_j = \begin{cases} \beta & \text{if } j \in \mathcal{B}_{\text{noun}}, \\ 1 & \text{otherwise.} \end{cases}$$

  This variant is agnostic to negation cues and only amplifies nouns/heads.

- **Neg-only Boost.** We first collect spaCy indices of negation cues ("not", "no", "without", "never", "n't", etc.) using `collect_negation_cue_spacy_indices`, then map them to subword indices to obtain a set $\mathcal{B}_{\text{neg}}$. We again use Eq. equation 5 with

$$\gamma_j = \begin{cases} \beta & \text{if } j \in \mathcal{B}_{\text{neg}}, \\ 1 & \text{otherwise.} \end{cases}$$

  Importantly, we *do not* structurally bind the negation token to the attribute; the cue remains an isolated token (or part of a short phrase) that competes with other tokens in cross-attention.

- **Merge Head Boost.** For this variant we apply phrase merging to all phrases as in Eq. equation 5, but we treat negation cues and non-negation phrases uniformly. Inside each phrase, only the head noun is boosted:

$$\gamma_j = \begin{cases} \beta & \text{if } j \in \mathcal{B}_{\text{head}}, \\ 1 & \text{otherwise,} \end{cases}$$

  where $\mathcal{B}_{\text{head}}$ contains subword indices of phrase heads. This tests whether phrase-level structural unification alone (without explicit negation-aware weighting) is sufficient.

- **Attn Bias.** Instead of modifying the text embeddings, this variant inserts a per-token bias into the cross-attention logits. For each negation cue index $j \in \mathcal{B}_{\text{neg}}$ we add a fixed bias $\delta > 0$ to the attention score for that token, leaving the tokenization structure unchanged. Concretely, if $A \in \mathbb{R}^{Q \times L_{\text{hf}}}$ is the query–key dot product matrix in a decoder block, we replace it by $A' = A + B_{\text{neg}}$, where $B_{\text{neg}}$ is a matrix whose $j$-th column is shifted by $\delta$ for negation indices and zero elsewhere. This variant corresponds to the `Attn Bias` row in Table S8.

- **Ours (NEGTOME).** For non-negated phrases $P_i$ we use Eq. equation 5 with uniform weights $\gamma_j = 1$, i.e., a simple average. For *negated phrases* $P_{\text{neg}}$ that contain a negation cue and a modified attribute (e.g., "not lying", "without hat") we set

$$\gamma_j = \begin{cases} \beta & \text{if } j \text{ corresponds to the negation cue,} \\ 1 & \text{for the other tokens in } P_{\text{neg}}. \end{cases}$$

Thus, Eq. equation 5 simultaneously performs *phrase merging* and *negation-aware boost*: all subwords in the negated phrase are collapsed into a single merged token $\bar{t}_{\text{neg}}$, whose representation is dominated by the cue contribution but remains conditioned on the attribute it negates.

| Method | AP ↑ | FPR ↓ |
|---|---|---|
| Noun-only Boost | 50.5 | 66.1 |
| Neg-only Boost | 49.7 | 62.0 |
| Merge Head Boost | 56.2 | 62.4 |
| Attn Bias | 46.2 | 59.6 |
| **Ours (NEGTOME)** | **57.3** | **47.7** |

Table S8: **Negation- vs. noun-only boosting ablations on OVDEval-Negation.** All models are fine-tuned on COVAND-L for 1 epoch with identical LoRA configuration. NEGTOME is the only variant that simultaneously improves AP and substantially reduces FPR.

All variants share the same training and evaluation protocol: we fine-tune Grounding DINO-B with LoRA on the COVAND-L split for one epoch and evaluate on OVDEval-Negation, using COCO-style AP and the False Positive Rate (FPR) computed after NMS.

**Quantitative results.**    Table S8 summarizes the results.

We observe three key trends:

1. **Negation-only boosting is insufficient.** The *Neg-only Boost* variant (boosting only negation cue embeddings without changing structure) yields AP = 49.7 and FPR = 62.0. Similarly, the *Attn Bias* variant, which adds attention bias toward negation tokens while preserving the original tokenization, collapses to the lowest AP (46.2) despite a small FPR reduction (59.6). This confirms that simply increasing the magnitude or attention weight of negation tokens does not translate into reliable negation understanding; it can even disrupt compositional semantics.

2. **Phrase merging alone is necessary but not sufficient.** The *Merge Head Boost* variant leverages the same phrase-merging operator as NEGTOME but boosts only the head noun in each phrase and does not treat negation cues specially. This already improves AP to 56.2, suggesting that structurally unifying multi-token phrases into single semantic units helps detection. However, FPR remains high (62.4), indicating that the model still tends to fire on both "with" and "without" queries (i.e., it detects the object but fails to respect the polarity).

3. **NEGTOME (merging + negation-aware boost) is required.** Only the full NEGTOME module, which *both* merges negation phrases and assigns a higher weight to the negation cue inside the merged representation, achieves the desired behavior: AP = 57.3 (best among all variants) and FPR = 47.7 (a reduction of 14.3–18.4 absolute points compared to the other ablations). These results support the view that the main bottleneck is not merely low attention on negation tokens, but the structural separation between cues and the attributes they modify.

Taken together, these controlled ablations provide converging evidence that (i) boosting the negation cue in isolation is not enough, and (ii) negation-aware phrase merging, as implemented by NEGTOME, is crucial for improving both localization (AP) and negation discrimination (FPR) on OVDEval-Negation.

# E   COMPARISON WITH POST-HOC VQA METHODS

**Motivation.**    An alternative to enhancing a detector's internal negation understanding is a two-stage pipeline, where a standard detector generates initial proposals and a powerful Multimodal Large Language Model (MLLM) then acts as a post-hoc filter to remove erroneous detections. To investigate the viability and trade-offs of this common alternative, we implemented two post-hoc VQA variants. We built these on top of the same baseline detector used in our main experiments and report results on the OVDEval Negation subset using both AP and class-ignored NMS-AP.

**Two post-hoc settings.**    *(A) Crop & Verify.* For each image, we take the detector's top-$k$ boxes, crop each region, and query an MLLM with a yes/no question about whether the crop satisfies the

Table S9: **Post-hoc VQA on OVDEval Negation.** Numbers are AP / NMS-AP (↑). "Ours" is the single-stage detector fine-tuned with deep-layer LoRA + NEGTOME. Crop & Verify improves the baseline but requires $k$ MLLM calls per image; Coordinate Prompting is faster but brittle. Stacking the expensive verifier on top of *our* improved detector yields the best overall numbers.

| Detector | Post-hoc Verifier | AP | NMS-AP |
|---|---|---|---|
| **(A) Crop & Verify** (top-$k$ crops ⇒ $k$ MLLM calls) | | | |
| G-DINO-B | | 54.0 | 36.8 |
| G-DINO-B | + Qwen-2.5-VL-3B | 59.2 | 54.4 |
| G-DINO-B + **Ours** | | 58.7 | 44.5 |
| G-DINO-B + **Ours** | + Qwen-2.5-VL-3B | **63.8** | **58.4** |
| **(B) Coordinate Prompting** (single MLLM call with all boxes) | | | |
| G-DINO-B | | 54.0 | 36.8 |
| G-DINO-B | + Qwen-2.5-VL-3B | 48.6 | 34.1 |
| G-DINO-B | + Qwen-2.5-VL-7B | 54.0 | 36.9 |
| G-DINO-B + **Ours** | | 58.7 | **44.5** |
| G-DINO-B + **Ours** | + Qwen-2.5-VL-3B | 49.6 | 37.0 |
| G-DINO-B + **Ours** | + Qwen-2.5-VL-7B | **58.6** | 44.4 |

input description. This yields $k$ separate MLLM calls per image. *(B) Coordinate Prompting.* We avoid cropping and instead pass all top-$k$ box coordinates and the description to the MLLM at once, asking it to indicate which boxes are inconsistent.

**Results.** As shown in Table S9, the *Crop & Verify* method substantially increases the baseline detector's NMS-AP from 36.8 to 54.4, confirming that a strong VQA filter can reduce contradictory detections. However, this accuracy gain comes with a heavy latency cost due to the $O(k)$ MLLM calls required per image. In contrast, the faster *Coordinate Prompting* method is unreliable for fine-grained reasoning and often degrades performance; for instance, the baseline's NMS-AP drops from 36.8 to 34.1 when paired with the 3B verifier. Notably, our single-stage method already achieves an NMS-AP of 44.5, closing much of this performance gap without any added latency. When the accurate but slow *Crop & Verify* filter is applied on top of our already-improved model, it achieves the highest NMS-AP of 58.4, indicating that our method and post-hoc verification are complementary rather than redundant.

**Conclusion.** These experiments demonstrate that while a two-stage VQA pipeline can be effective, it presents a clear trade-off between accuracy and speed. The crop-based verifier is accurate but slow, whereas coordinate prompting is fast but brittle. Our single-stage approach, by contrast, instills negation sensitivity directly within the detector, improving the stricter NMS-AP and reducing false positives in a single, efficient pass. This confirms that post-hoc filtering does not obviate the need for a negation-aware detector. For practical, real-time settings, integrating negation reasoning directly into the model's fusion layers remains the most effective path. If latency is not a concern, our work also shows that a costly verifier can be used to further refine the outputs of our model.

# F EVALUATION ON FULL OVDEVAL SUBSETS

OVDEval (Yao et al., 2024) is a comprehensive benchmark designed to evaluate the generalization capability of open-vocabulary detection (OVD) models across diverse linguistic aspects. The dataset includes 9 sub-datasets that test 6 distinct aspects: object, proper noun (landmark, logo, celebrity), attribute (color, material), position, relationship, and negation. Each subset features meticulously curated hard negative samples that challenge models to demonstrate true understanding of fine-grained linguistic descriptions rather than exploiting dataset biases. For instance, the color subset includes negative labels with the same object category but different colors, while relationship subsets maintain identical subjects and objects but alter the connecting verbs.

Figure S19: Failure Cases of Prior Models on Negation Descriptions

### F.1 THE INFLATED AP PROBLEM AND NMS-AP METRIC

Standard Average Precision (AP) metrics face limitations when evaluating fine-grained described object detection due to what OVDEval terms the *Inflated AP Problem*. This issue occurs when a model predicts multiple bounding boxes for the same object with different labels, including mutually exclusive ones as in Figure S19. For example, a model might predict both "outdoor dog led by rope" and "dog not led by ropes outside" for the same dog, artificially inflating its AP score. Mathematically, this manifests as:

$$\text{Precision} = \frac{TP}{TP + FP} = \frac{1}{1 + 1} = 0.50, \quad \text{Recall} = \frac{TP}{GT_{num}} = \frac{1}{1} = 1.0 \tag{6}$$

Where a model with no actual understanding of attributes can still achieve a mAP of 0.50. To address this, we follow OVDEval's Non-Maximum Suppression Average Precision (NMS-AP) metric (Yao et al., 2024), which applies class-ignored NMS to remove redundant predictions for the same object before AP calculation. This provides a more accurate assessment of a model's ability to understand fine-grained descriptions of contradictory pairs.

### F.2 GENERALIZATION TO NON-NEGATION SUBSETS

Table S10 demonstrates that our model maintains robust performance across all OVDEval subsets despite being trained exclusively on the negation-focused COVAND dataset. Notably, our approach shows improved NMS-AP scores for Logo (+0.2), Landmark (+4.8), Color (+0.7), and Relationship (+3.8) subsets compared to the baseline. This broad generalization suggests that our negation-sensitive adaptations enhance the model's overall reasoning capabilities for complex descriptions. These results confirm that our LoRA-based parameter-efficient fine-tuning and NEGTOME token merging strategy provide benefits beyond negation understanding, enhancing the model's capability to process compositional descriptions across multiple semantic aspects.

Table S10: **Evaluation Results on Full OVDEval.** Performance on OVDEval subsets, except for the Negation. Even though we only trained with negation-focused COVAND dataset, our models show robust results for other subsets.

|  | Logo | | Landmark | | Celebrity | | Color | | Material | | Position | | Relationship | | Average | |
|---|---|---|---|---|---|---|---|---|---|---|---|---|---|---|---|---|
|  | AP | NMS-AP | AP | NMS-AP | AP | NMS-AP | AP | NMS-AP | AP | NMS-AP | AP | NMS-AP | AP | NMS-AP | AP | NMS-AP |
| **G-DINO** | 11.7 | 7.6 | 20.5 | 16.5 | 6.7 | 0.8 | 7.9 | 5.6 | 15.2 | 5.5 | 74.7 | 60.6 | 41.3 | 18.3 | 25.4 | 16.4 |
| **+Ours** | 11.5 | 7.8 | 22.4 | 21.3 | 6.6 | 0.3 | 7.9 | 6.3 | 15.8 | 5.3 | 70.5 | 54.6 | 42.3 | 22.1 | 25.2 | 16.8 |

## G ANALYSIS ON RPN-BASED DETECTOR

### G.1 LIMITATIONS OF RPN-BASED DETECTORS UNDER NEGATION

**Marginal or negative gains with LoRA.** Two–stage region–proposal detectors such as GLIP (Li et al., 2022) and FIBER (Dou et al., 2022) obtain slight improvement on negation–focused benchmarks after attaching LoRA adapters as in Table S11. For GLIP, whose backbone consists of stacked multi–head attention blocks, we inject LoRA only into the FFN layers of the last two transformer

blocks, leaving all attention projections frozen. Even with this targeted fine-tuning, the gains remain marginal. These findings indicate that low-rank fine-tuning brings far smaller gains to GLIP than to DETR-style detectors. The gap can be traced to their attention layouts: GLIP employs self-MHA over a mixed token pool, whereas Grounding-DINO uses two modality-specific cross-attention blocks driven by a compact query set, a design that lets LoRA and NEGTOME act directly on phrase-level cues and thus respond much more strongly to negation.

**Affirmative bias and context insensitivity.** Recent negation benchmarks reveal a sharp drop in detection accuracy whenever a query expresses absence or negation (Alhamoud et al., 2025). GLIP and FIBER often treat a negated phrase (*"not X"*) as if it were *"X"*, triggering on object names while ignoring context qualifiers. Consequently, GLIP still localizes a *"microphone"* when the description states *"a person with no microphone"*, producing hallucinated objects. LoRA-adapted RPN detectors exhibit diminishing returns on negation-centric tasks because their proposal stage detects any region matching a noun, leaving little capacity to encode absence semantics.

**Performance gap between AP and NMS-AP on the Negation subset.** Table S11 further shows that model capacity alone does not resolve the issue: even the larger GLIP-L still exhibits a gap between AP and class-ignored NMS-AP, substantially wider than the gap of smaller DETR counterparts. The gap quantifies how many redundant, mutually exclusive boxes each model produces. A large drop after class-ignored NMS indicates that the detector continues to fire on the noun even when the query contains a negation cue, confirming the affirmative bias analyzed in the main paper.

**Effect of NEGTOME.** DETR keeps token granularity throughout the vision–language stack, allowing a merged phrase embedding to dominate $\langle q, k_i \rangle$ for its specific key $k_i$ while leaving other keys unaltered. By contrast, GLIP or FIBER fuse language either by (a) global pooling of the entire caption (`[cls]`), or (b) class-name pooling plus a separate visual prompt. Both strategies erase intra-sentence polarity (*dog* vs. *not dog*) before the detector sees it. Token merging cannot recover that lost contrast; at best it shortens a sequence that will be pooled anyway.

Table S11: **OVDEval-Negation Evaluation on Additional Architectures.** RPN-based detectors show a large gap between AP and NMS-AP. FT denotes fine-tuning of LoRA parameters only, with all other pretrained weights kept frozen as in the main paper. Results marked with $^\dagger$ are reproduced.

| | Image Backbone | Total Param. | AP | NMS-AP (Yao et al., 2024) |
|---|---|---|---|---|
| *Non RPN-based Detector* | | | | |
| MDETR (Kamath et al., 2021) | ResNet-101 | 185M | 41.1 | 28.3 |
| OmDet (Zhao et al., 2024b) | ConvNext-B | 242M | 55.9 | 35.1 |
| Grounding DINO$^\dagger$ (Liu et al., 2024b) | Swin-B | 233M | 54.0 | 36.8 |
| *RPN-based Detector* | | | | |
| FIBER (Dou et al., 2022) | Swin-B | 252M | 57.2 | 28.7 |
| GLIP-L (Li et al., 2022) | Swin-L | 430M | 51.8 | 29.3 |
| GLIP-T (Li et al., 2022) | | | 47.7 | 25.4 |
| +FT w.CoVAND | Swin-T | 232M | 47.8 | 26.1 |
| +FT w. CoVAND + NEGTOME | | | 48.3 | 26.0 |

## G.2 ADVANTAGES OF DETR–STYLE DETECTORS WITH LORA

**Compositional reasoning.** DETR-style detectors with transformer decoders (Kamath et al., 2021; Liu et al., 2024b; Li et al., 2023; Shen et al., 2024) perform joint text–image reasoning through cross-attention in decoder blocks. This design enables natural handling of relations such as *"X but not Y"*.

**Effectiveness of LoRA.** Injecting LoRA adapters into the decoder cross-attention layers of Grounding DINO and fine-tuning on a negation-focused dataset improves mAP by $+2.6$ and cuts the false positive rate by $11.5\%$. The same lightweight adaptation reduces spurious detections on the Negation subset of OVDEval by nearly half, while preserving general detection accuracy.

**Why the architecture helps.** Each decoder layer attends to textual tokens; negation words therefore, modulate visual attention directly. In RPN pipelines, language supervision is applied only after

proposals are fixed, limiting early rejection of forbidden objects. A fully fused DETR decoder yields a contextual representation of "what *not* to detect," which a small LoRA module can efficiently refine.

**Advantage of NEGTOME.** Both Grounding DINO and APE inherit the sub-token fragmentation of their text backbones with BERT and BPE in CLIP. NEGTOME merges those fragments into one polarity–aware phrase embedding and re-weights it by a boost factor $\beta$. In Grounding DINO, this merged vector is fed intact through token-level cross-attention, so every decoder layer receives a sharper gradient signal for the absence condition; the result is a $+10.8$ rise in NMS-AP and a $19.1\%$ drop in false positives on OVDEval-Negation. APE employs CLIP, whose text encoder pools all tokens into a single sentence vector before fusion. Here NEGTOME acts pre-pooling: by assigning larger softmax weights to the merged negation phrase it skews the sentence representation toward the correct polarity, yet does not increase sequence length. Consequently, the lightweight merger lifts APE-Ti by $+1.2$ in NMS-AP and reduces absent-object errors by $8.3\%$, despite updating only $0.017\%$ of parameters. NEGTOME aligns with the inductive bias of both encoders: it supplies BERT-based decoders with an explicit token for cross-modal attention, and it biases CLIP's global pooling toward the correct semantic polarity. The mechanism is encoder-agnostic and therefore complements LoRA across heterogeneous DETR frameworks.

DETR-based detectors fine-tuned with LoRA and NEGTOME achieve larger and more reliable gains on negation and other compositional queries than RPN counterparts. Their set-prediction decoder offers a single, expressive locus for parameter-efficient language adaptation.

# H  ZERO-SHOT DOWNSTREAM TASKS: MULTIPLE CHOICE QUESTIONS

To further analyze our model's semantic comprehension of negation, we evaluate it on the NegBench Multiple Choice Question (MCQ) benchmark (Alhamoud et al., 2025). This benchmark is specifically designed to diagnose a VLM's ability to handle negation by requiring it to select the most accurate caption for an image from four options. These options are structured into three challenging categories as detailed below, providing a fine-grained analysis of a model's capabilities.

## H.1  STRUCTURE OF THE NEGBENCH MCQ

The NegBench MCQ task (Alhamoud et al., 2025) generates multiple-choice questions where one answer is correct and the other three serve as hard negatives, designed to mislead models that do not properly understand negation. The questions are categorized into three distinct types based on the linguistic structure of the correct answer:

- **Positive Subset:** The correct caption is a simple affirmation that accurately describes objects present in the image (e.g., "*This image shows a baseball bat and baseball glove*"). This subset tests the model's fundamental visual grounding capabilities, as shown in Figure S20. Incorrect options often involve falsely negating a present object.

- **Negative Subset:** The correct caption accurately negates the presence of an object that is contextually relevant but absent from the image (e.g., "*A bowl is not present in this image*"). This directly tests the model's ability to comprehend explicit negation, as illustrated in Figure S21.

- **Hybrid Subset:** The correct caption combines both an affirmation and a negation within a single sentence (e.g., "*This image features a person, with no truck in sight*"). As shown in Figure S22, this is the most challenging subset as it requires compositional reasoning and an understanding of complex sentence structures that assign different polarities to different objects.

## H.2  ERROR PATTERN ANALYSIS OF BASELINE MODELS

Our qualitative analysis reveals that baseline models exhibit consistent and fundamental error patterns on the NegBench MCQ task, primarily stemming from a severe *affirmative bias* as below:

1. **Blatant Contradiction of Visual Facts:** The most common failure is choosing a caption that directly contradicts the visual evidence. For example, in Figure S21, the baseline model selects "*There is no horse in this image*" for an image clearly depicting a horse. This indicates that the model heavily weighs the noun ("*horse*") while effectively ignoring the negation cue, treating both affirmative and negative statements as semantically similar.

2. **Polarity Confusion in Hybrid Sentences:** In the Hybrid subset (Figure S22), baseline models systematically fail to parse sentences containing both positive and negative clauses. For instance, given the ground truth "*This image features a refrigerator, but lack of a bottle*," the baseline chooses "*This image features a bottle, but does not include a refrigerator*." This shows a critical failure in compositional reasoning, where the model cannot correctly assign presence and absence to different objects within the same logical construct.

3. **Selection of Suboptimal Negatives:** In some cases on the Negative subset, the baseline avoids direct contradiction but fails to select the most accurate description. As seen in Figure S21, when the ground truth is "*A bowl is not present*," the baseline chooses "*no cake is present*." While factually correct, this choice suggests the model lacks a deeper contextual understanding to identify the most salient absent object among multiple true negative options.

These error patterns underscore that many state-of-the-art VLMs do not understand negation. Instead, they rely on shortcut strategies that collapse the semantic meaning of affirmative and negative statements. This motivates the need for methods that can fundamentally address this architectural limitation.

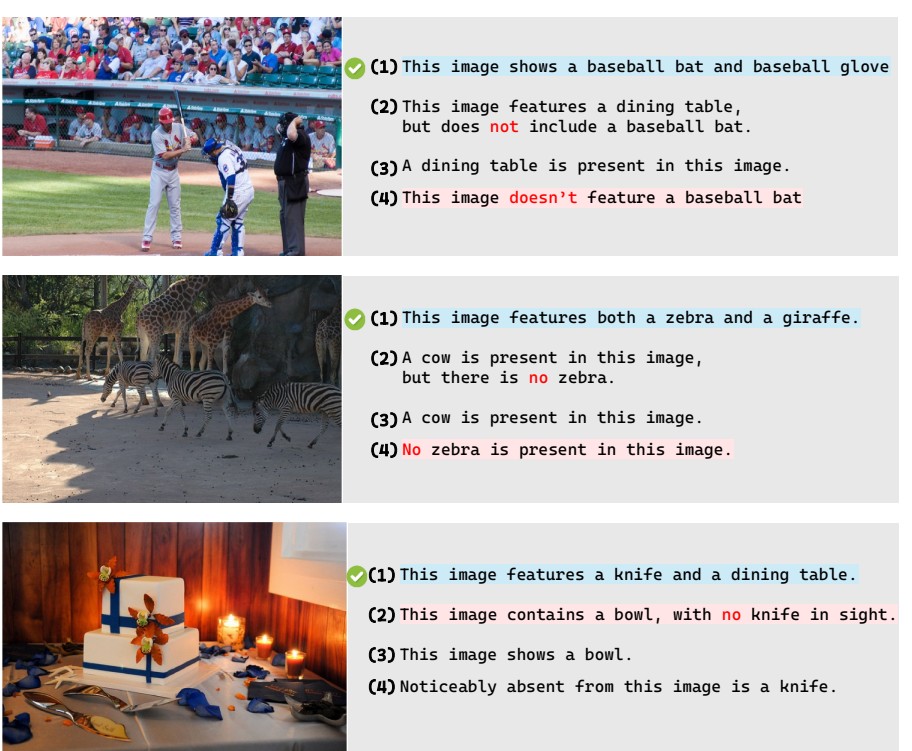

Figure S20: **Qualitative Results on the Positive subset of the Multiple Choice Question benchmark.** Captions with green checkmark ✅ is **GT**, pink refer to **Baseline**, and blue refer to **Ours**.

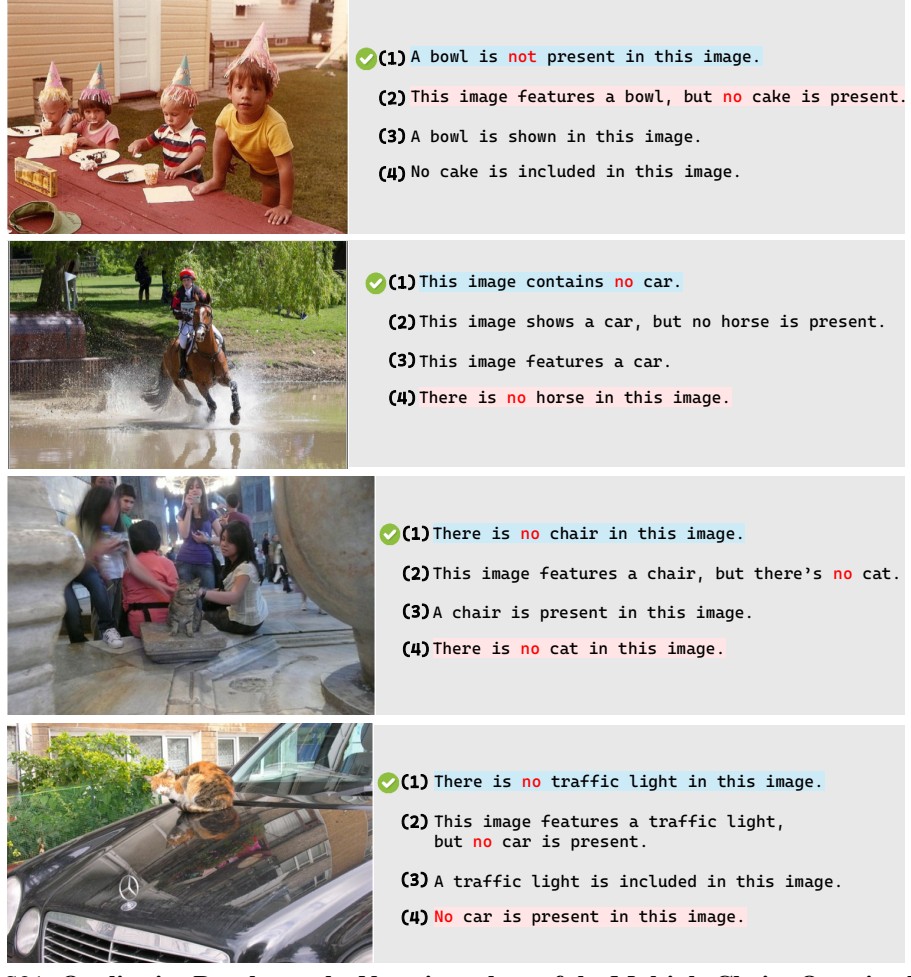

Figure S21: **Qualitative Results on the Negative subset of the Multiple Choice Question benchmark.** Captions with green checkmark ✅ is **GT**, pink refer to **Baseline**, and blue refer to **Ours**.

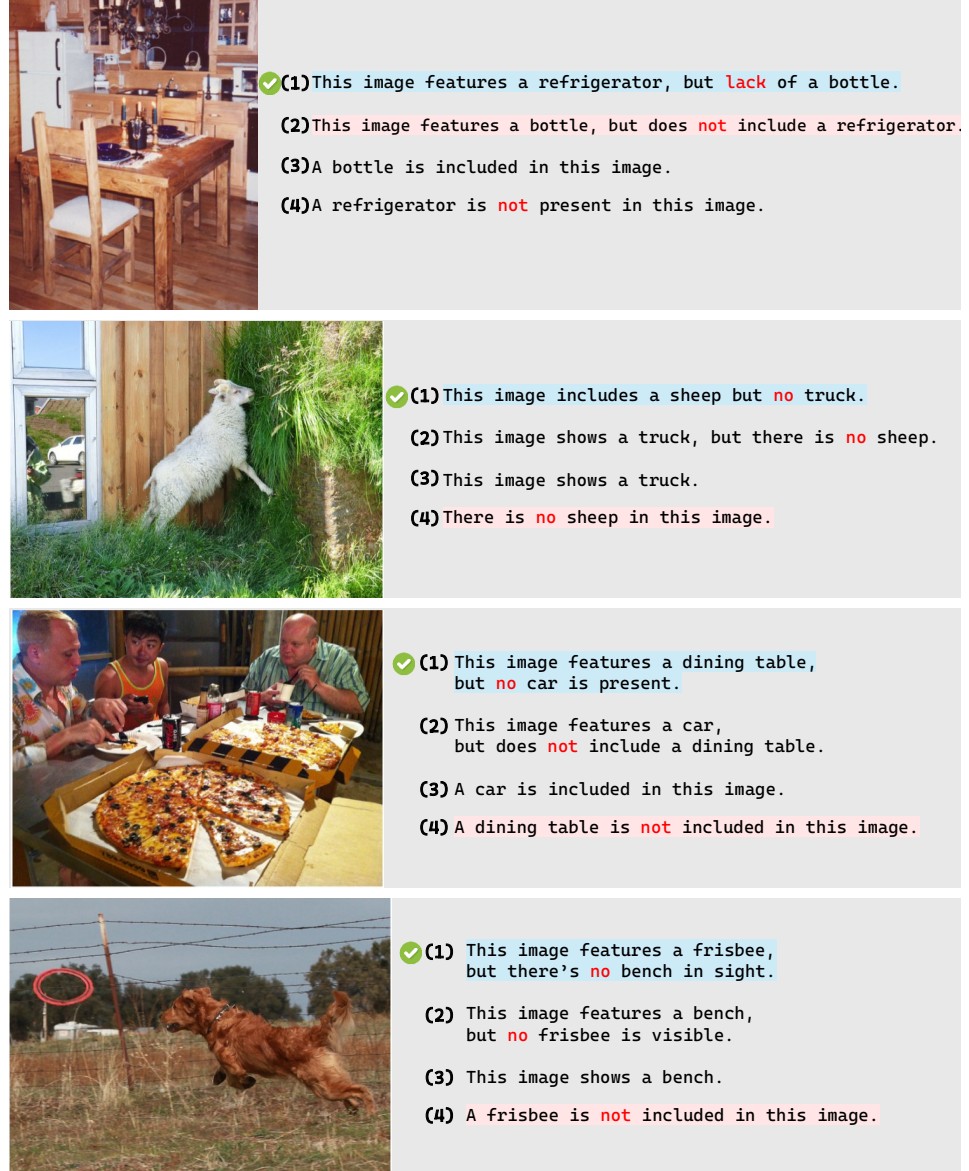

Figure S22: **Qualitative Results on the Hybrid subset of the Multiple Choice Question benchmark.** Captions with green checkmark ✅ is **GT**, pink refer to **Baseline**, and blue refer to **Ours**.

# I  ZERO-SHOT GENERALIZATION ON BIOMEDICAL DOMAIN

To validate that our method learns a robust mechanism of negation processing rather than merely memorizing specific object-negation pairs from the COVAND dataset, we conducted a zero-shot evaluation on the **Biomedical Domain**. We utilized the FG-CXR dataset (Pham et al., 2024), a fine-grained chest X-ray dataset that aligns radiologist gaze with diagnostic reports. This domain poses a severe generalization challenge due to the drastic shift in visual features (grayscale X-rays) and a distinct linguistic taxonomy of negation (e.g., "*normal*", "*clear*", "*no findings*").

## I.1  EXPERIMENTAL SETUP

Unlike standard captioning benchmarks, FG-CXR maps diagnostic sentences to 7 specific anatomical regions (e.g., *heart*, *upper left lung*). We formulated the evaluation as a **Zero-shot Binary Discrimination Task**:

- **Data Processing:** We flattened the dataset to evaluate every valid anatomical region individually. For each region in an image, we extracted the ground truth (GT) diagnosis.
- **Hard Negative Generation:** To strictly test negation understanding, we generated a contradiction for every GT sentence by syntactically flipping its polarity using a rule-based approach.
  - *State Negation:* "*The left lung is possibly **normal***" → "*The left lung is possibly **abnormal***".
  - *Absence of Findings:* "*No pleural effusion*" → "*Pleural effusion is present*".
  - *Disease Assertion:* "*Opacity in the right lung*" → "*No opacity in the right lung*".
- **Metric:** We report **Accuracy**. The model is considered correct if it assigns a higher matching score (max logit) to the GT caption than to the Hard Negative caption given the visual region.

## I.2  RESULTS AND ANALYSIS

The results of this cross-domain evaluation are presented in Table S12.

Table S12: **Zero-shot Generalization Results on FG-CXR.**

| Method | Domain | Accuracy |
|---|---|---|
| Baseline (G-DINO-B) | Biomedical (Zero-shot) | 54.86% |
| **+ Ours (NEGTOME)** | **Biomedical (Zero-shot)** | **62.55%** |

**Analysis:** The baseline model achieves an accuracy of 54.86%, which is only marginally above random chance (50%). This confirms that standard VLMs struggle deeply with medical negation, often failing to distinguish "*normal*" from "*abnormal*" even when visual features are distinct.

In contrast, our method achieves **62.55%** (+7.69%), a substantial improvement. Since our model was never exposed to medical images or jargon during fine-tuning, this gain cannot be attributed to memorizing in-domain data. Instead, it demonstrates that NEGTOME effectively structuralizes the binding between negation cues (e.g., "*no*", "*normal*") and their targets, allowing the model to generalize this reasoning mechanism to entirely new domains.

# J  QUALITATIVE RESULTS

We present additional qualitative examples from the OVDEval and $D^3$ datasets to further demonstrate the effectiveness of our negation understanding approach. Figure S23 and Figure S24 show our model's ability to distinguish between contradictory attribute pairs such as "horse urinating" versus "horse that is not urinating" and "complete pizza" versus "pizza that is not complete". The baseline model often detects identical regions for both negative and positive descriptions, demonstrating significant affirmative bias. In contrast, our method successfully differentiates between these contradictory descriptions by correctly emphasizing negation cues.

Figures S25 to S27 illustrate our model's performance on the $D^3$ dataset. For descriptions such as "hanger without clothes" and "a bed without patterns", our model correctly identifies only the objects that satisfy these negated constraints. The baseline frequently exhibits false positives by detecting objects regardless of negation markers. Our approach demonstrates particular effectiveness for simple negation cases involving physical attributes and object presence.

Despite these improvements, our method still exhibits limitations in scenarios requiring highly complex reasoning, as shown in Figure S28. These challenges often involve multi-step relational logic combined with negation, such as in the query "a woman in white wedding dress not beside any men in suits", or understanding negated states, as in "a volley ball in the middle of the air untouched". Furthermore, resolving ambiguous or implicit negation cues like "unlike" in "origami unlike bird" remains a difficult problem. A common failure pattern in these cases is that when a complex event or state is entirely absent from the image (e.g., "the person who was proposed to on one knee"), the model defaults to its affirmative bias, detecting the main subject of the query ("person") rather than correctly identifying that no object matches the full description. Crucially, the baseline model faces identical challenges in these cases, demonstrating that these are open problems for the current generation of VLM detectors. This confirms that our method, while not a complete solution for such intricate reasoning, does not degrade performance on these hardest examples. These limitations highlight important areas for future research in handling complex linguistic constructions and multi-step negation scope resolution.

## K  DECLARATIONS

**LLM usage.**  A large language model (LLM) was used during the preparation of this paper to proofread and refine the writing, including correcting grammar and improving sentence structure.

**Ethics Statement.**  Our work adheres to the ICLR Code of Ethics. The primary goal of this research is to improve the reliability and safety of vision-language models by addressing a fundamental flaw in their reasoning—the failure to understand negation. By reducing "affirmative bias," we aim to create models that align more closely with human language and intent, which can prevent critical errors in real-world applications (e.g., medical imaging or autonomous systems). Our new dataset, COVAND, is built upon the public Flickr30k Entities benchmark. The new captions are generated using a large language model (GPT-4o) with a systematic, multi-step pipeline designed to ensure high-quality, relevant, and grounded annotations. While our method improves a model's linguistic comprehension, it does not inherently address or remove societal biases that may be present in the underlying web-scale pre-training data or the baseline models themselves. We believe the contribution is a net positive, leading to more robust and predictable AI systems.

**Reproducibility Statement.**  We are committed to ensuring the reproducibility of our research. To this end, we will make our source code, including the implementation of the NEGTOME module, and the complete COVAND dataset publicly available upon publication.

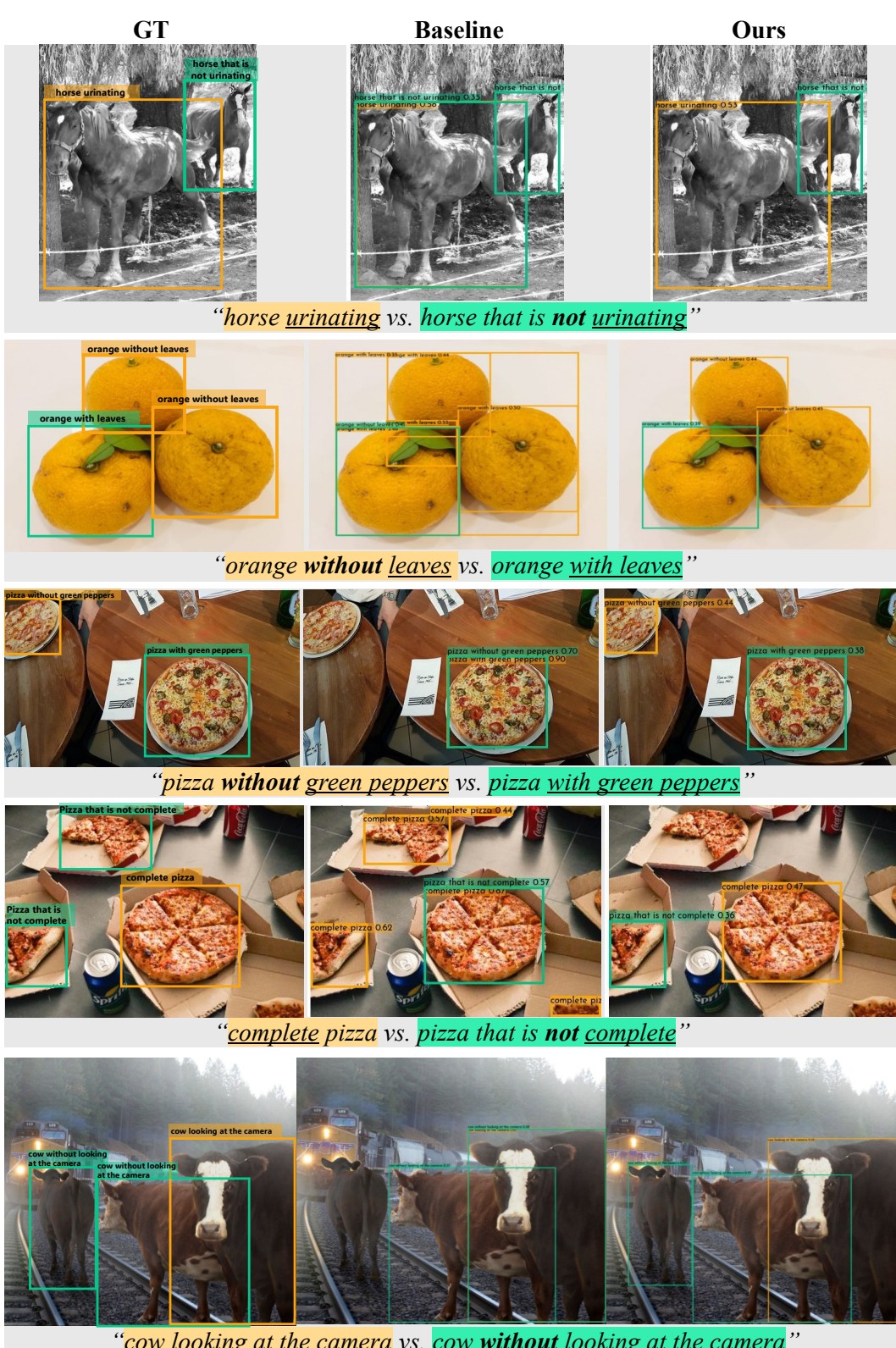

Figure S23: **Qualitative Results on OVDEval Datasets (1).** Prediction results on contradictory caption pairs (yellow box vs. green box) from the negation subset of OVDEval dataset. Each row displays (left) ground-truth boxes, (middle) baseline predictions, and (right) our predictions. Our model effectively reduces affirmative bias, no longer returning identical bounding boxes for captions that express opposite meanings.

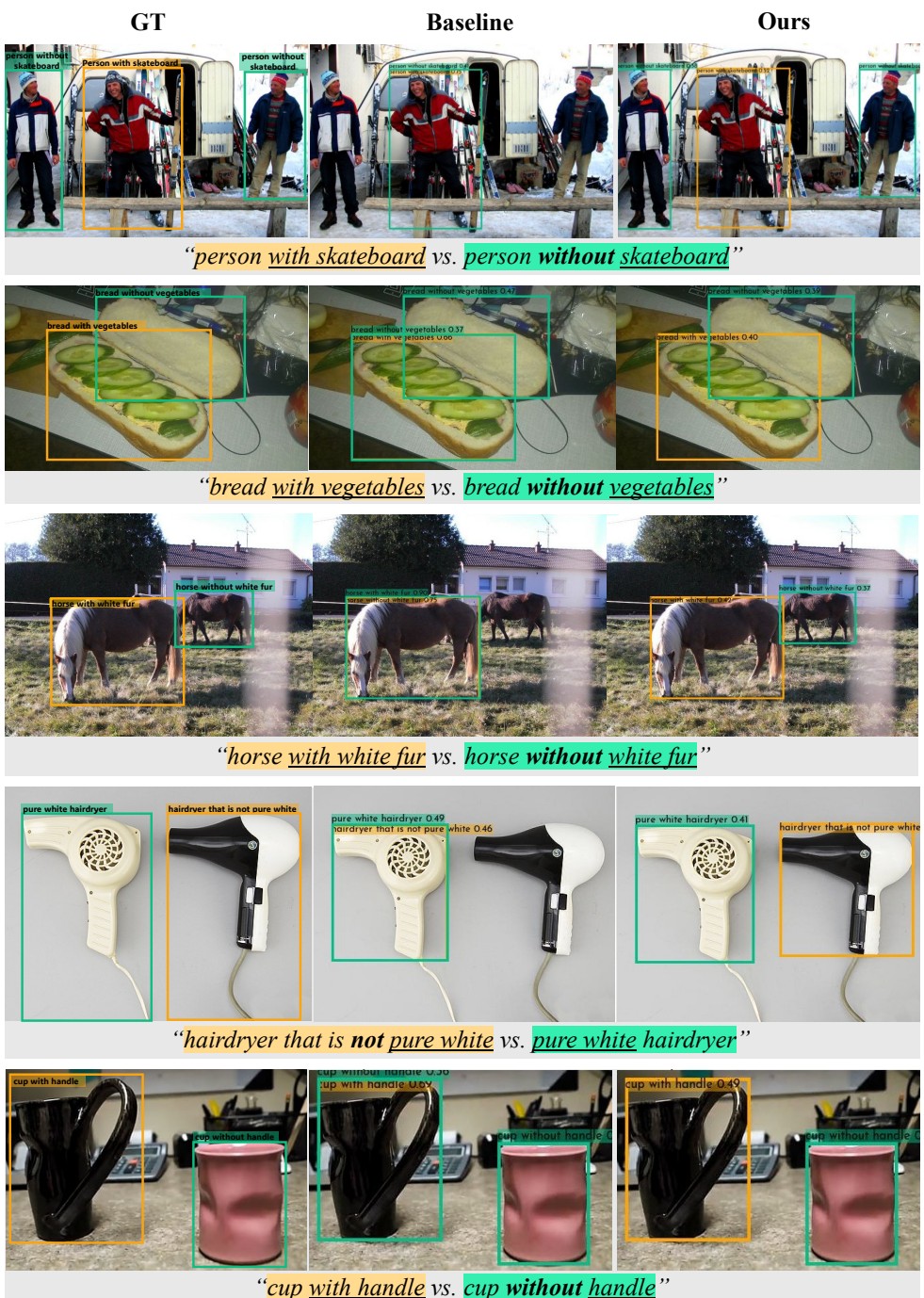

Figure S24: **Qualitative Results on OVDEval Datasets (2).** Prediction results on contradictory caption pairs (yellow box vs. green box) from the negation subset of OVDEval dataset. Each row displays (left) ground-truth boxes, (middle) baseline predictions, and (right) our predictions. Our model effectively reduces affirmative bias, no longer returning identical bounding boxes for captions that express opposite meanings.

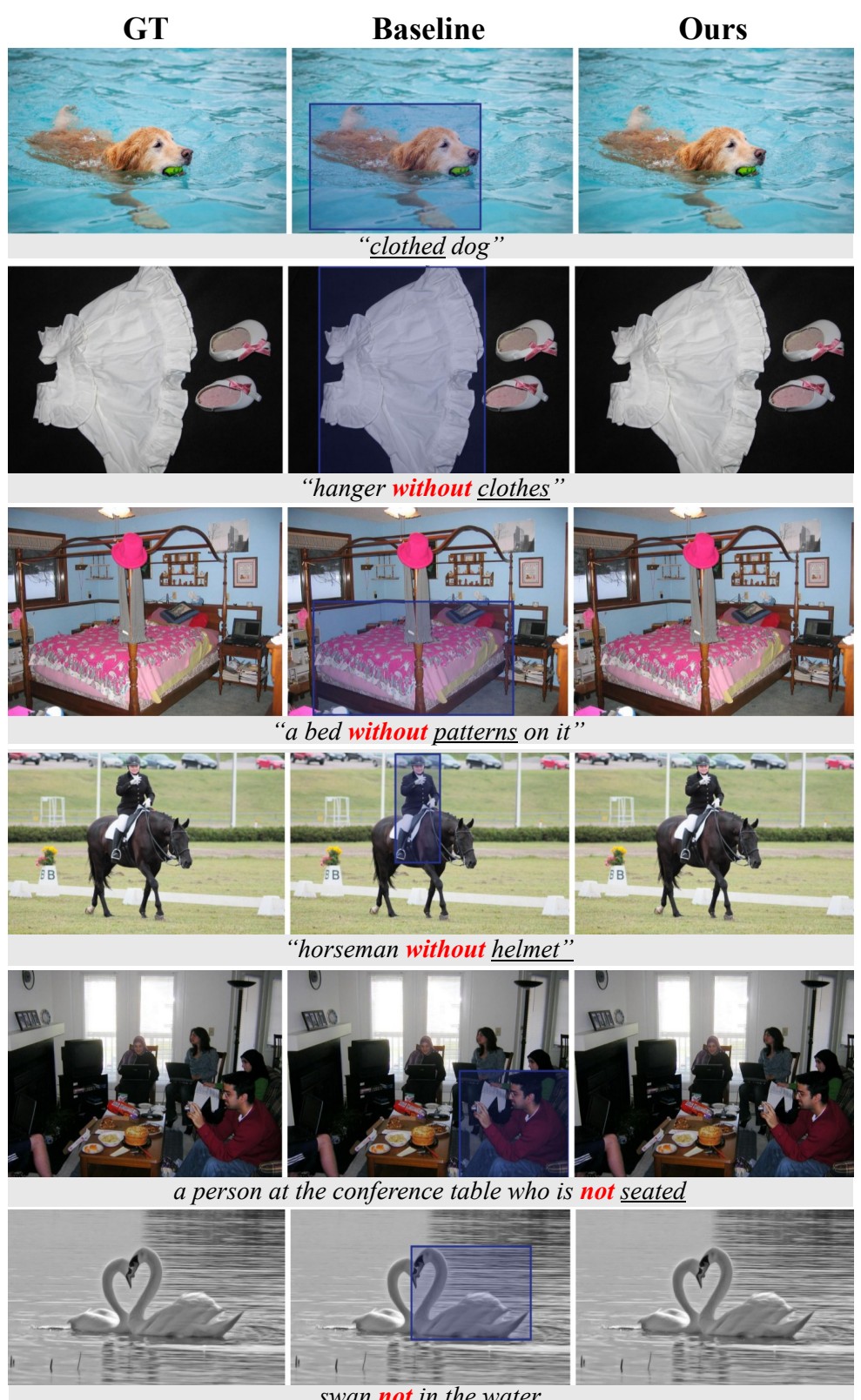

Figure S25: **Qualitative Results on D³ Datasets (1).** Absence of a bounding box shows the model has determined that no instance in the image matches the input description. By filtering out such invalid predictions, our approach reduces affirmative bias and lowers the false-positive rate.

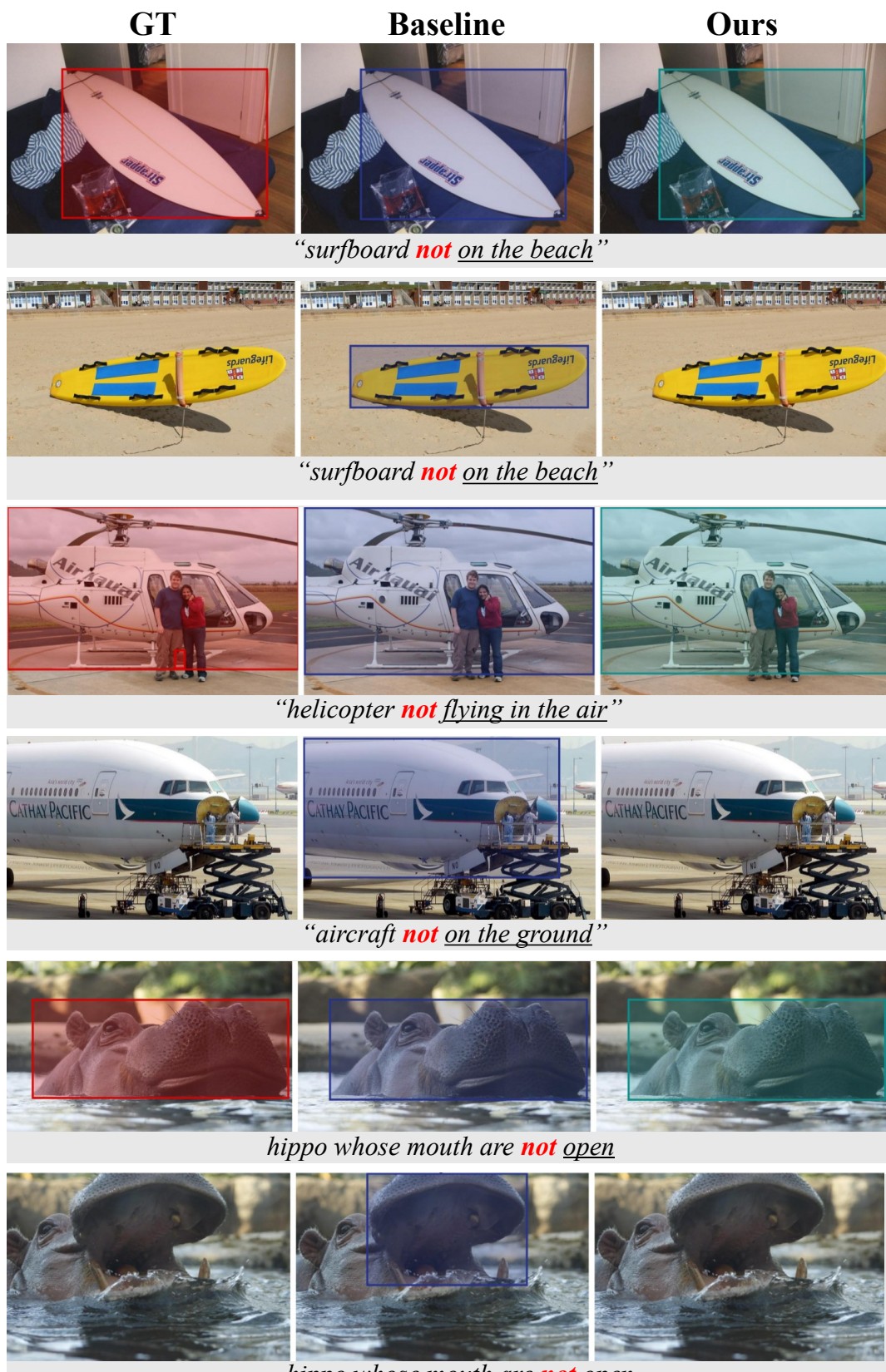

Figure S26: **Qualitative Results on D³ Datasets (2).** Absence of a bounding box means the model has determined that no instance in the image matches the input description. Our model effectively reduces the affirmative bias while keeping the correct predictions.

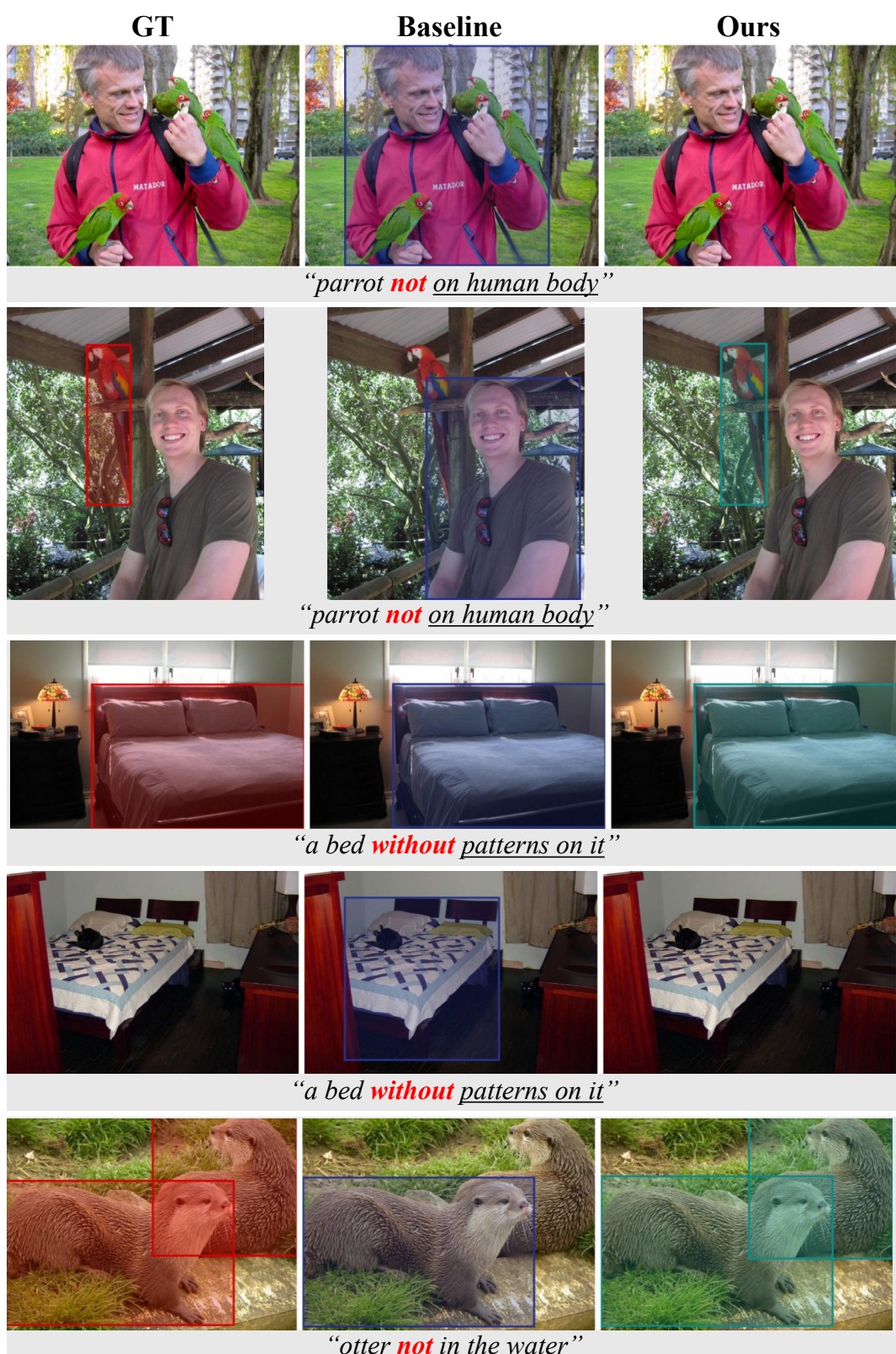

Figure S27: **Qualitative Results on D³ Datasets (3).** Absence of a bounding box means the model has determined that no instance in the image matches the input description. Our model effectively reduces the affirmative bias while keeping the correct predictions.

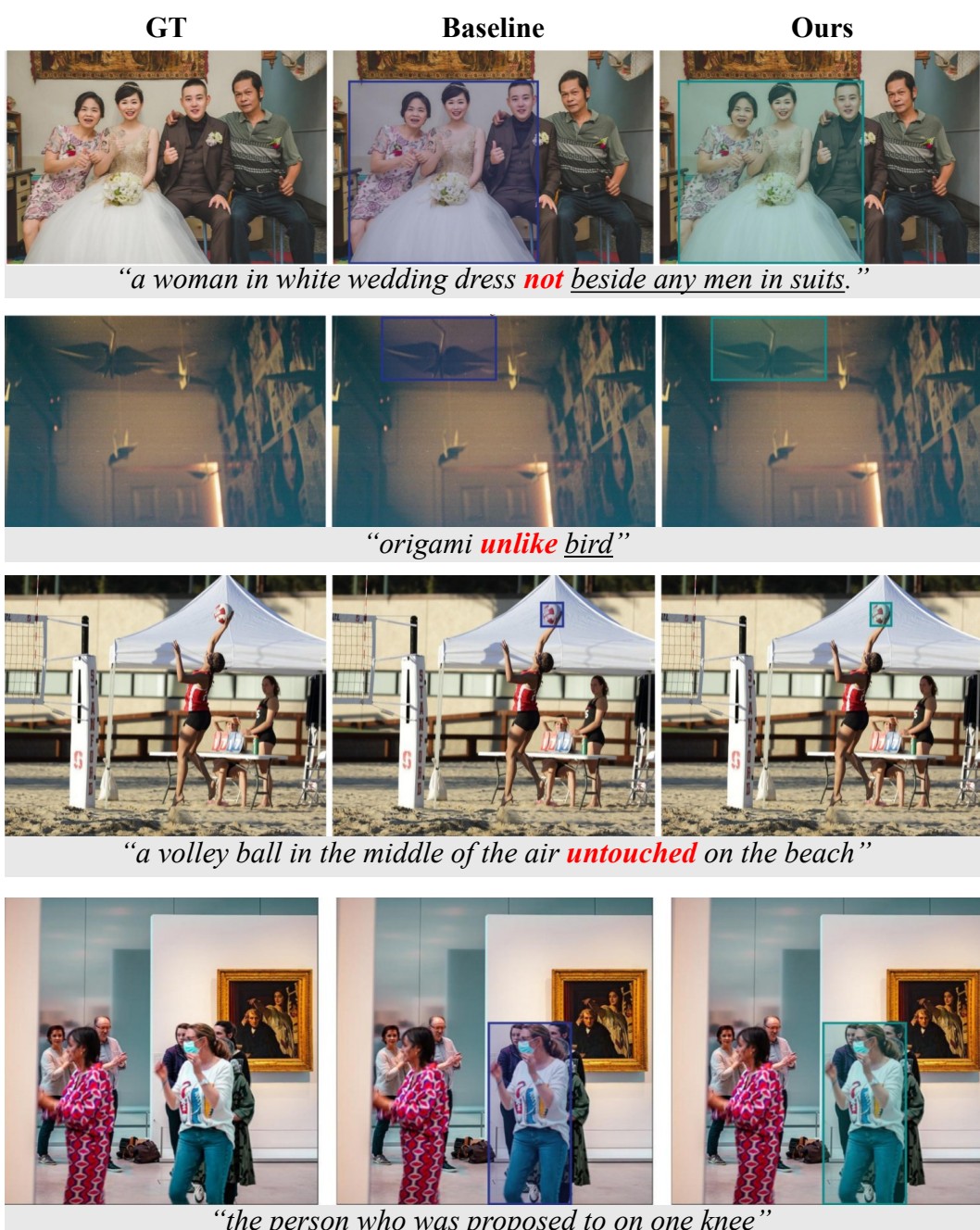

Figure S28: **Qualitative Analysis of Limitations on Complex Negation.** Despite overall improvements, our method, like the baseline, still struggles with highly complex linguistic constructions involving negation. The examples show failures in: (*i*) multi-step relational reasoning ("not beside any men in suits"), (*ii*) abstract or implicit negation ("unlike bird"), (*iii*) understanding negated states ("untouched"), and (*iv*) recognizing the absence of a complex event ("proposed to on one knee"). In these challenging cases, both models tend to default to their affirmative bias, detecting the main subject of the query rather than correctly concluding that nothing in the image matches the full description. These limitations highlight the need for more sophisticated compositional reasoning to ground complex negative constraints.

