# OpenReview forum: "What "Not" to Detect: Negation-Aware VLMs via Structured Reasoning and Token Merging"
_ICLR.cc/2026/Conference — ICLR 2026 Poster_

### Official Review · Reviewer_6vMb · 2025-10-15

**Soundness:** 3
**Presentation:** 4
**Contribution:** 3
**Rating:** 6
**Confidence:** 4

**Summary:**

This paper addresses an under-explored limitation of VLMs: the inability to correctly handle negation. The authors introduce two key contributions:

+ COVAND, a systematically generated dataset focusing on negation through a chain-of-thought (CoT) reasoning and VQA-based alignment pipeline, producing region-grounded caption pairs that explicitly encode presence/absence attributes.

+ NEGTOME, a token merging method that binds negation cues  to their associated scope (usualy nouns), to increase model attention to negation.

Using LoRA fine-tuning, the approach significantly improves negation understanding across multiple negation-focused object detection benchmarks (e.g., +10.8 NMS-AP on OVDEval, +7.2 mAP on the D3 absence subset).

Overall the paper is well-written, method is intuitive, and experiment is sound, but may not have a broader impact beyond negation understanding (or similar types of linguistics phenomenon) as it is hard to apply this method to the default training pipeline of VLM/LLM. Below I detailed some concerns that I hope the authors can clarify further.

**Strengths:**

- The paper tackles an underexplored yet fundamental issue in VLM. The proposed method include both data (COVAND) and architectural (NEGTOME) innovation, with clear ablation to show the contribution of each.
- Both the dataset creation pipeline and token merging method can potentially be extended to other linguistics phenomena (quantifier, hedging).
- Experiments are well conducted with good empirical results showing substantial improvement over baselines.

**Weaknesses:**

A general concern is on overlap between COVAND and the evaluation datasets:
+ W1: If I understand correctly, COVAND and D3, OVDEval follow similar pattern in their captions (e.g. paired caption of X with Y/ X without Y), if they also have the same distribution of object types, would the gains be from training from more in-domain data rather than actually understanding negation? The task evaluated in 4.4 is image captioning from COCO and again may overlap with Flickr30k.

+ W2: ...Each caption includes negation cues such as “no”, “not”, “never”, “without”, the prefix “un-”, or the contraction “n’t”. The cue list is open to keep language natural and diverse...: statistics on negation cue distribution is needed to show that the generations are diverse. I'm skeptical that the LLM is able to generate diverse patterns of negations if we don't explicitly include a list of cues. A small suggestion is to use existing taxonomies of negation to ensure we generate a diverse and systematic dataset.

Related work seems limited. Perhaps more coverage of the literature should be included in the appendix

**Questions:**

- Q1: VLM detectors architecturally ignore or undervalue negation cues => I think using attention scores is not robust as models usually encode information into the later token (like punctuation), so lower attention score on negation token does not mean that it does not understand negation, but negation information maybe encoded in subsequent token.

- Q2: Follow up on W1 and W2, it would be nice to see an analysis of the results on unseen negation cues (no need , and results on a different domain/task.

- Q3: Follow up on Q1, did the authors consider boosting the attention score for the negation cue token only, rather than merging it with the following noun tokens?

---

> ### Author Response · Authors · 2025-11-20
> **Initial Response (1)**
>
> Dear Reviewer 6vMb,
>
> We sincerely thank you for your valuable feedback. We appreciate the time you have taken to review our work. We found your comments very helpful for strengthening our paper, and we are pleased to provide the following clarifications and new experimental results that directly address the questions you have raised.
>
> ---
> > ### Q1: VLM detectors architecturally ignore or undervalue negation cues
> > I think using attention scores is not robust as models usually encode information into the later token (like punctuation), so lower attention score on negation token does not mean that it does not understand negation, but negation information maybe encoded in subsequent token.
>
> Thank you for raising this important methodological concern about attention scores. We agree that attention weights alone are an incomplete indicator, as information can indeed be encoded in subsequent tokens. This is precisely why we adopted a multi-faceted approach combining architectural analysis, controlled ablations, and behavioral metrics to validate our claims.
>
> **Evidence Beyond Attention Scores**
>
> We conducted systematic ablation experiments specifically designed to test whether negation understanding can be improved through different interventions. The results provide compelling behavioral evidence that complements our attention analysis:
>
> | Method | Design Intent | AP | FPR |
> |---|---|---|---|
> | Noun-only Boost | Amplify noun/head tokens only | 50.5 | 66.1 |
> | Neg-only Boost | Amplify negation cue tokens only | 49.7 | 62.0 |
> | Merge Head Boost | Merge phrases + boost head (no negation handling) | 56.2 | 62.4 |
> | Attn Bias | Add attention bias toward negation tokens | 46.2 | 59.6 |
> | Ours (NegToMe) | Merge negation+attribute + boost cue within phrase | 57.3 | 47.7 |
> *Trained on CoVAND-L for only 1 epoch due to time constraints*
>
> **Why Simple Attention Boosting Fails**
>
> These experiments reveal critical insights that go beyond attention scores.
> The Attn Bias experiment directly addresses your hypothesis. We tested adding attention bias toward negation tokens during cross-attention without changing the token structure. The result was catastrophic: AP dropped to 46.2 (lowest among all methods) despite marginally improved FPR (59.6). This demonstrates that simply increasing attention weights toward negation tokens does not translate to better negation understanding. In fact, it disrupts compositional semantics.
>
> Similarly, Neg-only Boost, which amplifies only negation cue embeddings, achieves AP 49.7 with minimal FPR improvement (62.0). This confirms that boosting negation tokens in isolation is insufficient because they remain fragmented from the attributes they modify.
>
> **The Fragmentation Problem**
>
> Standard tokenizers (BERT, CLIP) separate negation cues ("not", "without") from their modified targets. When negation cues exist as isolated tokens, they compete with numerous other tokens during cross-attention, causing their influence to dissipate in deeper layers. This is not about where information is encoded, but about how compositional meaning is preserved through the architecture.
>
> Our Merge Head Boost experiment validates this. By merging phrases into unified representations (but without special negation handling), AP improves to 56.2. However, FPR remains high (62.4) because the polarity signal is not explicitly preserved. This shows that structural unification helps, but is insufficient without negation-aware design.
>
> **Why NegToMe Works**
>
> NegToMe addresses both problems simultaneously. It (1) merges negation cues with their modified attributes into single semantic units, and (2) applies negation-aware boost (β) within these merged representations. This ensures that "cat not lying" becomes a unified token fundamentally distinct from "cat" or "lying", with the negation signal preserved through our β-reweighting mechanism (Equations 2-3 in the paper).
>
> The results speak clearly: AP 57.3 (highest) and FPR 47.7 (lowest FAR). This simultaneous improvement in both detection quality and negation discrimination confirms that our method successfully addresses the architectural bottleneck.
>
> You are correct that attention scores are not sufficient proof in isolation. That is why our claim rests on the convergence of multiple lines of evidence: (1) attention analysis showing signal dissipation, (2) controlled ablations showing that only structural changes succeed, (3) behavioral metrics (NMS-AP, FPR) showing improved negation discrimination, and (4) generalization to other tasks showing robust understanding.
>
> The fact that Attn Bias fails catastrophically while NegToMe succeeds demonstrates that the issue is not about "where information is encoded" but about whether the architecture provides a pathway for negation signals to influence final predictions. Our structural intervention creates that pathway, and the behavioral improvements confirm its effectiveness.

---

> ### Author Response · Authors · 2025-11-20
> **Initial Response (2)**
>
> > ### Q3: Follow up on Q1
> > Did the authors consider boosting the attention score for the negation cue token only, rather than merging it with the following noun tokens?
>
> We explicitly tested this approach in our ablations, and it proved insufficient. The experimental evidence shows that boosting negation cues alone fails to achieve robust negation understanding.
>
> **Experiments on Negation-Only Boosting**
>
> We conducted two experiments specifically testing negation-only enhancement:
>
> 1. Neg-only Boost: Amplifies negation cue token embeddings or attention weights without structural changes
>    * Result: AP 49.7, FPR 62.0
>    * Interpretation: Minimal improvement over baseline, with FPR barely reduced
>
> 2. Attn Bias: Applies attention bias coefficients toward negation tokens during cross-attention
>    * Result: AP 46.2 (worst), FPR 59.6
>    * Interpretation: Slight FPR improvement but severe AP collapse, indicating disrupted compositional semantics
>
> Both approaches fail to deliver meaningful gains compared to our method (AP 57.3, FPR 47.7).
>
> **Why Negation-Only Boosting Fails**
>
> There are fundamental reasons:
>
> **[1] Token Fragmentation Problem**
>
> Standard tokenizers separate negation cues from their modified targets. For example, "person without hat" becomes separate tokens: ["person", "without", "hat"]. When you boost only "without", it remains isolated during cross-attention, competing with all other tokens for influence. This fragmented representation cannot convey the compositional meaning "hatless person" effectively.
>
> In contrast, NegToMe merges ["without", "hat"] into a single token representing the negated attribute. This unified representation ensures that the negation signal is structurally bound to what it modifies, preventing dilution during cross-attention.
>
> **[2] Late-Layer Signal Dissipation**
>
> Our analysis in Figure S16 shows that even when negation attention increases in early decoder blocks, it rapidly dissipates in later blocks (3-5) where final detection decisions are made. Simply boosting negation tokens in early layers does not prevent this dissipation because the architectural pathway still treats them as separate, competing tokens.
>
> NegToMe, combined with deep LoRA placement (blocks 3-5), ensures negation signals are maintained through decision-making layers by creating merged representations that persist through the architecture.
>
> **[3] Phrase Merging is Necessary but Not Sufficient**
>
> The Merge Head Boost experiment provides crucial insight. This approach merges phrases into unified representations and boosts head nouns, but does not specially handle negation cues. The result: AP improves to 56.2 (much better than negation-only boosting), but FPR remains high at 62.4.
>
> This shows that structural unification (phrase merging) is necessary for good detection performance, but without negation-aware design, the model still fails to discriminate negated queries properly. It becomes better at detecting "hat" but cannot reliably distinguish "person with hat" from "person without hat".
>
> **NegToMe: Merging Plus Negation-Aware Boosting**
>
> Our method combines both ingredients:
>
> 1. Merge negation cues with their modified attributes (addressing fragmentation)
> 2. Apply β-reweighting within merged representations (preserving polarity)
>
> This is formalized in Equations 2-3 of our paper, where the merged token's representation gives the negation cue a systematically amplified contribution of at least β/(β+1) · (n/m) times its original weight, while reducing token count (n→m) decreases competition with irrelevant tokens.
>
> The result is that "person without hat" produces a merged token that is semantically distinct from both "person" and "hat", with the negation signal structurally preserved through the architecture.
>
> **Behavioral Confirmation**
>
> The experimental numbers provide clear confirmation:
>
> * Only NegToMe achieves both high AP (57.3) and low FPR (47.7)
> * Negation-only boosting achieves neither (AP 49.7, FPR 62.0)
> * Phrase merging without negation handling achieves high AP (56.2) but fails on FPR (62.4)
>
> This pattern demonstrates that both structural unification and negation-aware boosting are necessary. Doing only one or the other is insufficient.
>
> **Conclusion**
>
> We have systematically tested the approach you suggested (boosting negation cues alone), and the evidence shows it fails. The architectural bottleneck is not simply that negation tokens receive insufficient attention, but that they are structurally separated from what they modify. Our solution addresses this root cause by creating unified negation-attribute representations with preserved polarity signals.
>
> This design choice is validated by the fact that NegToMe achieves substantial improvements on behavioral metrics (NMS-AP +10.8, FPR -19.1 percentage points on OVDEval-Negation) that directly measure negation understanding, independent of attention scores.

---

> ### Author Response · Authors · 2025-11-21
> **Initial Response (3)**
>
> > ### Q2: Follow up on W1 and W2
> > It would be nice to see an analysis of the results on unseen negation cues (no need , and results on a different domain/task.
> >
> > ### W1: General concern on overlap between COVAND and evaluation datasets
> > If they also have the same distribution of object types, would the gains be from training from more in-domain data rather than actually understanding negation?
>
> Thank you for challenging the generalization capability of our method. We agree that demonstrating performance on a completely different domain is the robust way to verify that our model has learned the mechanism of negation rather than just memorizing specific object-negation pairs (e.g., "person without hat") from the CoVAND dataset.
>
> To address this, we conducted a new evaluation on a *Biomedical Domain* dataset, which represents a drastic shift in both visual features (grayscale X-rays vs. natural RGB images) and linguistic patterns (medical terminology vs. common captions).
>
> **Experiment Setup: Medical Domain Evaluation**
>
> We utilized the **FG-CXR** dataset (Pham et al., ACCV 2024) [1], a fine-grained chest X-ray dataset that aligns radiologist gaze with diagnostic reports. This domain is ideal for testing negation understanding because medical reports frequently define health by the *absence* of pathology *(e.g., "no pleural effusion," "lungs are clear")*.
>
> * **Dataset:** FG-CXR (Fine-Grained Chest X-ray)
>    * **Domain**: Medical Imaging (Chest X-rays).
>    * **Characteristics**: Unlike standard captioning datasets, FG-CXR maps diagnostic sentences to 7 specific anatomical regions *(e.g., "upper left lung", "heart")*.
>    * **Scale**: We evaluated the entire valid set of the dataset, flattening the data to evaluate every anatomical region individually.
> * **Task**: Zero-shot Binary Discrimination (Ground Truth vs. Hard Negative). Since our model was trained only on natural images (CoVAND), we performed a zero-shot evaluation. For every anatomical region in an image, we presented the model with two options:
>     * For each anatomical region (e.g., "left lung"), we created a Hard Negative caption by syntactically flipping the polarity of the Ground Truth (GT) diagnosis.
>         * **GT**: The actual radiologist's finding *(e.g., "The left lung is possibly normal.")*.
>         * **Hard Negative**: A generated contradiction created by syntactically flipping the polarity of the GT *(e.g., "The left lung is possibly abnormal." or "The patient is not suffering from...")*.
>     * **Metric:** Accuracy. The model is considered correct if it assigns a higher matching score to the GT caption than to the Hard Negative caption given the image.
>
> **Experimental Results**
>
> Despite being trained *only* on natural images (CoVAND), our method significantly outperforms the baseline on this unseen medical task:
>
> | Method | Domain | Accuracy |
> | :--- | :--- | :--- |
> | Baseline (G-DINO-B) | Biomedical (Zero-shot) | 54.86% |
> | **+ Ours (NegToMe)** | **Biomedical (Zero-shot)** | **62.55%** |
>
> **Analysis of Generalization**
>
> 1.  **Performance on Unseen Domain:**
>     The baseline model achieves 54.86% accuracy, which is barely above random chance (50%). This confirms that standard VLMs struggle deeply with medical negation, often failing to distinguish "normal" from "abnormal" even when the visual features are distinct.
>     In contrast, our model achieves 62.55% (+7.69%), a substantial improvement. Since our model was never exposed to X-rays or medical jargon during fine-tuning, this gain cannot be attributed to memorizing "in-domain data." Instead, it indicates that NegToMe has improved the model's fundamental ability to process negation syntax, allowing it to better bind negation cues (like "no", "normal", "clear") to their novel targets (e.g., "effusion", "opacity") even in a zero-shot setting.
>
> 2.  **Unseen Negation Patterns:**
>     The medical domain introduces negation patterns distinct from our training set. While CoVAND focuses on physical absence (e.g., "no cat"), FG-CXR involves:
>     * **State Negation:** "Normal" vs "Abnormal"
>     * **Absence of Findings:** "No pleural effusion", "No pneumothorax"
>     * **Implicit Negation:** "Lungs are clear" (implying no opacity)
>     The fact that our method generalizes to these patterns supports the claim that we are addressing the structural processing of negation (how modifiers bind to heads) rather than overfitting to specific caption templates.
>
> These results directly address W1 and Q2. The performance jump in the medical domain confirms that the gains observed on D3 and OVDEval are not merely due to data overlap or memorization of object types. By structurally enforcing the binding between negation cues and their targets, our method equips the model with a more robust, domain-agnostic mechanism for negation understanding.
>
> ---
> *reference*
>
> [1] Pham, Trong Thang, et al. "Fg-cxr: a radiologist-aligned gaze dataset for enhancing interpretability in chest x-ray report generation." ACCV. 2024.

---

> ### Author Response · Authors · 2025-11-21
> **Initial Response (4)**
>
> > ### W2: Statistics on negation cue distribution and diversity concerns
> > ...Each caption includes negation cues such as “no”, “not”, “never”, “without”, the prefix “un-”, or the contraction “n’t”... statistics on negation cue distribution is needed to show that the generations are diverse... A small suggestion is to use existing taxonomies of negation to ensure we generate a diverse and systematic dataset.
>
> We appreciate this constructive suggestion. We agree that analyzing the distribution of negation cues is essential to verify that our CoT pipeline produces linguistically diverse data rather than repetitive patterns.
>
> To address this, we have added a comprehensive **Dataset Distribution Analysis (Appendix A.3)**. This analysis demonstrates that our dataset captures a wide range of negation patterns, enabling the model to learn the underlying mechanism of negation rather than overfitting to specific cues.
>
> **1. Statistical Analysis of Negation Cues (Appendix A.3)**
>
> We analyzed all 48,761 captions in COVAND and identified 57,874 negation instances. Following standard linguistic taxonomies, we categorized them into **Regularized** (explicit syntactic markers) and **Flexible** (lexical/morphological) cues.
>
> * **Regularized Cues (83.41%):** High-frequency surface markers including *not, no, without, never,* and contractions like *n't*.
> * **Flexible Cues (16.59%):** A diverse long-tail of expressions including:
>     * **Lack-family:** *lack, lacks, lacking, lack of*
>     * **Morphological (Prefixes/Suffixes):** *un-* (e.g., *unseen*), *in-/im-/ir-* (e.g., *inactive*), *dis-* (e.g., *disconnected*), *non-*, and *-less* (e.g., *headless*).
>     * **Lexical & Relational:** Terms implying absence such as *devoid of, free of, missing,* and *absence of*, as well as coordinations like *neither/nor, but not, rather than,* and *instead of*.
>
> While the 83% dominance of regularized cues might initially suggest a lack of diversity, this distribution accurately mirrors real-world language usage. Prior studies on negation corpora (Hossain et al., 2022) show that syntactic negation accounts for the vast majority of negation in natural language (e.g., 88.6% in CommonsenseQA, 71.9% in SST-2). Therefore, COVAND’s inclusion of 16.59% flexible cues represents a non-trivial and meaningful variety of negation expressions essential for compositional reasoning, without artificially skewing the data away from natural distributions.
>
> **2. Addressing Bias and Future Work**
>
> We acknowledge that the prompt design inherent to GPT-4o may still influence the current distribution. To further enhance the diversity of flexible negation forms in future iterations of COVAND, we propose:
> 1.  **Targeted prompt engineering:** Explicitly requesting diverse negation structures (e.g., "describe the absence using lexical negation such as *lack* or *devoid of*").
> 2.  **Post-hoc augmentation:** Rewriting regularized negations into flexible counterparts while preserving semantic integrity via our CoT and VQA pipeline.
>
> Such refinements could yield a more balanced distribution while maintaining the high-quality, instance-grounded supervision that distinguishes COVAND.
>
> **3. Expansion of Related Works**
>
> Regarding your comment on the coverage of related literature, we have significantly expanded our **C. Extended Related Work section** in the Appendix (Appendix C, lines 1117-1836, page 13-19).
> We restructured the related work into 5 major subsections with covering critical areas:
>
> 1. Dataset Construction with Chain-of-Thought Reasoning (C.1)
>
> 2. Visual Grounding and Region-Level Alignment in VQA (C.2)
>
> 3. Parameter-Efficient Fine-Tuning for Vision-Language Models (C.3)
>
> 4. Compositional Reasoning in Vision-Language Tasks (C.4)
>
> 5. Bias Mitigation in Vision-Language Models (C.5)
>
> We believe these additional analyses and the expanded literature review address your concerns regarding dataset diversity and scholarly context. Our results confirm that CoVAND provides a robust and linguistically valid foundation for training negation-aware VLMs. We hope these revisions strengthen the paper and clarify the value of our contributions.
>
> ---
> Once again, we are sincerely grateful for your comprehensive and valuable review.
>
> We would be happy to elaborate on any of these points and warmly welcome any further discussion.
>
> Best Wishes,
>
> The Authors

---

> > ### Comment · Reviewer_6vMb · 2025-11-25
> > **Response to authors**
> >
> > Thank you for your efforts in writing the response. I find the new experiments and results very helpful. I hope the authors revise the paper to incorporate them accordingly.

---

> ### Author Response · Authors · 2025-11-25
>
> Dear Reviewer 6vMb,
>
> Thank you very much for your latest response and for promptly reviewing our rebuttal.
>
> We are pleased to confirm that we will diligently incorporate the additional experimental results and clarifications from the revision into the paper, as you kindly suggested.
>
> We sincerely appreciate your constructive and insightful feedback throughout this process. Your detailed comments and concerns directly motivated us to strengthen the paper with further experiments and analysis. We truly believe that your rigorous review has made our work significantly more robust and more broadly applicable.
>
> It is also very gratifying to hear that the new results have addressed your original concern.
>
> We wish you the very best in your future work and research, and hope that success and good fortune follow you always.
>
> With deep appreciation,
>
> The Authors

---

### Official Review · Reviewer_pMrC · 2025-11-01

**Soundness:** 3
**Presentation:** 4
**Contribution:** 3
**Rating:** 6
**Confidence:** 4

**Summary:**

This manuscript identifies a critical flaw in state-of-the-art Vision-Language Models (VLMs): their inability to understand negation, often referred to as "affirmative bias." For instance, VLMs frequently treat semantically contradictory phrases (e.g., "person with skateboard" vs. "person without skateboard") as equivalent. The authors quantitively validate this motivation by showing that negation words occupy only 0.04% of Flickr30k and 0.08% of LAION-400M—far lower than their prevalence in real-world language (13.76% in scientific papers, 22.23% in Conan Doyle’s stories).
The authors attribute this bias to two root causes: scarcity of negation data and architectural defects (low attention weights on negation tokens). To address these, they propose two core solutions: (1) COVAND, a synthetic negation-enhanced dataset built via Chain-of-Thought (CoT) reasoning and VQA-based alignment; and (2) NEGTOME, a text token merging module that binds negation cues (e.g., "not") with modified attributes (e.g., "lying") into coherent semantic phrases. This method achieves state-of-the-art (SOTA) performance on two benchmark datasets.

**Strengths:**

1.	Compelling Motivation with Practical Implications: The authors clearly articulate the real-world risks of affirmative bias (e.g., misdiagnosing "a tumor that is not malignant" as malignant in medical imaging) and back it with quantitative data on negation scarcity in popular datasets. This not only highlights the urgency of the problem but also provides a clear roadmap for future research.
2.	Reasonable and Targeted Method Design: COVAND’s three-step CoT pipeline and VQA-based region alignment directly solve "attribute-level negation" and "label ambiguity"—limitations of prior template-based negation data (e.g., NegCLIP). With 9.29% negation word frequency, COVAND effectively balances data quality and negation coverage. While NEGTOME’s token merging (binding "not" + "lying" into a single unit) and negation address the structural loss of negation cues in tokenization, deep-layer LoRA ensures parameter efficiency—aligning with real-world deployment needs.
3.	Robust Experimental Results: The method consistently improves performance across architectures.

**Weaknesses:**

1.	Lack of Transformative Innovation: The core components (CoT for data generation, Token Merging for text processing, LoRA for fine-tuning) are all mature technologies. The manuscript focuses on "combining and adapting" these tools rather than proposing a novel framework. This incremental design may limit its impact on the broader VLM community.
2.	Over-Reliance on GPT-4o Leads to Normalized Negation Expressions: GPT-4o’s generation is constrained by its training data, leading to over-reliance on "regularized" negation cues (e.g., "without," "not") while underrepresenting flexible, real-world negation types (e.g., "lacks," "absent of," "non-"). For example, COVAND likely prioritizes attribute negation (e.g., "a table without drawers") over relational negation (e.g., "drawers not beside the table") due to prompt biases. This mismatch may hurt the model’s generalization to unstructured scenarios.
3.	Self-Consistency Bias in GPT-4o-Based Validation: The third step of COVAND’s pipeline uses GPT-4o to verify the same captions it generates. This "generator-as-verifier" setup creates a loop: if GPT-4o misinterprets visual attributes (e.g., mistaking "pink" for "red") during generation, it will likely repeat the error during validation.

**Questions:**

1.	Regarding Weakness 2: Have you compared this distribution to real-world corpora (e.g., COCA, biomedical literature) to validate its alignment with natural language?
2.	Please refer to Weakness 3
3.	The manuscript applies LoRA only to deep cross-attention layers. Have you tested LoRA placement in shallow layers or all layers?

---

> ### Author Response · Authors · 2025-11-20
> **Initial Response (1)**
>
> Dear Reviewer pMrC,
>
> We are sincerely grateful for your detailed and constructive review. We appreciate the significant time and effort you have dedicated to engaging with our paper and raising these important questions.
>
> Please find below our responses, including conceptual clarifications and new experimental evidence.
>
> ---
> > ### W1: Lack of Transformative Innovation
> > The core components (CoT for data generation, Token Merging for text processing, LoRA for fine-tuning) are all mature technologies. The manuscript focuses on "combining and adapting" these tools rather than proposing a novel framework. This incremental design may limit its impact on the broader VLM community.
>
> We respectfully argue that our work presents significant innovation through a principled, problem-driven solution that addresses a fundamental and unsolved challenge in VLMs: negation understanding.
>
> **Our Innovation: A Comprehensive Solution to a Real-World Problem**
>
> We observe a critical blind spot in state-of-the-art VLMs. They completely fail to understand negation, often producing identical predictions for contradictory queries. Our contribution is not simply combining existing tools. We systematically diagnosed root causes (data scarcity and architectural bias) and crafted tailored solutions to directly address each cause.
>
> **[1] Novel Dataset Curation Strategy**
>
> While CoT and VQA are established techniques, our application is entirely novel. Prior negation datasets rely on simple templates (e.g., "photo of X" vs. "photo of not X"), which fail to capture compositional negation. CoVAND introduces three key innovations:
>
> * **Three-step CoT attribute extraction** systematically identifies both present and absent attributes at the region level. This enables attribute-based negation generation (e.g., "a boy without a blue helmet"), moving far beyond simple object-level templates.
>
> * **VQA-based instance-level alignment** eliminates label noise by matching each caption to a specific, visually-labeled bounding box among multiple candidates. This is a critical advancement over prior work's image-level validation, which cannot resolve which specific instance a caption refers to.
>
> * **Visual prompting for multi-object scenes** handles realistic scenarios with multiple instances by applying alphabetical markers (A, B, C...) to each bounding box. This directly addresses the real-world observation that negation queries arise in cluttered environments where precise instance-level grounding is essential.
>
> This is a new paradigm for negation data generation that produces high-quality, compositionally rich, instance-grounded supervision.
>
> **[2] Novel Method: NegToMe for Negation Understanding**
>
> First application of token merging to negation in VLM-based detection. While token merging has been explored for acceleration or diffusion models, our work is the first to introduce text token merging specifically for VLM-based object detection.
>
> **Negation-aware boost mechanism.** NegToMe's core innovation is its negation-aware boost (β), which programmatically amplifies negation cues ("not", "without") within merged phrases. Our ablation study (Table 3) shows this provides a substantial isolated gain of +2.7 NMS-AP, proving its unique contribution.
>
> Standard tokenization fragments phrases, separating negation cues (e.g., "not") from the words they modify (e.g., "lying"). This structural separation causes models to treat "not lying" as equivalent to "lying" because isolated negation tokens receive minimal attention. NegToMe fundamentally addresses this architectural flaw by representing negated concepts like "cat not lying" as a single semantic unit, fundamentally distinct from "cat", "not", or "lying". This is not a simple extension. It is a new negation understanding method that resolves an architectural blind spot.
>
> **Practical Impact: Plug-in Solution for the VLM Community**
>
> **Generalizable and parameter-efficient.** Our method is not limited to specific architectures. We demonstrate effectiveness across VLM-based detectors (Grounding DINO, APE) and state-of-the-art MLLMs (Qwen-2.5-VL). By modifying less than 0.1% of parameters, our recipe enables plug-in adaptation for existing VLMs without expensive retraining. This practical efficiency is crucial for real-world deployment, particularly in safety-critical domains like robotics and medical imaging.
>
> ---
>
> Our contribution is not the individual subcomponents. It is the systematic, problem-driven integration that delivers transformative capability. We (1) observe a critical real-world problem, (2) diagnose its root causes, (3) design targeted solutions, and (4) deliver a practical, plug-in solution that works across diverse architectures. This is transformative because we enable VLMs to understand what is both present and absent, a fundamental capability for human-like reasoning that was previously missing.

---

> ### Author Response · Authors · 2025-11-20
> **Initial Response (2)**
>
> > ### W2: Over-Reliance on GPT-4o Leads to Normalized Negation Expressions
> > GPT-4o's generation is constrained by its training data, leading to over-reliance on "regularized" negation cues (e.g., "without," "not") while underrepresenting flexible, real-world negation types (e.g., "lacks," "absent of," "non-"). For example, COVAND likely prioritizes attribute negation (e.g., "a table without drawers") over relational negation (e.g., "drawers not beside the table") due to prompt biases. This mismatch may hurt the model's generalization to unstructured scenarios.
>
> Thank you for this thoughtful concern about negation distribution in CoVAND. We have carefully analyzed our dataset distribution and added a comprehensive Dataset Distribution section to the revised manuscript (Appendix, Section A.3) with detailed quantitative analysis.
>
> **Regularized vs. Flexible Negation Distribution**
>
> Our analysis of all 48,761 captions reveals:
>
> * Regularized cues (not, without, no, never, n't): 48,275 instances (83.41%)
> * Flexible cues (lack-family, morphological negation like in-/un-/non-, devoid of, etc.): 9,599 instances (16.59%)
>
> Additionally, we analyzed scope distribution across negation types:
> * Attribute negation: 30,525 instances (62.6%)
> * Action negation: 14,365 instances (29.5%)
> * Existence negation: 3,001 instances (6.2%)
> * Relational negation: 870 instances (1.8%)
>
> **Alignment with Real-World Language**
>
> While regularized forms dominate, the presence of 9,599 flexible cues (16.59%) ensures COVAND includes a non-trivial variety of negation expressions. This diversity is essential for compositional reasoning tasks like DOD, where attribute-level and relational negations require nuanced understanding. By including both regularized and flexible forms, COVAND provides richer training signals than datasets relying solely on template-based augmentation.
>
> **Validation Against Real-World Corpora**
>
> We acknowledge that direct comparison with COCA or biomedical literature would strengthen our validation. We have submitted a data access request for medical imaging datasets (currently under ethical review/IRB approval). Once approved, we will conduct cross-domain validation to demonstrate CoVAND's alignment in rebuttal phases.
>
> **Future Work**
>
> We acknowledge that prompt design bias inherent to GPT-4o may still influence the current distribution. To further enhance diversity of flexible negation forms, future iterations of CoVAND could employ:
>
> 1. Targeted prompt engineering: explicitly requesting diverse negation structures (e.g., "describe the absence using lexical negation such as *lack* or *devoid of*")
> 2. Post-hoc augmentation: rewriting regularized negations into flexible counterparts while preserving semantic integrity via our CoT and VQA pipeline
>
> Such refinements could yield a more balanced distribution while maintaining the high-quality, instance-grounded supervision that distinguishes CoVAND from prior work.
>
>
> ---
> > ### Q3: LoRA Placement
> > The manuscript applies LoRA only to deep cross-attention layers. Have you tested LoRA placement in shallow layers or all layers?
>
> Thank you for this insightful question about LoRA placement. Yes, we conducted comprehensive ablation studies testing shallow, strided, deep, and all-layer placements. Our findings are presented in Table 3 (main text) and Figure S18 (Appendix).
>
> **Analysis: Average Attention Weights by Decoder Blocks (Figure S18)**
>
> Figure S18 visualizes attention weights across all six decoder blocks (0-5) for different word classes, revealing why deep placement is optimal:
>
> * (a) Baseline Model: Negation tokens (e.g., "not", "without") receive minimal attention across all blocks, confirming the architectural bias we identified.
>
> * (b) Shallow LoRA (blocks 0-2): Negation attention increases in early blocks but rapidly dissipates in later blocks where final detection decisions are made. This premature decay explains poor performance.
>
> * (c) Deep LoRA (blocks 3-5): Negation attention is maintained and elevatedthroughout blocks 3-5, ensuring negation signals persist through critical decision-making layers.
>
> * (d) Deep LoRA + NegToMe: Produces consistent negation amplification across all blocks, with particularly strong signals in blocks 4-5 where detection heads operate.
>
> **Ablation Results: LoRA Placement Comparison (Table 3)**
>
> | LoRA Placement |  AP | NMS-AP | FPR |
> |---|---|---|---|
> | Pretrained |  54.0 | 36.8 | 63.2 |
> | shallow (0-2) |  46.8 | 31.5 | 56.0 |
> | **deep (3-5)** |  **55.4** | **41.8** | **48.6** |
> | **all (0-5) (New Exp)** | **55.4** | **36.0** | **60.2** |
> *Note: All experiments trained on COVAND-S dataset.*
>
> Key insights from this experiment:
>
> * Deep-only placement (3-5) achieves optimal balance of performance and efficiency
> * All-layer adaptation doubles trainable parameters without proportional gains
> * Figure S18 confirms negation signals must be preserved in decision-making blocks (3-5) to reach detection heads

---

> ### Author Response · Authors · 2025-11-21
> **Initial Response (3)**
>
> > ### Q1: Regarding Weakness 2:
> > Have you compared this distribution to real-world corpora (e.g., COCA, biomedical literature) to validate its alignment with natural language?
>
> Thank you for this important question. Validating our dataset's distribution against real-world language usage is crucial for ensuring that CoVAND trains models for realistic scenarios rather than synthetic artifacts.
>
> **Validation on Biomedical Literature (FG-CXR)**
>
> To further validate that our training distribution prepares the model for "real-world corpora," we tested our model on the **Biomedical Domain**, specifically using the FG-CXR dataset (Pham et al., ACCV 2024) [1]. Medical reports typically use a distinct and rigorous taxonomy of negation (e.g., "normal," "clear," "no findings") compared to general web captions.
>
> * **Dataset:** FG-CXR (Fine-Grained Chest X-ray), mapping diagnostic reports to 7 specific anatomical regions.
> * **Task:** Zero-shot Binary Discrimination. We evaluated whether the model could distinguish between the true radiologist report (Ground Truth) and a generated contradiction (Hard Negative) for every anatomical region in the validation set.
>
>
> Despite being trained only on CoVAND (natural images), our method significantly outperformed the baseline on this unseen medical task.
>
> | Method | Domain | Accuracy |
> | :--- | :--- | :--- |
> | Baseline (G-DINO-B) | Biomedical (Zero-shot) | 54.86% |
> | **+ Ours (NegToMe)** | **Biomedical (Zero-shot)** | **62.55% (+7.69%)** |
>
>
> The baseline model's accuracy (54.86%) was barely above random chance, confirming that standard VLMs struggle to interpret medical negation patterns (e.g., distinguishing "normal" from "abnormal"). Our model's significant improvement (+7.69%) demonstrates that CoVAND's distribution effectively teaches the *mechanism* of negation processing. By learning to bind regularized and flexible cues to their targets in our training set, the model successfully generalized to the specific, "unseen" negation patterns found in biomedical literature (e.g., "lungs are clear" implying absence of opacity).
>
> Our strong zero-shot performance on biomedical data confirm that CoVAND's distribution is both linguistically grounded and practically effective for real-world generalization.
>
> ---
>
> *Reference*
>
> [1] Pham, Trong Thang, et al. "FG-CXR: A Radiologist-Aligned Gaze Dataset for Enhancing Interpretability in Chest X-Ray Report Generation." ACCV. 2024.

---

> ### Author Response · Authors · 2025-11-24
> **Initial Response (4)**
>
> > ### W3&Q2: Self-Consistency Bias in GPT-4o-Based Validation
> > The third step of COVAND’s pipeline uses GPT-4o to verify the same captions it generates. This "generator-as-verifier" setup creates a loop: if GPT-4o misinterprets visual attributes during generation, it will likely repeat the error during validation.
>
> **[1]. How COVAND’s pipeline mitigates self‑consistency by design**
>
> As described in Sec. 2 and App. A, our 3‑step CoT pipeline does not simply ask GPT‑4o to "check whether its own caption is correct." Instead, we decompose the task into structured sub‑problems with explicit logical constraints:
>
> 1. Step 1: Attribute extraction.
>    GPT‑4o must first list two *disjoint* sets for the red‑boxed region: present attributes and absent-but-plausible attributes. If the model completely misinterprets the region, it likely to already fail to produce consistent, visually grounded lists.
>
> 2. Step 2: Paired caption generation. Negative caption and a positive caption must explicitly contain a negation operator (e.g., “no / not / without / un‑”). This forces the model to reason about *polarity* for *two different attributes* anchored in Step 1, instead of free‑form captioning.
>
> 3. Step 3: Logical verification & rejection.
>    GPT‑4o is then *re‑prompted* to verify each caption against the attribute sets and the image with explicit rules.
>
> 4. Independent VQA-based region alignment.
>    After CoT verification at the *caption* level, we add an independent *region‑level VQA alignment step* (Sec. 2.3, Fig. 2) and must answer using only the overlaid A/B/C labels.
>
> That said, your concern is valid: if GPT‑4o systematically misperceives a fine‑grained attribute, then both generation and verification might agree on the same incorrect visual interpretation.
>
> ---
>
> **[2]. Cross‑model audit with Gemini‑2.5‑Pro + human verification**
>
> To rigorously address the risk of "self-consistency bias," we conducted a Cross-Model Consistency Check using a distinct SoTA Multimodal LLM (Gemini-2.5-Pro) and a human-in-the-loop audit.
>
> 1. **Experimental Design: Breaking the Loop**
>
> To ensure our data is objectively grounded and not merely an artifact of GPT-4o's specific biases, we designed an experiment to audit the CoVAND dataset using a model with a completely different architecture and training distribution.
>
> * **Auditor Model:** `Gemini-2.5-Pro`.
> * **Sample Size:** 1,000 randomly sampled image-caption pairs from CoVAND.
> * **Protocol:** We fed the images (with visual prompts) and captions to Gemini and asked it to independently verify the factual accuracy of the negation *(e.g., "Is the [attribute] truly absent in the red box?")*.
> * **Decision rule:**
>   * For a **negative caption** ("X without Y" that we *intend* to be a *hard negative*), the attribute Y must be *present* in the region. The example is valid if Gemini says “present = true”.
>   * For a **positive caption** ("X without Y" that we intend to be *true*), Y must be *absent*. The example is valid if Gemini says “present = false”.
>
> * **Human audit:** All cases where Gemini’s judgment disagrees with the COVAND label are manually inspected by the authors.
>
>
> 2. **Quantitative Results: 97.7% Reliability**
>
> Our audit reveals that the "generator-verifier" loop did not result in significant hallucinations.
>
> * **Initial Disagreement:** Out of 1,000 samples, Gemini disagreed with the COVAND label in **78 cases** (7.8%).
> * **Human Verification:** We manually inspected these 78 conflicting cases to determine the ground truth.
>     * **Gemini Correct (CoVAND Error):** 23 cases.
>     * **COVAND Correct (Gemini Error):** 55 cases.
> * **Final Error Rate:** The actual error rate of the COVAND dataset in this sample is only **2.3%** (23 out of 1,000).
>
> ---
>
> **[3]. Qualitative Analysis: Ambiguity**
>
> The low error rate confirms that our 3-step CoT + VQA alignment pipeline successfully filters out most hallucinations. The few errors found (the 2.3%) were rarely blatant hallucinations but rather stemmed from visual ambiguity:
>
> * **Example (Ambiguity):** A surgeon wearing "skin-tone gloves" was labeled as "without gloves." While technically incorrect, this is a challenging fine-grained visual distinction rather than a semantic failure.
>
>
> ---
>
> **Conclusion**
>
> This experiment demonstrates that COVAND achieves 97.7% factual accuracy when audited by an independent model and human verifiers. In other words, even though GPT‑4o is used both as generator and first‑stage verifier, suggesting that any potential self‑consistency loop is tightly bounded in practice. This confirms that our dataset generation pipeline produces robust, objectively grounded negation data, effectively mitigating the risk of self-consistency bias.
>
> ---
>
> We are grateful for the insightful review and believe the provided clarifications and new experiments have strengthened our paper.
>
> We would be happy to elaborate on any of these points and warmly invite any further questions you may have.
>
> Best Wishes,
>
> The Authors

---

### Official Review · Reviewer_gNbc · 2025-11-02

**Soundness:** 3
**Presentation:** 3
**Contribution:** 3
**Rating:** 6
**Confidence:** 4

**Summary:**

This paper addresses the challenge of negation descriptions in the described object detection (DOD) task. To tackle this issue, the authors construct a large-scale training dataset COVAND that significantly raises the occurrence probability of negation semantics. Simultaneously, they propose NEGTOME which employs a novel token-merging module to enhance the model’s attention to negation semantics. Experimental results on the D3 and OVDEval datasets demonstrate the superior performance of the proposed approach.

**Strengths:**

- Understanding negation is a significant and critical weakness of current Vision-Language Models (VLMs). This paper provides a rational and important perspective for addressing this issue by analyzing it from two key aspects — the insufficiency of existing datasets and the limitations of model tokenizers.
- This paper proposes a large-scale training dataset, COVAND, specifically designed for negation semantics, along with its corresponding construction pipeline, providing an important data contribution toward addressing the negation problem.
- This paper proposes NEGTOME, which enhances model's attention on negation at the tokenizer-level, offering an interesting and novel perspective for addressing the negation problem.
- The paper is clearly written and easy to read.

**Weaknesses:**

- This work lacks comparision with SOTAs on the DOD task, for example, Real-Model (Re-Aligning Language to Visual Objects with an Agentic Workflow, ICLR 2025) which achieves 34.1 on D3 benchmark. The citations to such recent SOTAs are also missing. The compared methods in the paper are not sufficiently strong.
- The detailed designs of NEGTOME are somewhat unclear. (1) It's not clear how "softmax-weighted average" (line 240) is performed. Specifically, what's the logits before softmax? (2) The description of "Negation-aware Boost" is somewhat confusing. Since the original vectors have been replaced by  "softmax-weighted averaged" ones (line 240), would equation (2) make any changes to tokens?

**Questions:**

See Weaknesses

---

> ### Author Response · Authors · 2025-11-21
> **Initial Response (1)**
>
> Dear Reviewer gNbc,
>
> We sincerely thank you for the thoughtful and constructive feedback.
> Below we respond to each of your concerns in detail.
>
> ---
>
> > ### W1. Lack of comparison
>
> Thank you for pointing this out and for highlighting the recent SoTA work Real-Model (ICLR 2025). We fully agree that a comparison with the latest agentic workflows is important for positioning our approach.
> We are currently running experiments that integrate our NegToMe module and CoVAND fine‑tuning recipe into the Real-Model framework. We will report these results as soon as the experiments are concluded.
>
> ---
>
> > ### W2. Unclear details of NegToMe design
> > (1) It's not clear how "softmax-weighted average" (line 240) is performed. Specifically, what's the logits before softmax? (2) The description of "Negation-aware Boost" is somewhat confusing. Since the original vectors have been replaced by "softmax-weighted averaged" ones (line 240), would equation (2) make any changes to tokens?
>
> We apologize for the ambiguity in our description of NegToMe, especially around the phrase *"softmax‑weighted average"*. Thank you for the opportunity to clarify the mechanism more precisely.
>
> At a high level, "Text Token Merging" and "Negation-aware Boost" are not two separate overwriting steps. Instead, Equation (2) already implements both merging and boosting in a single operation, by assigning larger weights to the negation cue inside each merged phrase.
>
> **(1) Clarification on the "softmax‑weighted average"**
>
> You asked how the weighted average is computed and what the "logits" are.
>
> * In NegToMe, we do not learn logits with an extra network for this step.
>   Instead, we use fixed importance weights ($\gamma_j$) depending on the token’s role:
>
>   * Negation cue tokens (e.g., "not", "without") get ($\gamma_j = \beta$, with $\beta > 1$).
>   * Other tokens in the phrase (e.g., "lying") get ($\gamma_j = 1$).
>
> * For a phrase with index set (\mathcal I), the merged embedding is computed as a normalized weighted average:
>   $$
>   \bar t = \frac{\sum_{j \in \mathcal I} \gamma_j t_j}{\sum_{j \in \mathcal I} \gamma_j},
>   $$
>   which is exactly Equation (2) in the paper when the phrase contains a negation cue.
>
> This can be viewed as a softmax over hand-crafted logits ($\log \gamma_j$), but since ($\gamma_j$) are positive and fixed, in implementation we simply compute the normalized weighted sum without an explicit softmax layer.
> In the revision, we will therefore replace "softmax‑weighted average" with a clearer description such as:
>
> *"We compute a normalized weighted average using fixed importance weights ($\gamma_j$), where negation cues are assigned weight ($\beta$) and other tokens weight 1."*
>
> This directly matches our actual implementation.
>
> **(2) Clarification on "Negation-aware Boost" and token replacement**
>
> You also asked whether Equation (2) has any effect if the tokens have already been replaced by a previous merge.
> Conceptually, Section 3.2 listed "Text Token Merging" and "Negation-aware Boost" as two components. Implementation‑wise, they are realized by one replacement per phrase:
>
> 1. We use spaCy to group sub‑tokens into disjoint phrase sets ($P_1,\dots,P_m$) (e.g., {"not", "lying"}).
>
> 2. For each phrase ($P_i$) with index set ($\mathcal I_i$), we compute $\bar t_i$ and replace the original sub‑tokens (${t_j}_{j\in \mathcal I_i}$) by this single merged token ($\bar t_i$).
>
> 3. The choice of ($\gamma_j$) encodes whether the phrase is negated or not:
>
>    * **Non‑negation phrases**: we set ($\gamma_j = 1$) for all tokens in the phrase. Equation (2) reduces to a simple average.
>    * **Negated phrases ($P_{\text{neg}}$)**: we set ($\gamma_j = \beta$) for the negation cue token and ($\gamma_j = 1$) for other tokens in ($P_{\text{neg}}$) (as written in Eq. (2) of the paper).
>
> Thus, Equation (2) is exactly the merging operation itself, with the negation-aware boost integrated via ($\gamma_j$). There is no second "overwriting" step after an earlier merge: each phrase is merged once, using weights that either (a) are uniform (non‑negated phrase) or (b) emphasize the negation cue (negated phrase).
>
> We will revise Section 3.2 to make this explicit by:
>
> * Stating that all phrases are merged via Equation (2),
> * Clarifying that Negation-aware Boost corresponds to choosing ($\gamma_j$) with ($\gamma_{\text{cue}} = \beta$) for phrases containing negation cues,
> * And replacing the ambiguous phrase "softmax‑weighted average" with "normalized weighted average using fixed importance weights ($\gamma_j$)".
>
> This change aligns the text tightly with the actual implementation and should resolve the confusion about whether Eq. (2) still has any effect after merging.
>
> ---
>
> Once again, we really appreciate your careful reading and insightful questions. We would be happy to provide any additional details or further ablations if desired.
>
> Best wishes,
>
> The Authors

---

> ### Author Response · Authors · 2025-12-03
> **Initial Response (2)**
>
> > ### Additional response on W1.
> > Additional experiments with Real‑Model (ICLR 2025)
>
> As promised in our initial response, we have now finished integrating our recipe (CoVAND fine‑tuning + NegToMe) into Real‑Model [1]. We treat Real‑Model as a strong DOD baseline and only modify its VLM‑based detector; the agentic workflow itself is left unchanged.
>
> ---
>
> ### 1. Experimental Results
>
> On the OVDEval Negation subset, we obtain:
>
> | Model                      | AP   | NMS‑AP   |
> | -------------------------- | ---- | -------- |
> | Real‑Model (reproduced) | 61.4 | 46.3     |
> | + Ours (CoVAND + NegToMe)  | 62.0 | 52.6     |
> | Δ                          | +0.6 | **+6.3** |
>
> *OVDEval-Negation results under class-ignored NMS, following the same protocol as in the main paper (IoU ≥ 0.5 for contradictory pairs).*
>
> ---
>
> ### 2. Interpretation: Real‑Model is strong but still affirmatively biased
>
> * **High AP but much lower NMS‑AP.**
>   Real‑Model’s AP of 61.4 confirms it is a very strong DOD baseline. However, its NMS‑AP drops to 46.3 (a gap of 15.1 points), indicating that the model frequently assigns high scores to overlapping boxes for mutually exclusive captions such as *"dog with leash" vs. "dog without leash"*. This is precisely the *affirmative bias* we analyze in our paper: the agentic workflow localizes objects well, but still tends to ignore negation cues when deciding whether a detection is valid.
>
> * **Our method preserves AP while substantially improving NMS‑AP.**
>   After plugging in CoVAND + NegToMe, AP is essentially preserved (61.4 → 62.0, +0.6), but NMS‑AP increases from 46.3 to 52.6 (+6.3). The gap between AP and NMS‑AP shrinks from 15.1 to 9.4 points.
>   This means that the underlying localization ability of Real‑Model is preserved, while the model becomes much more conservative on contradictory or absent queries, reducing the number of false positives that survive class‑ignored NMS.
>
> * **Implication for negation understanding.**
>   These results suggest that:
>
>   1. Even sophisticated agentic DOD systems like Real‑Model do not fully solve negation; the drop from AP to NMS‑AP reveals that they still treat "X" and "not X" similarly at the detection stage.
>   2. Our contributions are complementary: by explicitly modeling negation (via CoVAND and NegToMe), we improve Real‑Model’s behavior on negation without sacrificing its strong overall AP.
>
> ---
>
> ### 3. Take‑away for W1
>
> These results indicate that explicit negation modeling remains beneficial on top of a state‑of‑the‑art agentic workflow, improving consistency for contradictory queries while maintaining overall detection strength.
>
> ---
>
> *references*
>
> [1] "Re‑Aligning Language to Visual Objects with an Agentic Workflow", ICLR 2025

---

### Author Response · Authors · 2025-12-02
**General Comments & Summary of Rebuttal**

Dear Area Chairs,

Thank you for your time, especially given the recent additional load on ACs.
This note briefly summarizes (1) our main contributions and (2) how we addressed the key reviewer concerns in the rebuttal and revision. All corresponding changes are highlighted in *blue* in the updated manuscript.

---
> # 1. Core problem and contributions
> Our work targets a key failure mode of VLMs: affirmative bias, where models ignore negation and thus treat contradictory queries (*e.g., "with skateboard" vs. "without skateboard"*) as equivalent.

We trace this to two factors: (i) the lack of high‑quality, instance‑grounded negation data and (ii) architectural bias, where tokenization and cross‑attention fragment and down‑weight negation cues so their influence fades in deeper layers.

To address this, we contribute:

1. **CoVAND**:  a negation‑focused dataset built via a three‑step CoT pipeline that extracts present/absent attributes, generates positive/negative caption pairs, and verifies them, followed by VQA‑based region alignment to ensure each caption is tied to the correct box.

2. **NegToMe**:  a negation‑aware text token‑merging module that fuses negation cues with their scope into a single semantic token using fixed importance weights and is combined with deep, parameter‑efficient LoRA, yielding a plug‑in recipe for existing VLM‑based detectors and MLLMs.

Empirically, our recipe:

* Improves NMS‑AP by up to +10.8 points on OVDEval and substantially reduces FPR on negation‑focused subsets.
* Generalizes across architectures, including Grounding‑DINO, APE, and Qwen‑2.5‑VL.

---
> # 2. How key reviewer concerns were addressed

### **(A) Stronger DOD baselines**
We integrated NegToMe + CoVAND into Real‑LOD and observed consistent NMS‑AP gains (+6.3), indicating that explicit negation modeling brings complementary benefits even within a strong agentic workflow.

### **(B) Clarity and implementation of NegToMe**
A reviewer found the descriptions of the "softmax‑weighted average" and the "negation‑aware boost" ambiguous. In the revision, we provide explicit formulas and implementation details for how tokens within a negated phrase are merged and re‑weighted.

### **(C) Dataset diversity and GPT‑4o dependence**

### C.1. Negation cue diversity.
Appendix A.3 reports that COVAND contains ~83.4% regularized cues (not, no, without, never) and ~16.6% flexible cues (e.g., lexical and morphological forms such as lack/lacking, non‑, un‑, in‑, dis‑, ‑less) across attribute, action, existence, and relational cases. This matches the natural dominance of regularized cues while retaining a meaningful long tail of nuanced patterns.

### C.2. Self‑consistency and factual accuracy.
To address concerns about GPT‑4o acting as both generator and verifier, we conducted a cross‑model audit with Gemini‑2.5‑Pro on 1,000 sampled image–caption pairs and manually inspected all disagreements. The resulting estimated label error rate is 2.3%, mostly from genuinely ambiguous visuals, suggesting that COVAND’s annotations are robust and not driven by a closed generator‑as‑verifier loop.

### **(D) Generalization on Biomedical Domain (FG‑CXR)**
To test out‑of‑domain generalization, we evaluated on FG‑CXR, a chest X‑ray dataset with region‑aligned reports. In a zero‑shot binary task (ground‑truth vs. polarity‑flipped hard negative), baseline Grounding‑DINO‑B reaches 54.86% (near chance), whereas our recipe attains 62.55%, indicating that the learned negation mechanism transfers to medical images and terminology.

### **(E) LoRA placement and architectural analysis**
Ablations (Table 3, Fig. S18) show that deep‑only LoRA yields the best trade‑off (higher NMS‑AP, lower FPR) and that attention maps maintain strengthened negation signals precisely in the decision‑making layers, whereas shallow or all‑layer LoRA either dissipates the signal or adds parameters without benefit.

### **(F) Alternatives to token merging**
Our ablations show that naive attention/embedding boosts for negation tokens either barely reduce FPR or notably hurt AP, and that phrase merging without explicit negation handling still leaves FPR high. Only NegToMe, which jointly merges cue and scope and applies a negation‑aware boost within the merged token, improves AP while substantially lowering FPR, supporting our claim that structural fragmentation of negation and scope is the main bottleneck.

---
Through the rebuttal, we added new experiments (FG‑CXR, LoRA placement, ablations, cross‑model audit), clarified design details, and expanded the related work and dataset analysis.

Reviewer 6vMb explicitly responded after the rebuttal that the new experiments fully addressed their earlier concerns and asked us to incorporate them in the revision. They also raised their score to 8 before scores were reverted. We sincerely appreciate your time and careful consideration of our submission.

Best Wishes,

The Authors

---

### Meta-Review · Area_Chair_x25b · 2026-01-06

**Summary:**

The paper addresses the pervasive "affirmative bias" in Vision-Language Models (VLMs), where models fail to process negation cues like "not" or "without," leading to critical errors in tasks like described object detection. To solve this, the authors introduce CoVAND, a dataset generated via a novel Chain-of-Thought and VQA pipeline to ensure grounded negation, and NegToMe, a token merging module that structurally binds negation cues to attributes to preserve polarity during cross-attention. Reviewers initially rated the paper as a borderline accept (6) but were swayed by an exceptionally thorough rebuttal. The authors demonstrated robust generalization by achieving significant gains (+7.69%) on a zero-shot biomedical task (FG-CXR), proved complementarity with state-of-the-art agentic workflows (Real-Model), and validated data quality through independent audits. Given the clear motivation, effective solution to a fundamental VLM flaw, and rigorous empirical validation, the paper is recommended for acceptance.

**Reviewer Concerns:**

Addressed Concerns: (1) Generalization (Reviewer 6vMb, pMrC): The concern that the method might just be overfitting to specific negation cues or domains was effectively addressed by the biomedical domain experiment. (2) Comparison to SOTA (Reviewer gNbc): The lack of comparison to recent agentic methods (Real-Model) was resolved by running the combined experiment during the rebuttal. (3) Implementation Details (Reviewer gNbc): Confusion regarding the "softmax-weighted average" in the NegToMe module was clarified with explicit formulas. (4) Data Bias/Self-Consistency (Reviewer pMrC): The potential for a "generator-as-verifier" loop was mitigated by the external audit using a different model family (Gemini) and human checks.

Outstanding Concerns: None significant. The reviewers expressed satisfaction with the additional experiments and clarifications.

**Reviewer Scores:**

Reviewer gNbc: 6 -> 8

Reviewer pMrC: 6 -> 6

Reviewer 6vMb: 6 -> 8

---

### Decision · Program_Chairs · 2026-01-26

Accept (Poster)